# Transplacental SARS-CoV-2 protein ORF8 binds to complement C1q to trigger fetal inflammation

Tamiris Azamor[1,10], Débora Familiar-Macedo [1,10], Gielenny M Salem [1], Chineme Onwubueke[1,2], Ivonne Melano[1], Lu Bian[1], Zilton Vasconcelos [3], Karin Nielsen-Saines [3,4], Xianfang Wu [1,2], Jae U Jung[1,2,5], Feng Lin[6], Oluwatosin Goje[7], Edward Chien [7], Steve Gordon[8], Charles B Foster[8], Hany Aly [9], Ruth M Farrell [7,11✉], Weiqiang Chen [1,2,11✉] & Suan-Sin Foo [1,2,11✉]

## Abstract

**Prenatal SARS-CoV-2 infection is associated with higher rates of pregnancy and birth complications, despite that vertical transmission rates are thought to be low. Here, multi-omics analyses of human placental tissues, cord tissues/plasma, and amniotic fluid from 23 COVID-19 mother-infant pairs revealed robust inflammatory responses in both maternal and fetal compartments. Pronounced expression of complement proteins (C1q, C3, C3b, C4, C5) and inflammatory cytokines (TNF, IL-1α, and IL-17A/E) was detected in the fetal compartment of COVID-19-affected pregnancies. While ~26% of fetal tissues were positive for SARS-CoV-2 RNA, more than 60% of fetal tissues contained SARS-CoV-2 ORF8 proteins, suggesting transplacental transfer of this viral accessory protein. ORF8-positive fetal compartments exhibited increased inflammation and complement activation compared to ORF8-negative COVID-19 pregnancies. In human placental trophoblasts in vitro, exogenous ORF8 exposure resulted in complement activation and inflammatory responses. Co-immunoprecipitation analysis demonstrated that ORF8 binds to C1q specifically by interacting with a 15-peptide region on ORF8 (C37-A51) and the globular domain of C1q subunit A. In conclusion, an ORF8-C1q-dependent complement activation pathway was identified in COVID-19-affected pregnancies, likely contributing to fetal inflammation independently of fetal virus exposure.**

**Keywords** Classical Complement Activation; Fetal Inflammation; Pregnancy; SARS-CoV-2; Transplacental ORF8
**Subject Categories** Development; Immunology; Microbiology, Virology & Host Pathogen Interaction

## Introduction

Infections caused by severe acute respiratory syndrome coronavirus 2 (SARS-CoV-2) over the past four years resulted in nearly 800 million cases and 7 million deaths worldwide, with continued cases and mortality occurring even after the public health emergency declaration for Coronavirus disease 2019 (COVID-19) ended in May 2023 (CDC, 2023; WHO, 2023). Despite advances in therapeutic and vaccine strategies, a significant knowledge gap persists regarding the mechanisms underlying various pathogenic aspects of COVID-19, making it an ongoing threat to humanity, especially in the context of pregnancy (Douglass et al, 2021; Ellington et al, 2020; Yu et al, 2020). So far, case-control studies demonstrated that pregnant women with COVID-19 present a higher risk for maternal, neonatal, and perinatal morbidity and mortality (Villar et al, 2023). Emerging clinical evidence has also shown that prenatal exposure to SARS-CoV-2 infection results in a two-fold increase in risk of future neurodevelopmental delay in some exposed infants (Edlow et al, 2022; Fajardo Martinez et al, 2023; Rasile et al, 2022).

The maternal-fetal interface is composed of maternal-skewed tissues (choriodecidua), fetal-skewed tissues (amnion, umbilical cord), and fluids (amniotic fluid, cord blood). The interplay among those two sides is critical to fetal outcomes. In general, exacerbated maternal immune activation has been associated with infant neurodevelopmental delay (Boulanger-Bertolus et al, 2018). At the choriodecidua, activation of maternal leukocytes leads to cytokine production from decidual trophoblasts such as lambda interferon. These are critical in determining fetal outcomes in other viral infections during pregnancy. In contrast, trophoblasts from the fetal compartment of the placenta seem to be less responsive to viral infections (Azamor et al, 2024). Our prior study found that COVID-19 during pregnancy alters immune responses in both mothers and newborns (Foo et al, 2021). A premature immune maturation as a result of placental immunological crosstalk

[1]Infection Biology Program, Global Center for Pathogen Research and Human Health, Lerner Research Institute, Cleveland Clinic, Cleveland, OH, USA. [2]Cleveland Clinic Lerner College of Medicine of Case Western Reserve University, Cleveland, OH, USA. [3]Fundação Oswaldo Cruz, Rio de Janeiro, Brazil. [4]Department of Medicine, Division of Infectious Diseases, David Geffen School of Medicine, University of California, Los Angeles, Los Angeles, CA, USA. [5]Department of Cancer Biology, Lerner Research Institute, Cleveland Clinic, Cleveland, OH, USA. [6]Inflammation and Immunity, Lerner Research Institute, Cleveland Clinic, Cleveland, OH, USA. [7]Department of Obstetrics and Gynecology, Obstetrics and Gynecology Institute, Cleveland Clinic, Cleveland, OH, USA. [8]Section of Pediatric Infectious Diseases, Children's Institute, Cleveland Clinic Children's, Cleveland Clinic, Cleveland, OH, USA. [9]Cleveland Clinic Children's, Cleveland Clinic, Cleveland, OH, USA. [10]These authors contributed equally: Tamiris Azamor, Débora Familiar-Macedo. [11]These authors jointly supervised this work: Ruth M Farrell, Weiqiang Chen, Suan-Sin Foo. ✉E-mail: farrellr@ccf.org; chenw3@ccf.org; foos@ccf.org

(Foo et al, 2021) revealed structural changes, including villous chorangiosis, acute chorioamnionitis, and chronic villitis (Blasco Santana et al, 2021). Diffuse perivillous fibrin and infiltration of inflammatory CD3 + T cells and CD68+ macrophages along with high expression of pro-inflammatory and complement C1q genes have also been observed (Hosier et al, 2020; Sureshchandra et al, 2022). In fact, complement is tightly regulated at the maternal-fetal interface throughout pregnancy stages. It consists of three pathways—classical (C1q-dependent), lectin, and alternative (Amari Chinchilla et al, 2020). The excessive activation of one of these complement pathways during pregnancy can lead to pregnancy complications, such as preeclampsia (Sureshchandra et al, 2022). In the context of COVID-19, transcriptomics analysis of placenta villi, which predominantly consist of fetal trophoblasts, revealed upregulated expression of complement genes (Amari Chinchilla et al, 2020). However, the exact role of complement activation in COVID-19-affected pregnancies remains poorly understood.

As SARS-CoV-2 vertical transmission is rare, the altered immunity observed in COVID-19-affected pregnancies is potentially triggered by a viral replication-independent mechanism (Ezechukwu et al, 2022). In this context, SARS-CoV-2 ORF8 protein, a secreted viral cytokine with an immunoglobulin-like domain, has been well-associated with COVID-19 severity and inflammatory responses (Wu et al, 2023; Wu et al, 2022). A Singaporean clinical cohort study identified the circulation of a SARS-CoV-2 variant with a deletion in the ORF8 genome region, associated with reduced COVID-19 severity and inflammation compared with the wild-type strain (Young et al, 2020). Patients with severe COVID-19 in the intensive care unit (ICU) had high serum ORF8 levels (Wu et al, 2022). In addition, ORF8 contributes to SARS-CoV-2 virulence by inhibiting host defenses, including MHC class I antigen presentation, endoplasmic reticulum stress, and type I interferon signaling (Vinjamuri et al, 2022). Specifically, ORF8 mimics IL-17, directly binding to host receptors IL17-RA/B/C to induce pro-inflammatory IL-17 signaling (Wu et al, 2022).

Herein, we conducted a multi-omics strategy investigating the immune crosstalk in the maternal-fetal interface exposed to SARS-CoV-2 using a diversity of biospecimens collected at term from 23 mother-infant pairs as compared to 6 control pairs to investigate whether residual viral components may be responsible for the inflammatory processes which potentially lead to adverse maternal and infant repercussions. Despite vertical transmission events comprising ¼ of cases, both maternal and fetal compartments of COVID-19 placentas displayed significant inflammation, specifically showing overt complement activation on the fetal side. Furthermore, we detected elevated levels of SARS-CoV-2 ORF8 protein in maternal plasma, cord plasma, amniotic fluid, and amnion tissue from COVID-19-affected pregnancies, which correlated with increased complement activation and inflammation. Mechanistically, we found that ORF8 can directly stimulate placental trophoblasts to trigger complement activation and inflammation. Importantly, we identified the specific interaction between ORF8 and complement C1q complex, highlighting a 15-peptide region on ORF8 (C37-A51) and the globular domain of C1q subunit A as the interaction sites. These findings offer insight into potential therapeutic targets for mitigating COVID-19-related immune dysregulation in pregnancy.

# Results

## Clinical characteristics of the COVID-19 pregnancy cohort

A total of 29 mother-infant dyads were enrolled in this study, consisting of (i) 6 pregnant controls with a SARS-CoV-2 negative diagnosis and absence of clinical symptoms and (ii) 23 COVID-19 pregnant patients (Fig. 1A, Table 1). None of the patients who tested positive for SARS-CoV-2 were hospitalized for COVID-19. Within the COVID-19 group, among 21 patients with data on hypertension, three (14.3%) had preeclampsia, while this was not observed in any of the patients in the control group. Among SARS-CoV-2 positive participants, two patients (8.7%) tested positive in the first trimester, five (21.7%) in the second, and 16 (69.6%) in the third. The average gestational age at delivery and fetal weight were similar between exposed and non-exposed groups. One neonate was born prematurely in the COVID-19 group (34 weeks and 2 days). All other neonates in both groups were term infants (born at or after 37 weeks). For fetuses of mothers infected in trimesters 1, 2, and 3, the average number of weeks between infection and delivery was $28.9 \pm 2.8$, $15.0 \pm 2.9$, and $4.6 \pm 3.8$, respectively. Within the COVID-19 group, among seven patients with data on placental pathology, three (42.9%) had chorioamnionitis, three (42.9%) had villitis, and two (28.6%) had deciduitis. None placenta pathology was observed in patients that presented SARS-CoV-2 infection in the first trimester. Within the control group, two patients had placental pathology evaluations, with none of these conditions observed (Table 1).

## SARS-CoV-2 is poorly detected in the maternal-fetal interface of COVID-19-affected pregnancies

Biospecimens from both groups of participants were collected at delivery, which included maternal plasma, newborn whole blood (WB), newborn plasma, cord plasma, amniotic fluid, umbilical cord, maternal-dominant chorion placental tissues, and fetal-dominant amnion tissues. The biospecimens collected were subjected to (i) ultrasensitive detection of SARS-CoV-2 viral RNA (vRNA) using droplet digital (dd) PCR, (ii) global transcriptomics using bulk RNAseq, (iii) secreted protein using targeted bead-based multiplexing technology to access the proteomics and (iv) circulating ORF8 viral protein using ELISA (Fig. 1A). To determine the rate of vertical transmission in the present pregnancy cohort, we first performed ultrasensitive ddPCR detection of SARS-CoV-2 Nucleocapsid (N) 1 and N2 vRNA in the biospecimens obtained from control and COVID-19 pregnancies. We evaluated viral load in the maternal (maternal plasma, chorion) and fetal compartments (amnion, amniotic fluid, cord plasma, and newborn plasma), considering the cutoff of 2 counts/mL for a SARS-CoV-2-positive interpretation (Fig. 1B). The absolute quantitation of both N1 and N2 copies resulted in the detection of at least one SARS-CoV-2-positive sample in all biospecimen types analyzed, and all samples from control group tested negative for N1 and N2 (Fig. 1B). Specifically, 24% of the chorion ($n = 5/21$), 20% of cord plasma ($n = 4/20$), 11% of amniotic fluid ($n = 1/9$), 10% of amnion ($n = 2/21$), 8% of newborn plasma ($n = 1/12$) and 4% of maternal plasma (4%, $n = 1/23$) were positive for

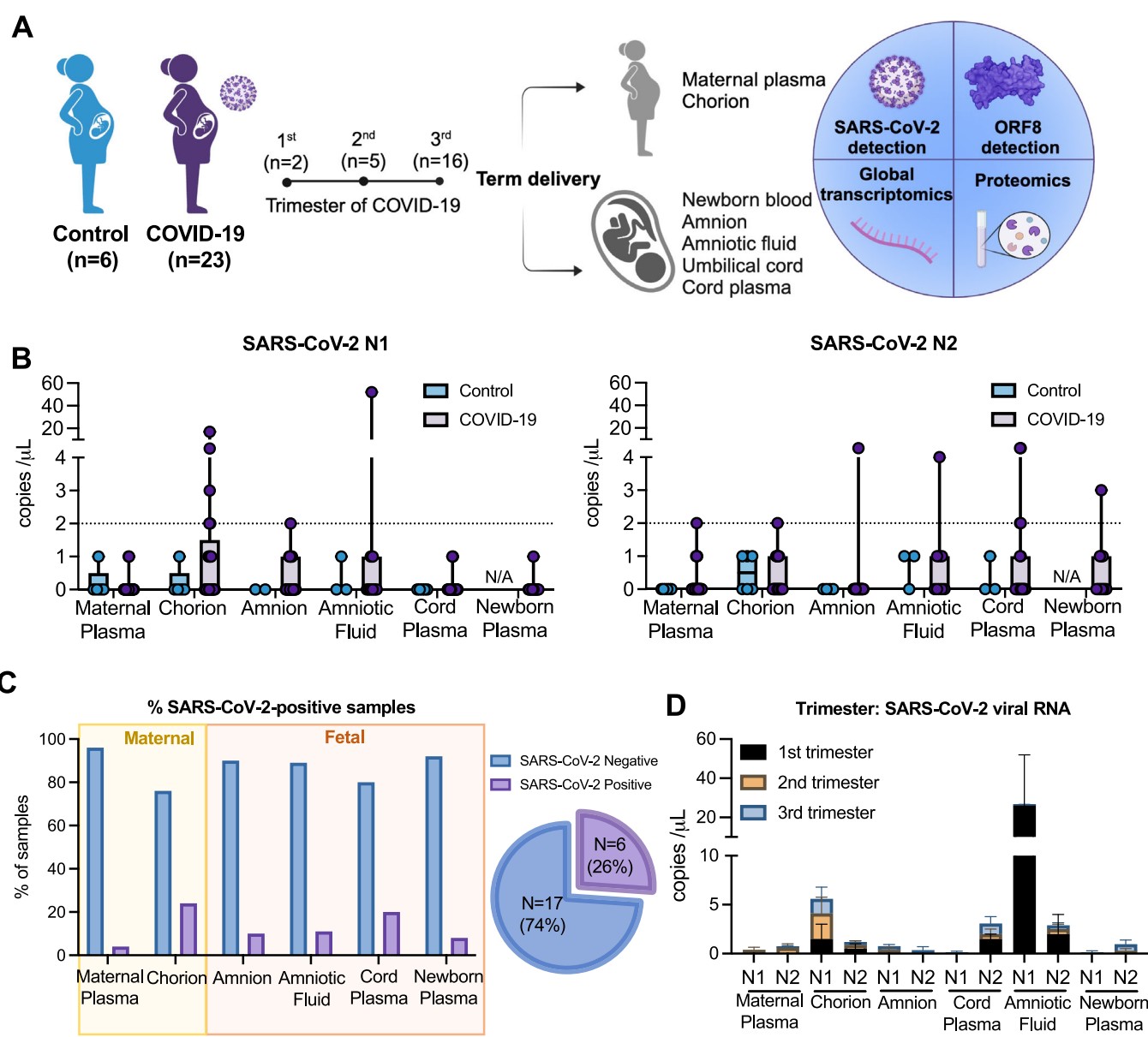

**Figure 1. Vertical transmission rate in maternal-fetal biospecimens derived from COVID-19-affected pregnancy cohort.**

(A) Overview of the COVID-19 pregnancy cohort (total $n = 29$). Pregnant women included in the study had negative (Control, $n = 6$), or positive SARS-CoV-2 diagnoses (COVID-19 $n = 23$) during the first ($n = 2$), second ($n = 5$), or third ($n = 16$) trimesters of pregnancy. Participants had biospecimens collected at the time of the delivery. Biospecimens collected included (i) biofluid specimens: maternal plasma, newborn blood, cord plasma, and amniotic fluid; and (ii) placental tissues: umbilical cord, chorion, and amnion. The biospecimens underwent ultrasensitive SARS-CoV-2 detection using digital droplet PCR (ddPCR), SARS-CoV-2 ORF8 ELISA, proteomics profile for 97 inflammatory biomarkers, and global transcriptomics profile (~22,000 host genes) by RNAseq. (B) Box and whiskers plot representing average and minimum and maximum values of copies/mL of SARS-CoV-2 N1 and N2 proteins detected maternal plasma (controls, $n = 5$ for N1; $n = 6$ for N2; COVID-19, $n = 23$ for N1/N2), chorion (controls, $n = 5$ for N1, $n = 6$ for N2; COVID-19, $n = 21$), amnion (controls, $n = 2$ for N1; $n = 4$ for N2; COVID-19, $n = 21$), amniotic fluid (controls, $n = 3$; COVID-19, $n = 9$), cord plasma (controls, $n = 5$ for N1, $n = 3$ for N2; COVID-19, $n = 20$), newborn plasma (controls, $n = 0$; COVID-19, $n = 12$) from controls and COVID-19-affected pregnancies. (C) Bar charts representing the percentage of samples positive (purple bar) and negative (blue bar) from SARS-CoV-2 in the biospecimens analyzed. Pie chart representing the total SARS-CoV-2 positivity considering at least one fetal-skewed specimen. Cut-off value defined as counts/mL$^3 \geq 2$. (D) Stacked bar charts (mean with SD) representing the levels of viral RNA of samples positive for SARS-CoV-2 in the fetal compartment, according to the trimester of pregnancy that maternal SARS-CoV-2 occurred (First trimester, $n = 2$; Second trimester, $n = 5$ and Third trimester, $n = 16$). Source data are available online for this figure.

SARS-CoV-2 vRNA. Therefore, our COVID-19 pregnant cohort presented a vertical transmission rate of 26% ($n = 6/23$), considering the detection of SARS-CoV-2 vRNA in at least one of the biospecimens from the fetal compartment (amnion, amniotic fluid, cord plasma, or

newborn plasma) per each COVID-19-affected pregnancy (Fig. 1C). We observed a higher viral load in the first trimester specifically in amniotic fluid samples to N1 and cord plasma and amniotic fluid to N2. Regarding the frequency of positivity for SARS-CoV-2 (N1, N2) in

**Table 1.  Clinical characteristics of COVID-19 pregnant cohort.**

| Maternal characteristics | Control (N = 6) | COVID-19 (N = 23) |
|---|---|---|
| Race, N (%) | | |
| - White, non-Hispanic | 6 (100%) | 15 (65.2%) |
| - Black, non-Hispanic | 0 (0%) | 3 (13.0%) |
| - Asian | 0 (0%) | 1 (4.3%) |
| - Multiracial, Hispanic | 0 (0%) | 3 (13.0%) |
| - Multiracial, non-Hispanic | 0 (0%) | 1 (4.3%) |
| Trimester of Infection, N (%) | | |
| - First trimester | N/A | 2 (8.7%) |
| - Second Trimester | N/A | 5 (21.7%) |
| - Third Trimester | N/A | 16 (69.6%) |

| Fetal characteristics | Control (N = 6) | COVID-19 (N = 23) |
|---|---|---|
| Average infection to delivery time, weeks mean (±SD) | | |
| - First | N/A | 28.9 (±2.8) |
| - Second | N/A | 15.0 (±2.9) |
| - Third | N/A | 4.6 (±3.8) |
| Average gestational age at delivery, weeks mean (±SD) | 39.5 (±0.7) | 38.9 (±1.4) |
| Average fetal weight, g mean (±SD) | 3527 (±211) | 3380 (±523) |
| Preterm birth, N (%) | 0 (0%) | 1 (4.3%) |

| Pregnancy complications | Control (N = 6) | COVID-19 (N = 23)[a] |
|---|---|---|
| Hypertension, N (%), Trimester of infection | 1 (16.7%) | 8 (38.1%) |
| - First | 0 (0%) | 0 (0%) |
| - Second | 0 (0%) | 2 (9.5%) |
| - Third | 1 (16.7%) | 6 (28.6%) |
| Preeclampsia, N (%), Trimester of infection | 0 (0%) | 3 (14.3%) |
| - First | 0 (0%) | 0 (0%) |
| - Second | 0 (0%) | 1 (4.8%) |
| - Third | 0 (0%) | 2 (9.5%) |

| Placental characteristics | Control (N = 2)[b] | COVID-19 (N = 7)[c] |
|---|---|---|
| Chorioamnionitis, N (%), Trimester of infection | 0 (0%) | 3 (42.9%) |
| - First | 0 (0%) | 0 (0%) |
| - Second | 0 (0%) | 1 (14.3%) |
| - Third | 0 (0%) | 2 (28.6%) |
| Villitis, N (%), Trimester of infection | 0 (0%) | 3 (42.9%) |
| - First | 0 (0%) | 0 (0%) |
| - Second | 0 (0%) | 0 (0%) |
| - Third | 0 (0%) | 3 (42.9%) |

**Table 1.  (continued)**

| Placental characteristics | Control (N = 2)[b] | COVID-19 (N = 7)[c] |
|---|---|---|
| Deciduitis, N (%), Trimester of infection | 0 (0%) | 2 (28.6%) |
| - First | 0 (0%) | 0 (0%) |
| - Second | 0 (0%) | 0 (0%) |
| - Third | 0 (0%) | 2 (28.6%) |

[a]Only 21 of 23 patients have data regarding pregnancy complications including preeclampsia and hypertension.
[b]Only 2 of 6 patients have data regarding placental characteristics.
[c]Only 7 of 23 patients have data regarding placental characteristics.

fetal samples, we observed 50% (1/2) of detection in the first trimester with 188 days after the infection, 20%(1/5) of detection in the second trimester with 98 days after the infection, and 25%(4/16) of detection in the third trimester with an average of 23.8 days after the infection (Fig. 1D, Tables EV1 and EV2).

## Distinct inflammatory transcriptomics profiles in maternal and fetal placental tissues

Next, we aimed to characterize distinctively altered immunological landscapes of specific maternal and fetal placental compartments during COVID-19-affected pregnancies. We performed bulk RNAseq analysis on 74 placental-derived tissues, including (i) maternal-skewed chorion/choriodecidua (Control, $n = 5$; COVID-19, $n = 21$), (ii) fetal-skewed amnion (Control, $n = 3$; COVID-19, $n = 18$) tissues (Fig. 2A), and (iii) umbilical cord tissues (control, $n = 6$; COVID-19, $n = 21$) (Fig. EV1A).

Chorion tissues derived from COVID-19 pregnancies presented 3129 upregulated and 1400 downregulated differentially expressed genes (DEGs) compared to controls (Fig. 2B). In contrast, amnion tissues from COVID-19 pregnancies showed a smaller subset of DEGs compared to chorion, with 1158 upregulated and 1075 downregulated DEGs relative to controls (Fig. 2C). Interestingly, despite the close anatomic proximity of the chorion and amnion membrane tissues in the placenta, their immune landscapes were notably different. We identified only 588 DEGs commonly affected in both chorion and amnion, with 3941 DEGs exclusively altered in chorion and 1645 DEGs exclusively altered in the amnion (Fig. 2D). The enrichment analysis performed on the DEGs that were commonly upregulated in both chorion and amnion placental membranes from COVD-19-affected pregnancies revealed a broad activation of various inflammation-related pathways. Among these pathways were responses involving Tumor necrosis factor (TNF), NF-kB, Interleukin (IL)-1, and IL-6, as shown in Fig. 2E.

Despite similar inflammatory responses elicited in both chorion and amnion, upregulated DEGs exclusively found in COVID-19 maternal-chorion were enriched for antiviral ISG-15, MDA-5, RIG-I, and interferon pathways (Fig. 2F). In contrast, upregulated DEGs exclusively presented in COVID-19 fetal-amnion exhibited pathways associated with leukocyte (lymphocytes, granulocytes) chemotaxis and activation (Fig. 2G). The distinctive immune landscapes of these placental tissues were reflected in the heatmap, showing an upregulated trend of inflammatory genes in both maternal-chorion and fetal-amnion, while IFN-related genes and chemokines were exclusively induced in chorion and amnion,

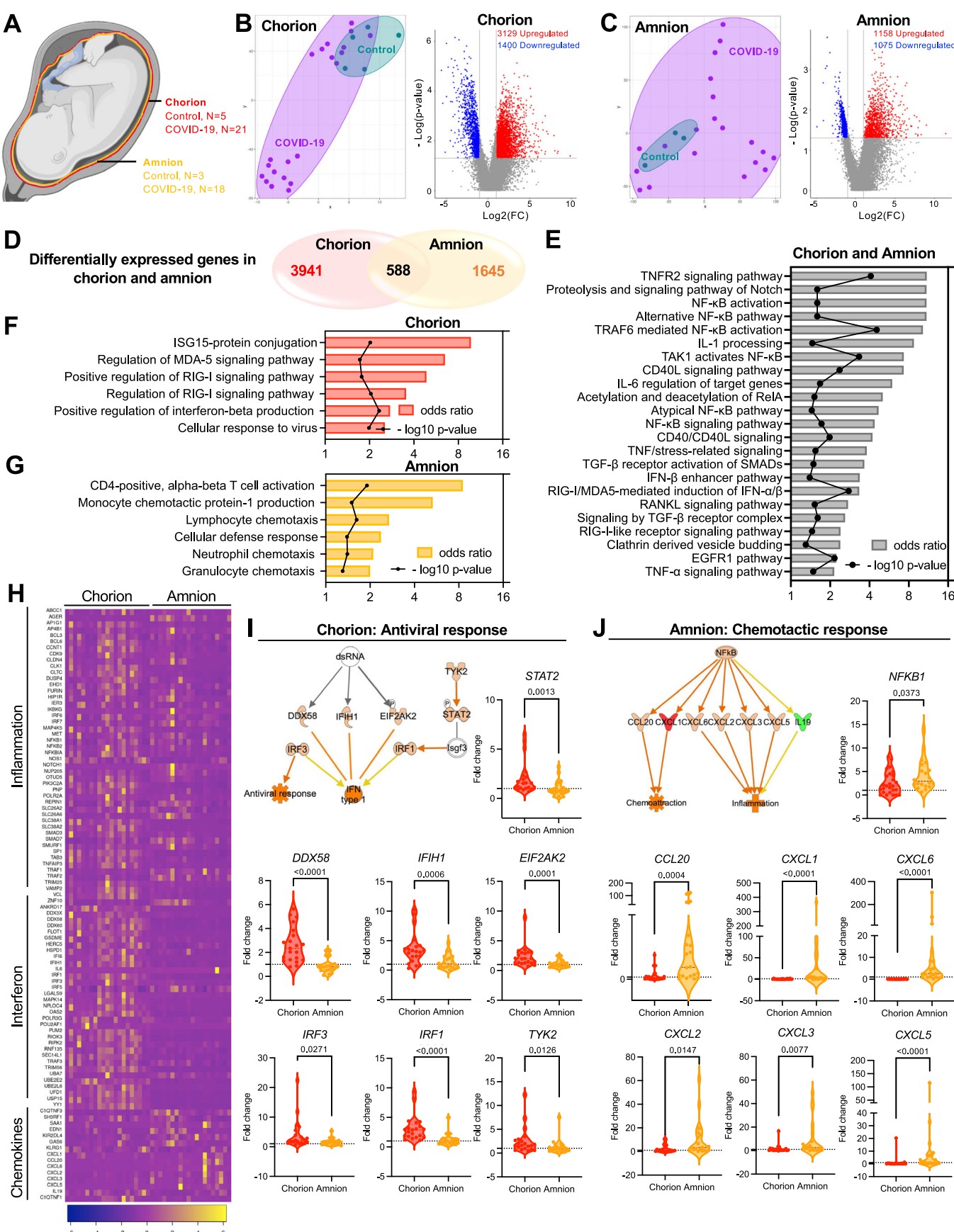

◄ **Figure 2.  Gestational exposure to SARS-CoV-2 leads to a differential immune response at the placental barrier.**

(A) Graphical representation of tissues analyzed for global transcriptomics: chorion (Control, $n = 5$; COVID-19 $n = 21$) and amnion (Control, $n = 3$; COVID-19 $n = 18$). (B) tSNE and Volcano plot of global transcriptomics of COVID-19 chorion relative to controls. (C) tSNE and Volcano plot of global transcriptomics of amnion obtained from COVID-19-affected pregnancies relative to controls. (D) Comparative analysis of DEGs ($-2 < FC > 2$, $p < 0.05$) in chorion and amnion from COVID-19-affected pregnancies. For (B–D), significantly differentially expressed genes between the groups analyzed were identified in Partek Flow by fold-change $\geq|2|$ and FDR-adjusted p-value $< 0.05$ using the Gene set Analysis (GSA) method. (E–G) Bar plot representing gene ontology of upregulated biological pathways in (E) both COVID-19+ amnion and chorion, (F) only in amnion, and (G) only in chorion. Odds ratio and p-values were calculated using Enrichr gene set enrichment analysis. (H) Heatmap illustrating the expression of previously detected genes related to inflammation, interferon, and chemokines in chorion and amnion from COVID-19 affected pregnancies. Fold change was calculated using raw transcript counts minus the mean and divided by the standard deviation of depicted transcripts in the chorion and amnion tissues. (I, J) Network analysis of DEGs exclusive to (I) COVID-19 chorion ($n = 21$) and (J) COVID-19 amnion ($n = 20$). Violin plots representing the fold change values for individual genes related to (I) antiviral response and (J) chemotactic response in chorion and amnion. All data relative to controls. Fold change calculated by individual expression divided by average expression of controls. Data are presented as means ± SEMs, using Mann–Whitney U test ($p < 0.05$). Source data are available online for this figure.

respectively (Fig. 2H). We further confirmed the findings from the enrichment pathways by validating the gene expression of the antiviral response and the chemotactic response in both chorion and amnion tissues (Fig. 2I,J). The expressions of antiviral IFN-related genes, including signal transducer and activator of transcription (*STAT*)2, DEAD box protein 58 (*DDX58* or *RIG-I*), interferon induced with helicase C domain 1 (*IFIH1*), eukaryotic translation initiation factor 2 alpha kinase 2 (*EIF2AK2*), IFN regulatory factor (*IRF*)3, *IRF1*, and tyrosine kinase 2 (*TYK2*), were significantly higher in chorion compared to amnion in COVID-19-affected pregnancies (Fig. 2I). In contrast, genes related to nuclear factor kappa B (NF-kB) downstream chemotactic and inflammatory responses, such as nuclear factor kappa B subunit 1 (*NFKB*), C-C motif chemokine ligand 20 (*CCL20*), C-X-C motif chemokine ligand 1 (*CXCL1*), *CXCL6*, *CXCL2*, *CXCL3*, and *CXCL5*, presented significantly higher expressions in amnion than chorion during COVID-19-affected pregnancies (Fig. 2J). These findings characterized distinct inflammatory transcriptomic landscapes on both the maternal and fetal sides.

## Prenatal SARS-CoV-2 infection induces overt complement activation and inflammation on the fetal side

To further explore the inflammatory immune landscape within the fetal compartment in COVID-19-affected pregnancies, we performed targeted bead-based multiplexing technology to access the proteomics profiling of 97 secreted inflammatory cytokines and chemokines using amniotic fluid (Control $n = 3$–7, COVID-19 $n = 8$) and umbilical cord plasma (Control, $n = 5$; COVID-19, $n = 18$) specimens (Fig. 3A). Strikingly, among the induction of several inflammatory cytokines/chemokines, amniotic fluid specimens derived from COVID-19-affected pregnancies exhibited a significant increase in complement proteins compared to controls (Fig. 3B). Specifically, the heightened production of C1q and C4 suggested the involvement of the classic complement pathway, followed by the downstream complement cascade activation as evidenced by increased levels of C3, C3b, C5, and C5a compared to controls (Fig. 3C,D). The activation of the complement cascade can lead to the subsequent formation of the membrane attack complex (MAC) and inflammation (Fig. 3C). In fact, when compared to controls, amniotic fluid from COVID-19-affected pregnancies demonstrated elevated levels of several pro-inflammatory cytokines, including IL-12 p40, CCL4, CCL7, CCL13, TNF, IL-1a, IL-17A, and IL-17E (Fig. 3E).

Next, we assessed the proteomic profiles of cord plasma from COVID-19-affected pregnancies. An upregulation of complement proteins C1q, C2, and C4 from the classical complement pathway was shown, along with downstream C5 and complement factor H (CFH) (Fig. 3F). In addition, there was increased expression of pro-inflammatory matrix metalloproteinase (MMP)7 and sCD40L (Fig. 3G). These findings were further supported by the transcriptomic profiles of umbilical cord tissues obtained from COVID-19-affected pregnancies. Umbilical cord tissues from COVID-19-affected pregnancies presented 1245 upregulated and 564 down-regulated DEGs compared to controls (Fig. EV1A). The enrichment analysis highlighted a pro-inflammatory profile in the fetal compartment, with induction of chemotaxis, T cell signaling, monocyte activation, and nitric oxide (NO) responses (Fig. EV1B). Similarly, umbilical cords from COVID-19-affected pregnancies showed higher expressions of *CFB* and *C4A* compared to controls (Fig. EV1C). In addition, C5 was identified as an upstream regulator of the COVID-19 umbilical cord transcriptomics profile, indicating the upregulation of a panel of downstream C5-induced inflammatory mediators (Fig. EV1D,E).

To identify if there are any overlapping biomarkers affected across the different specimen types at the maternal-fetal interface, we further evaluated the overlapping biomarkers that were significantly altered in placental chorion/amnion tissues, amniotic fluids and cord plasma. While no common biomarker was identified to be affected in all four specimen types (Fig. EV2A,B), we observed that complement factors—C1q, C4, and C5 were commonly induced in in cord plasma and amniotic fluid specimens derived from COVID-19 pregnancies (Fig. EV2A). Specifically, we compared the pathways that were predicted to be upregulated in amniotic fluid and cord plasma, identifying only five upregulated pathways commonly affected in both amniotic fluid and cord plasma, while 17 other pathways exclusively altered in amniotic fluid and one pathway exclusively altered in the cord plasma (Fig. EV2C). Among the five pathways that were commonly upregulated in both amniotic fluid and cord plasma from COVID-19-affected pregnancies, four were related to opsonization and complement activation (Fig. EV2D). Indeed, apart from the 5 common pathways, there are several other upregulated pathways observed in both amniotic fluid and cord plasma specimens from COVID-19-affected pregnancies are associated with inflammation and immune cell recruitment (Fig. EV2E,F). In summary, the proteomics and transcriptomics profiling of fetal-derived amniotic fluid and umbilical cord specimens clearly demonstrate overt complement activation in the fetal compartment during COVID-19-affected pregnancies, although the exact mechanism remains unknown.

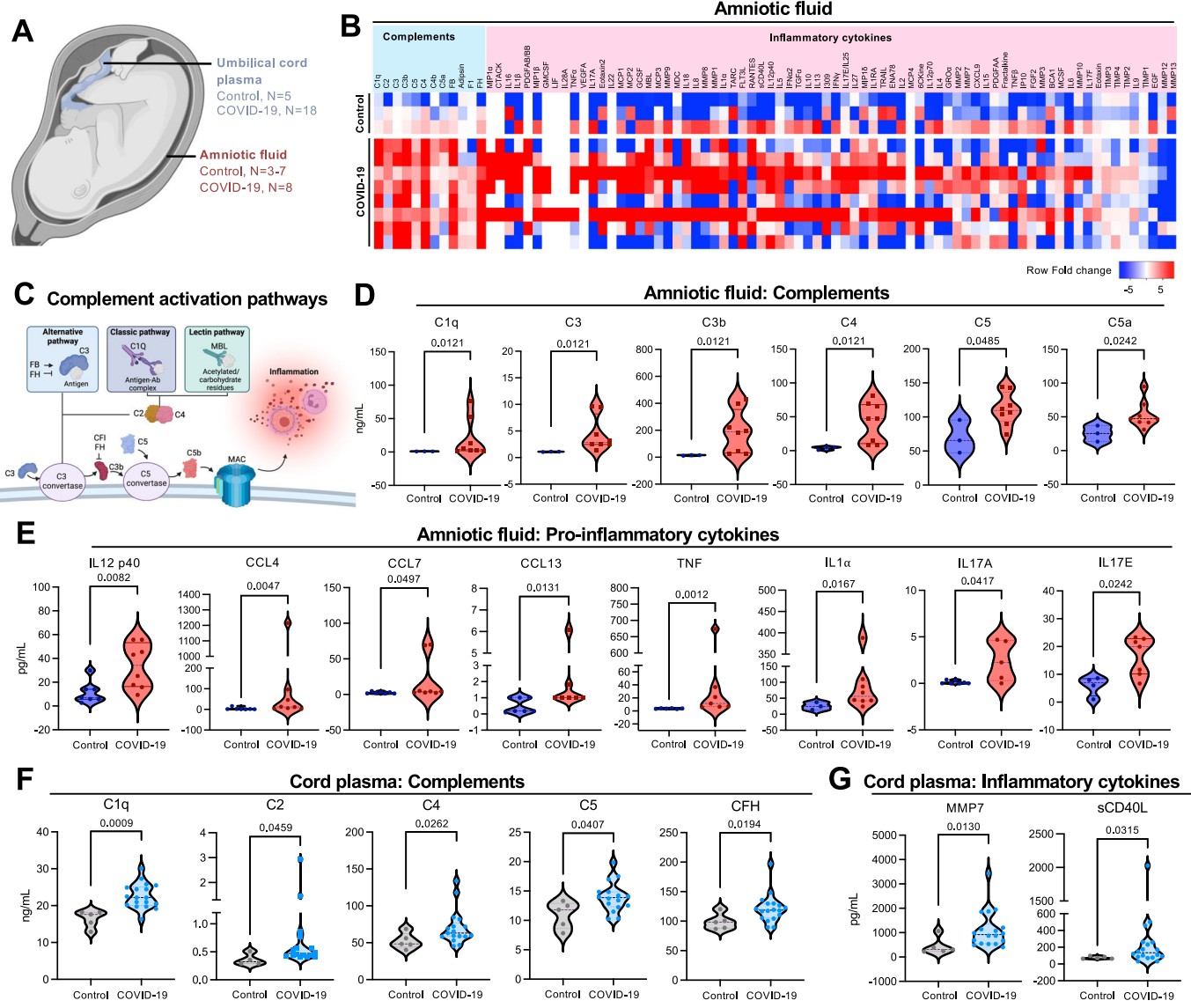

**Figure 3. Complement activation and inflammation in the fetal compartment of COVID-19-affected pregnancies.**

(A) Graphical representation of tissues analyzed for proteomics: umbilical cord plasma (Control, $n = 5$; COVID-19 = 18), and amniotic fluid (Control, $n = 3$–7; COVID-19 $n = 8$). (B) Heatmap illustrating the levels of proteins related to complement and inflammatory cytokines in the control and COVID-19 groups quantified in amniotic fluid. (C) Graphical representation of complement activation by non-self-antigens through alternative, classic, and lectin pathways and the role of proteins analyzed. (D–G) Violin plots representing levels of individual proteins related to complement and pro-inflammatory cytokines in amniotic fluid (D and E, respectively) and cord plasma (F and G, respectively) for control ($n = 3$) and COVID-19 ($n = 8$) groups. Data are presented as means ± SEMs pg/mL, using Mann–Whitney U test ($p < 0.05$). Source data are available online for this figure.

## SARS-CoV-2 ORF8 crosses the placental barrier

Next, we hypothesized that the secreted viral cytokine SARS-CoV-2 ORF8 might be responsible for triggering overt complement activation and inflammation observed in the fetal compartment during COVID-19-affected pregnancies. We examined maternal plasma (Control, $n = 4$; COVID-19, $n = 23$), umbilical cord plasma (Control, $n = 4$; COVID-19, $n = 20$), and amniotic fluid (Control, $n = 3$; COVID-19, $n = 8$) specimens derived from controls and COVID-19-affected pregnancies to detect the presence of SARS-CoV-2 ORF8 protein in all sample types examined. All the controls samples were negative to ORF8 (Fig. 4A).

Interestingly, in COVID-19-affected pregnancies, the highest levels of ORF8 were found in amniotic fluid (mean 1312 ng/mL), while similar levels were observed in maternal plasma (mean 143.8 ng/mL) and cord plasma (mean 126.8 ng/mL) (Fig. 4A). We detected circulating ORF8 protein in 65.22% ($n = 15/23$) of maternal plasma specimens, 60% ($n = 12/20$) in umbilical cord plasma and 62.5% ($n = 5/8$) circulating ORF8 protein in amniotic fluids (Fig. 4B). Among the ORF8 positive amniotic fluids, 60% ($n = 3/5$) were consistently positive in other fluids analyzed. Interestingly, in the absence of active SARS-CoV-2 infection, we were able to detect ORF8 in maternal plasma up to 216 days after

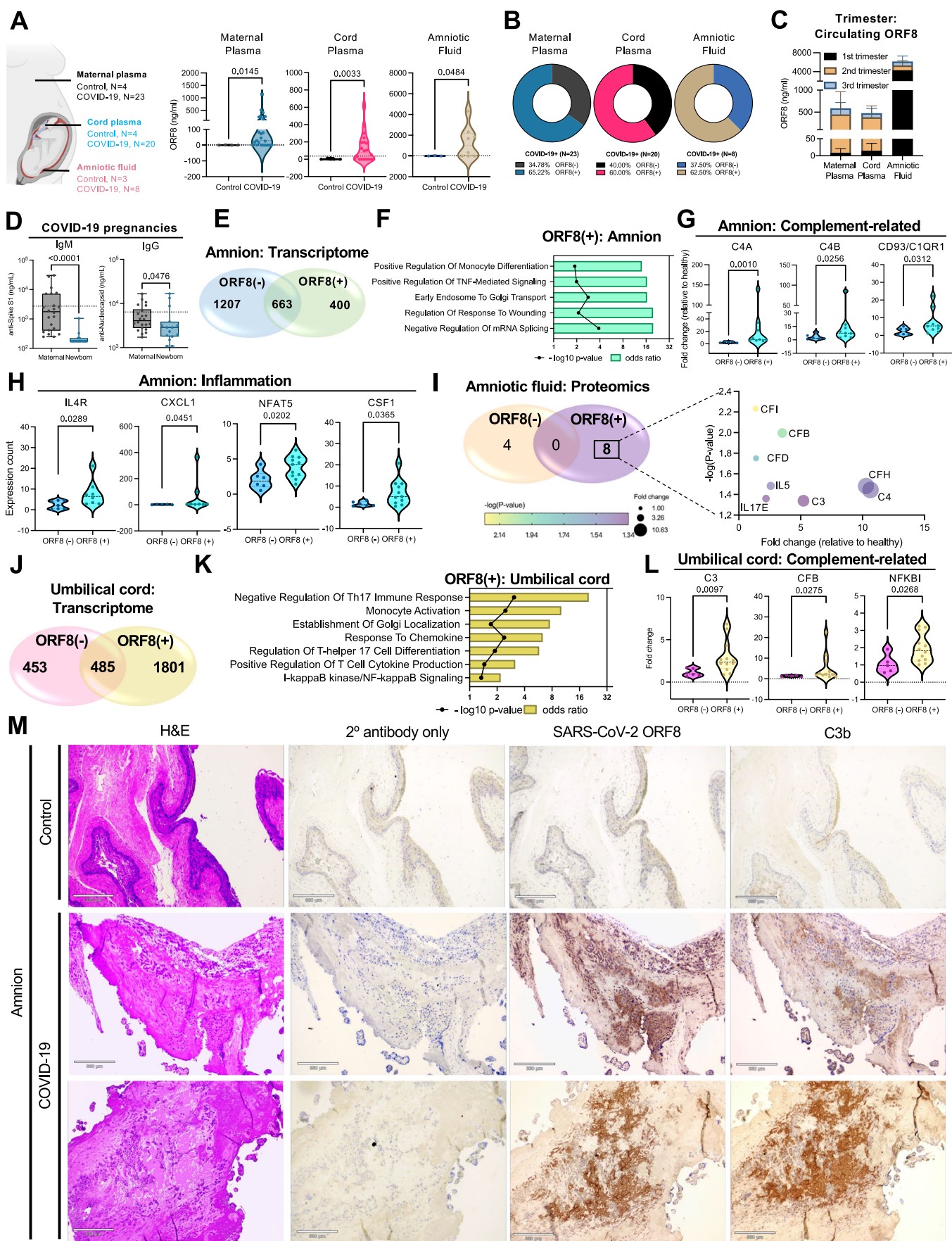

**Figure 4. Transplacental ORF8 is associated with augmented complement and inflammation in the fetal compartment of COVID-19-affected pregnancies.**

(A) Graphical representation of samples analyzed for ORF8 quantitative and qualitative ELISA: maternal plasma (Control, $n = 4$; COVID-19, $n = 23$), umbilical cord plasma (Control, $n = 4$; COVID-19, $n = 20$), and amniotic fluid (Control, $n = 3$; COVID-19, $n = 8$). ORF8 levels in samples from control and COVID-19-affected pregnancies. Data are presented in violin plots showing the median (middle line) and describe numerical data distributions using density curves. Values are represented in ng/mL with cut off 1.7 ng/mL in maternal plasma, cut off of 13.4 ng/mL in umbilical cord plasma and cut off of 36.1 ng/mL in amniotic fluid. (B) Pie charts illustrating qualitative analysis of ORF8 ELISA. (C) Stacked bars representing the circulating ORF8 according to trimester of SARS-CoV-2 infection (First trimester, $n = 2$; Second trimester, $n = 4$; Third trimester, $n = 10$). (D) Boxplots representing levels of anti-Spike S1 IgM with cut off of 2720 ng/mL and anti-Nucleocapsid IgG with cut off of 6420 ng/mL in maternal plasma ($n = 22$) and newborn cord plasma ($n = 15$) from COVID-19-affected pregnancies. Data are presented in boxplot, middle line represents means, the bound of box represent interquartile range, and whiskers represent maximum and minimum values. (E) Comparative analysis of DEGs ($-2 < FC > 2$, $p < 0.05$) in COVID-19 amnion ORF8 negatives and positives. (F) Bar plot representing gene ontology of upregulated biological pathways in COVID-19 amnion ORF8 positive group. (G, H) Violin plots representing the fold change values for individual genes related with (G) complement (ORF8 ($-$), $n = 5$–6; ORF8 ($+$), $n = 9$–10) and (H) inflammation in amnion (ORF8 ($-$), $n = 5$–7; ORF8 ($+$), $n = 8$–11) transcriptomics. (I) Comparative analysis of differentially expressed proteins (DEPs) ($p < 0.05$) in COVID-19 amniotic fluid ORF8 negatives ($n = 4$) and positives ($n = 8$). Bubble plot of DEPs upregulated exclusively in COVID-19 amniotic fluid ORF8 positive group. (J) Comparative analysis of DEGs in COVID-19 umbilical cord ORF8 negatives and positives. (K) Bar plot of gene ontology biological pathways associated with DEGs upregulated exclusively in COVID-19 umbilical cord ORF8 positive. (L) Violin plots representing the fold change values for individual genes related to complement in umbilical cord transcriptomics. (M) Amnion serial sections were analyzed by immunohistochemistry for detection of SARS-CoV-2 ORF8 and C3b. From left to right, tissues stained with (i) Hematoxylin and Eosin (H&E) (hematoxylin—purple, eosin—pink); (ii) Hematoxylin (purple) and the secondary antibody anti-rabbit (Motulsky and Brown, 2006); (iii) Hematoxylin (purple) and anti-ORF8 produced (Motulsky and Brown, 2006); and (iv) Hematoxylin (purple) and anti-C3b (Motulsky and Brown, 2006). Images were taken at 4X magnification. Representative images from three control and three COVID-19 amnions. Fold change calculated by individual expression divided by average expression of control. Violin plots data are presented as means ± SEMs pg/mL, using Mann–Whitney U test ($p < 0.05$). Comparative analysis relative to controls. Source data are available online for this figure.

initial infection, with higher levels in the second trimester. ORF8 was detected in the infant samples up to 188 days after the initial infection with higher levels of ORF8 in cord plasma when the infection occurred in the second trimester and higher levels of ORF8 in amniotic fluid when the infection occurred in the first trimester (Fig. 4C, Table EV2). To confirm that the presence of ORF8 detected is not due to an active SARS-CoV-2 infection on the fetal side, we quantified the IgM anti-SARS-CoV-2 Spike S1 and IgG levels of anti-SARS-CoV-2 Nucleocapsid in maternal and cord/newborn plasma. During pregnancy, maternal IgG can be passively transferred across placenta but not IgM. Hence, a positive detection of SARS-CoV-2 IgM in the cord/newborn plasma would be indicative of an active SARS-CoV-2 infection. Regarding IgM anti-Spike 1, our results show that all newborns were negative (0/15), and in 31.8% ($n = 7/22$) of maternal plasma specimens, suggesting no vertical transmission of SARS-CoV-2. Furthermore, we were able to detect circulating IgG anti-Nucleocapsid in 20% ($n = 3/15$) of umbilical cord plasma and 27.3% ($n = 6/22$) of maternal plasma (Fig. 4D). All this data together provides evidence of transplacental transfer of ORF8 and IgG anti-SARS-CoV-2 from the maternal to the fetal compartment.

## Transplacental ORF8 is associated with overt complement activation and inflammation on the fetal side

To assess the impact of transplacental ORF8 on the immunological landscape of the fetal compartment in COVID-19-affected pregnancies, we compared the transcriptomic profiles of amnion tissues between the ORF8 positive ($+$) and ORF8 negative ($-$) COVID-19-affected pregnancies. Analysis of amnion transcriptomics identified 400 DEGs exclusively affected in ORF8 ($+$) amnions (Fig. 4E). Enrichment analysis of this subset of DEGs revealed induction of biological pathways associated with monocyte differentiation and TNF response (Fig. 4F). Furthermore, we observed significantly higher expressions of complement genes *C4A*, *C4B*, and complement component 1 Q subcomponent receptor 1 (*C1QR1*), and inflammatory genes such as *IL-4R*, *CXCL1*, nuclear factor of activated T cells 5

(*NFAT5*), and colony-stimulating factor 1 (*CSF1*) in ORF8($+$) amnion tissues when compared with ORF8(-) COVID-19 amnion tissues (Fig. 4G,H). Similarly, we evaluated the proteomic profiles between ORF8($+$) and ORF8(-) amniotic fluids from COVID-19-affected pregnancies. We identified eight secreted proteins that were exclusively affected in the ORF8($+$) group, notably showing elevated levels of C3, C4, IL-5, and IL-17E (Fig. 4I). Analysis of the umbilical cord transcriptomics also revealed 1801 DEGs exclusively affected in COVID-19 ORF8($+$) situations, associated with IL-17 and NF-kB responses, and monocyte activation (Fig. 4J,K). Expressions of *C3*, *CFB*, and *NFKBI* were also found to be higher in ORF8($+$) umbilical cord tissues compared to the ORF8(-) group (Fig. 4L).

To confirm the localization of transplacental ORF8 and the presence of complement activation, we performed immunohistochemistry (IHC) using serial-sectioned amnion and chorion tissues obtained from control and COVID-19-affected pregnancies. The IHC staining demonstrated widespread staining of both ORF8 protein and C3b, with apparent co-localization between both proteins (Figs. 4M and EV3). The co-localization was further confirmed by immunofluorescence (IF) in both tissues (Fig. 5). We evaluated the colocalization of ORF 8 and C3b channels using Mander's coefficient and Pearson's correlation coefficient which demonstrated similar intensity and a positive correlation between ORF8 and C3b in both chorion (Fig. EV4A) and amnion (Fig. EV4B) tissues derived from COVID-19-affected pregnancies. Furthermore, to demonstrate ORF8 distribution in the placental trophoblasts, we have performed immunofluorescence staining on the placental chorion and amnion tissues using antibodies against ORF8 and trophoblast marker—cytokeratin 8/18 (Krt8/18), which is found on syncytiotrophoblasts, villous trophoblasts in the fetus side (amnion) and extravillous trophoblasts in the maternal side (chorion). Specifically, our results demonstrated the presence of ORF8 within the extravillous trophoblasts in maternal chorion tissue (Fig. 5A). In the fetal amnion tissue, ORF8 is located at close proximity to syncytiotrophoblasts and villous trophoblasts, but not within (Fig. 5B). These findings suggest a potential role of ORF8 in directly activating the complement cascade during COVID-19-affected pregnancies.

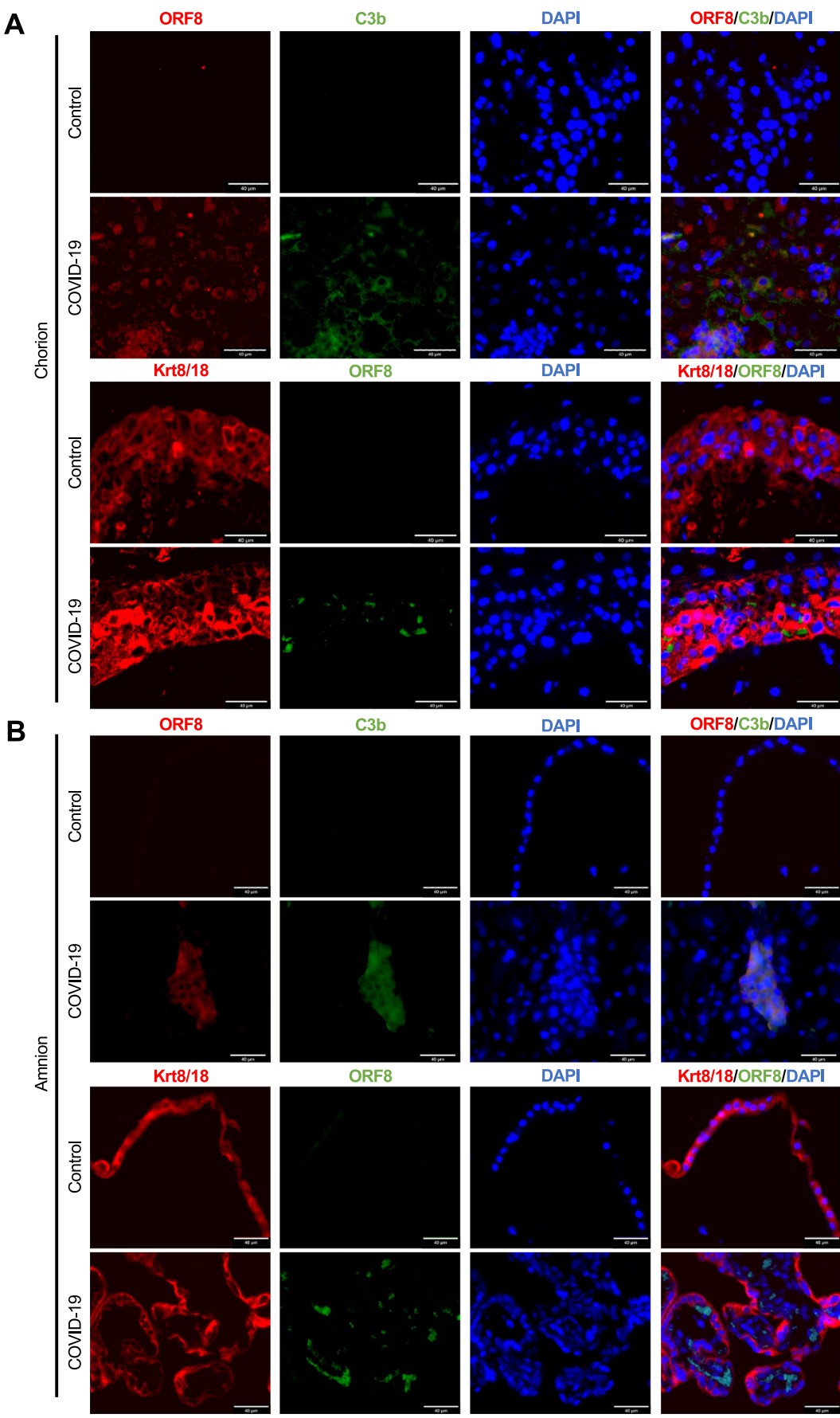

**Figure 5.  Co-localization of ORF8 and C3b in placental tissues derived from COVID-19 pregnancies.**

(**A**) Chorion and (**B**) amnion serial sections were stained with SARS-CoV-2 ORF8 (red) and C3b (green); or with Krt8/18 (red) and ORF8 (green). Images were taken at 40× magnification. Images are representative of one control and one COVID-19+. Scale bars: 40 μm. Source data are available online for this figure.

## SARS-CoV-2 ORF8 consistently triggers complement activation in trophoblasts in vitro

To investigate the potential role of ORF8 in activating the complement cascade at the maternal-fetal interphase during COVID-19-affected pregnancies, we conducted in vitro SARS-CoV-2 ORF8 treatment experiments on HTR8/SVneo immortalized placental villous trophoblasts, primary human villous trophoblasts, and iPSC-derived trophoblasts (Fig. 6A). First, HTR8/SVneo cells were treated with different concentrations (10, 20, and 100 ng/mL) of purified SARS-CoV-2 ORF8 protein for 8 and 16 h. Interestingly, data shows an inverted dose-response curve from 20 ng/mL to 100 ng/mL of ORF8 (Fig. 6B). It has been identified that ORF8 can form stable trimeric and tetrameric oligomers (Assadizadeh and Azimzadeh Irani, 2023). Hence, one explanation for the inverted dose-response seen with gene expression after ORF8 treatment is that the high concentration of 100 ng/mL promotes the formation of these alternative oligomers preventing their binding to C1Q. However, the stimulation with ORF8 led to increased expressions in complement genes, including *C1QA, C1QB, C1QC, CFH, IL-1A, IL-17A,* and *IL-17E* in one or both of the evaluated time points (Fig. 6B). Notably, HTR8/SVneo trophoblasts, treated with 20 ng/mL of ORF8 presented the highest magnitude of complement genes expressions at 8 h post-treatment (Fig. 6B,C).

Other cellular models of trophoblasts also responded to the treatment with 20 ng/mL ORF8 for 8 h. Primary human villous trophoblasts presented significantly higher levels of *C1QA* and *C1QC* after treatment (Fig. 6D). Furthermore, trophoblasts differentiated from human immortalized pluripotent stem cells (iPSCs), with a mixed cytotrophoblast (CTB)/syncytiotrophoblast (STB)/extravillous trophoblast (EVT) phenotype (Fig. EV5), showed significantly high expression of *C1QB*, and a suggestive increase in *C1QA* and *C1QC* expressions as well (Fig. 6E). Altogether, these results demonstrate that SARS-CoV-2 ORF8 was consistently capable of triggering expression of complement-related transcripts in the placental cell models tested.

To confirm the activation of complement cascade in these trophoblast cells, we performed surface staining of C3b after 30 min, 8 h, and 16 h of ORF8 (20 ng/mL) treatment (Fig. 6F). We observed a growing trend of C3b activation at 30 min (~2%), 8 h (~4%) and 16 h (~8%), indicative of ORF8-mediated complement activation in these cells (Fig. 6F). In addition, increased C3 and C5 proteins were detected in the supernatants of ORF8-treated cells compared to mock controls (Fig. 6G). Typically, classical complement activation is initiated by the binding of specific immunoglobulin epitopes to the globular domain of C1q (Duncan and Winter, 1988). However, our data suggested that ORF8, an immunoglobulin-like protein, can directly mediate complement activation without immunoglobulins.

## SARS-CoV-2 ORF8 binds to the functional domain of C1q subcomponent A

In the face of the clinical and in vitro evidence of ORF8 triggering classical complement activation, we further investigated the possible interaction between ORF8 and C1q, the central player in classical complement activation. In an in vitro ELISA-based binding assay using four different concentrations of purified ORF8 (5, 10, 50, and 100 nM) and human C1q (5, 10, 25, 50 nM), we observed a dose-dependent binding interaction between ORF8 and C1q, with a binding saturation occurring at 25 nM of C1q (Fig. 7A). C1q is a complex molecule composed of 18 polypeptide chains, including six A chains, six B chains and six C chains, each containing an N-terminal disordered collagen-like domain and a C-terminal globular domain region (Kishore and Reid, 2000). Next, to identify the specific C1q subcomponent (A, B, C) interacting with ORF8, we performed co-transfection experiments in HRT8/SVneo and HEK-293T cells using plasmids expressing ORF8 and full-length C1q subcomponents A, B or C. Co-immunoprecipitation using anti-ORF8 antibody and western blot analysis revealed that ORF8 specifically binds to C1qA but not C1qB or C1qC (Figs. 7B and EV6A). To further pinpoint the specific domains of C1qA responsible for binding to ORF8, we co-expressed plasmids containing the ORF8-HA tag and full-length C1qA-3xFLAG, C1qA-globular-3xFLAG, or C1qA-disordered domain-3xFLAG in HTR8/SVneo and HEK-293T cells. After co-immunoprecipitation with an anti-ORF8 antibody or anti-FLAG tag, we confirmed that the interaction between ORF8 and C1qA is mediated through the globular functional domain (Figs. 7C and EV6B). Furthermore, we performed immunoprecipitation coupled with mass spectrometry to confirm the interaction between ORF8 and components of the complement system. Interestingly, we found complement C3, a downstream product of C1q signaling pathway, to be one of the binding partners of ORF8. These findings collectively demonstrated that ORF8 can induce classical complement activation through its specific interaction with the globular domain of C1q subcomponent A in trophoblast cells.

## SARS-CoV-2 ORF8 amino acids C37-A51 binds to C1q and triggers complement activation

Lastly, we aimed to identify the specific binding epitope on ORF8 essential for its interaction with C1q and subsequent complement activation. To achieve this, we designed a peptide library covering the entire full-length ORF8 protein sequence. This library consisted of 28 peptides, each 15 amino acids (aa) long, with an 11 amino acid overlap. At two concentrations (0.1 and 1 nM), these peptides were subjected to in vitro binding with 10, 20, or 50 nM of purified human C1q protein. The screening of the ORF8 peptide library identified four potential interaction sites, namely peptides (i) #4 (S1-H15), (ii), #10 (C37-A51), (iii) #19 (Y73-T87), and (iv) #25 (S97-Y111), which displayed dose-dependent interaction with C1q protein (Fig. 8A).

With access to the crystal structure for ORF8 and C1qA-globular domains, we performed in silico modeling for protein–protein docking simulations using HDOCK. Among the 10 ORF8-C1q docking models, we selected docking model 1 with

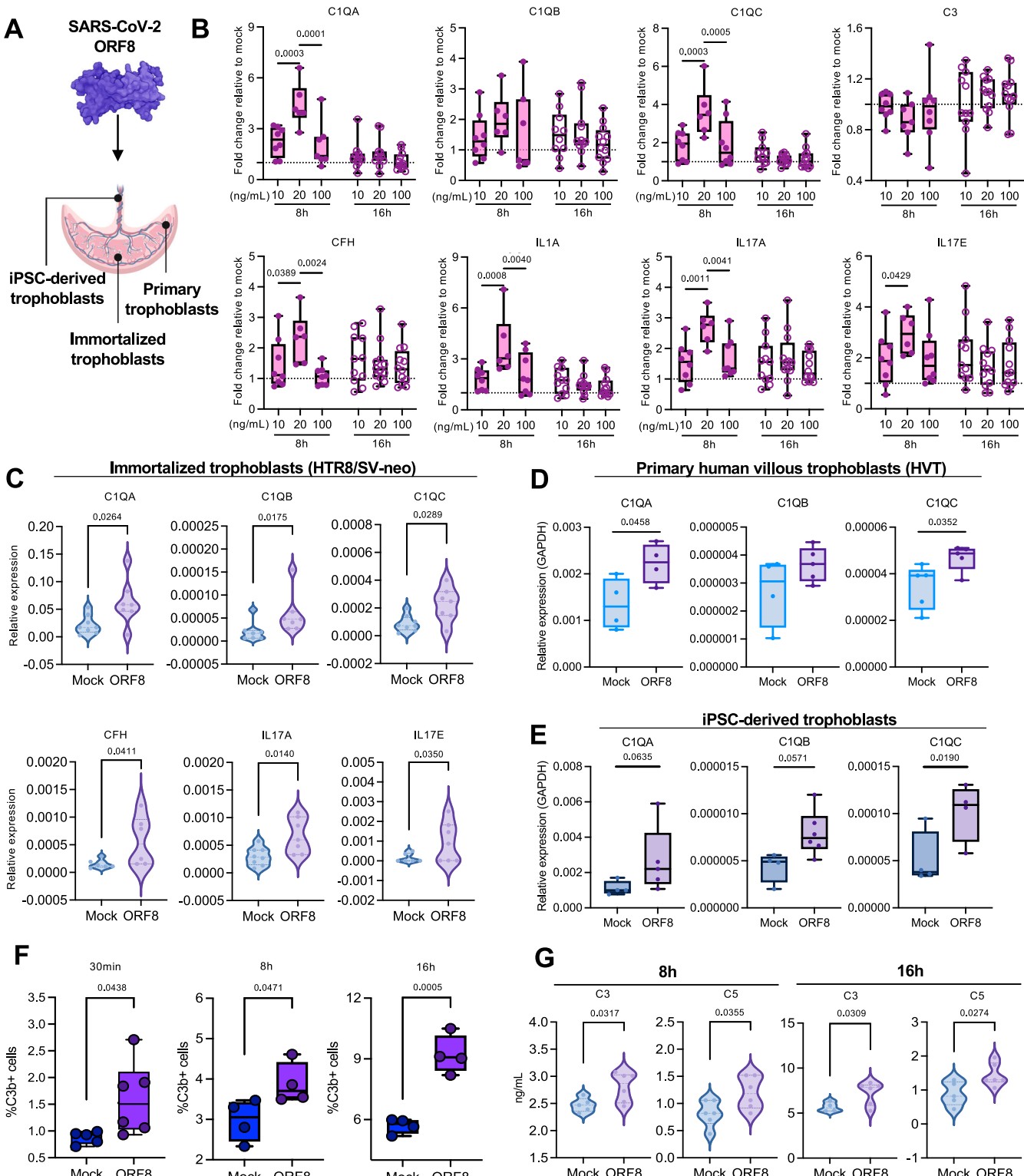

the lowest docking score (-258.39) and highest confidence score (0.8973) for structure visualization. This model showed the location of the four ORF8 peptides (#4 in red, #10 in pink, #19 in blue, and #25 in yellow) (Fig. 8B). By aligning the ORF8 aa sequence with the predicted ORF8 interface residues across the 10 docking models, we observed two groups of models. One group (7 out of 10 models)

involved predicted interacting aa residues present on ORF8 peptides #4, #10, and #25. In contrast, the other group (3 out of 10 models) involved only peptides #4 and #19 (Fig. 8C). Hence, based on docking model 1, ORF8 peptides #4, #10, and #25 are located in the closest proximity to the C1qA-globular domain, suggesting potential binding sites (Fig. 8B).

**Figure 6.  SARS-CoV-2 ORF8 treatment on placental trophoblasts triggers complement activation and inflammation.**

(A) Graphical representation detailing the placental cell culture models for in vitro SARS-CoV-2 ORF8 treatment. (B) Boxplots representing the fold change values of gene expression for individual genes, obtained from standardization of in vitro SARS-CoV-2 ORF8 treatment with 10 ng/mL, 20 ng/mL, or 100 ng/mL for 8 h from eight independent experiments ($n = 8$) and 16 h from eleven independent experiments ($n = 11$) on immortalized trophoblasts (HTR8/SVneo cells). Fold change calculated by DDCt method, relative to mock. Data are presented in boxplot where the middle line represent means, the bound of box represent interquartile range, and whiskers represent maximum and minimum values in ng/mL, using one-way ANOVA Kruskal–Wallis with uncorrected Dunn's test. (C–E) Relative expression of genes associated with complement activation in the placental cell culture models treated with 20 ng/mL of SARS-CoV-2 ORF8 for 8 h. The models included (C) HTR8/SVneo cells (mock, $n = 6$–8; and ORF8, $n = 7$–8) (D) primary human villous trophoblasts (HVT) (mock, $n = 4$–5; and ORF8, $n = 4$–5), and (E) iPSC-derived trophoblasts (mock, $n = 5$–6; and ORF8, $n = 6$). Relative expression was calculated by DCt method using *GAPDH* as the normalizing gene. Mann–Whitney U test was used for analysis. (F) Boxplots representing the percentage of C3b + HTR8/SVneo cells treated with SARS-CoV-2 ORF8 for 30 min (mock, $n = 5$; and ORF8, $n = 6$), 8 h (mock, $n = 4$; and ORF8, $n = 4$) or 16 h (mock, $n = 4$; and ORF8, $n = 4$). Data are presented in boxplot, middle line represents means, the bound of box represent interquartile range, and whiskers represent maximum and minimum values. (G) Violin plots representing the levels of complement proteins in the supernatant of HTR8/SVneo cells treated with SARS-CoV-2 ORF8 for 8 h (mock, $n = 5$–6; and ORF8, $n = 6$) or 16 h (mock, $n = 4$–5; and ORF8, $n = 6$). Mann–Whitney U test was used for analysis ($p < 0.05$). Immortalized pluripotent stem cells (iPSC). Source data are available online for this figure.

## Verification of SARS-CoV-2 ORF8-C1qA docking using 2D interaction models

We further validated the potential binding between the linear epitopes of ORF8 peptides #4, #10, #19, and #25 and the globular domain of C1qA, as well as to identify the specific residues involved in the stable complex formation. The top-docking poses of C1qA and ORF8 peptides are represented in 2D interaction models (Fig. 8D). We observed that several residues from peptide #4 (Gln18 and Gln19), peptide #10 (Pro36 and His40), and peptide #25 (Glu106) formed multiple hydrogen bonds at several interfaces of the C1qA globular head. Strong salt bridges were also observed between Gln19 and Arg100 or Arg128 at less than 1.0 Å distance, while an additional salt bridge was observed between Pro36 and Thr116 (Fig. 8D, left and middle panels). Prominent pi-pi and van der Waals interactions were observed between peptide #25 residues Phe104, Tyr105, and His112 and the hydrophobic C1qA residues (Phe212, Phe215, and Phe217). The latter C1qA residue, Phe217, was also involved with Phe41, located in peptide #10 via a similar pi-pi interaction. Interestingly, Phe104 interacted strongly with both chains of C1qA through residues Phe217 (chain A, 1.2 Å) and Phe212 (chain C, 2.1 Å), respectively (Fig. 8D, right panel). Conversely, we observed eight C1qA residues distributed in chains A and C that extensively interact with multiple sites located at peptides #4 and #25, peptides #4 and #10, or peptides #10 and #25, using various bonding interactions (Table EV3). Interestingly, several of these residues from peptides #4, #10, and #25 are also engaged with other C1qA residues in the globular domain (Table EV4), suggesting the potential synergistic binding mode of two or more ORF8 peptides to stabilize the ORF8-C1qA globular domain complex.

## SARS-CoV-2 ORF8 peptides trigger complement activation in trophoblasts in vitro

To determine which ORF8 peptide(s) could trigger complement activation in vitro, we treated HTR8/SVneo and primary trophoblasts with ORF8 peptides #4, #10, #19, or #25. The treatment with peptide #10 induced the highest expressions of *C1QA*, *C1QB*, *C1QC*, *CFH*, *IL17A*, and *IL17E* in trophoblast cells after 8 h post-treatment (Fig. 8E,F). We further confirmed the activation of complement cascade through the quantification of surface C3b deposition on HTR8/SVneo trophoblast cells, which demonstrated the highest C3b staining at 30 min after treatment with 1 nM peptide #10 (Fig. 8G). In summary, these results identified four

potential ORF8 epitopes interacting with the C1qA-globular domain, ultimately narrowing down to a 15-aa ORF8 epitope (C37-A51) sufficient to elicit complement activation in trophoblasts (Fig. 8H).

## Discussion

COVID-19 during pregnancy is associated with a higher risk of maternal disease severity and death characterized by a dysregulated inflammatory response when compared to non-pregnant individuals (Fajgenbaum and June, 2020). Previously, we identified COVID-19-induced NF-κB-dependent pro-inflammatory immune activation in mothers and immune rewiring in infants at delivery (Foo et al, 2021). In this study, we aimed to characterize the maternal-fetal crosstalk that occurs during COVID-19-affected pregnancies to better elucidate the mechanisms behind SARS-CoV-2-driven pregnancy complications and prenatal immune rewiring.

Prenatal SARS-CoV-2 infection, typically based on an infant nasopharyngeal swab test, has been associated with vertical transmission, albeit at a low incidence rate of about 2 to 3% (Ezechukwu et al, 2022). A recent meta-analysis consolidated findings from all COVID-19 pregnancy research studies that determined the occurrence of vertical transmission either through detecting SARS-CoV-2 qRT-PCR or levels of anti-SARS-CoV-2 IgM antibodies. This analysis demonstrated SARS-CoV-2 detection in 20 of 606 neonates (3.3%) (Ezechukwu et al, 2022). In this study, we aimed to accurately determine the presence of SARS-CoV-2 viral RNA (N1 and N2) in maternal and fetal biospecimens using ddPCR, a highly sensitive and precise technique for SARS-CoV-2 viral RNA detection compared to conventional real-time-PCR (Liu et al, 2020; Suo et al, 2020). Indeed, we detected a higher frequency of 26% ($n = 6/23$) of SARS-CoV-2 viral RNA in the placental fetal compartment (amnion, amniotic fluid, cord plasma, or newborn plasma) but no IgM anti-Spike 1 detection in the cord plasma within our COVID-19+ pregnant cohort. Interestingly, SARS-CoV-2 RNA was identified across all three trimesters of pregnancy, from 8 to 39 weeks of gestation. While these findings are representative of the 23 COVID-19-affected pregnancies reported in our study, the higher rate of vertical transmission identified here suggests that SARS-CoV-2 vertical transmission rates in prior studies may be highly underestimated due to differences in SARS-CoV-2 detection methods and biospecimens available for testing.

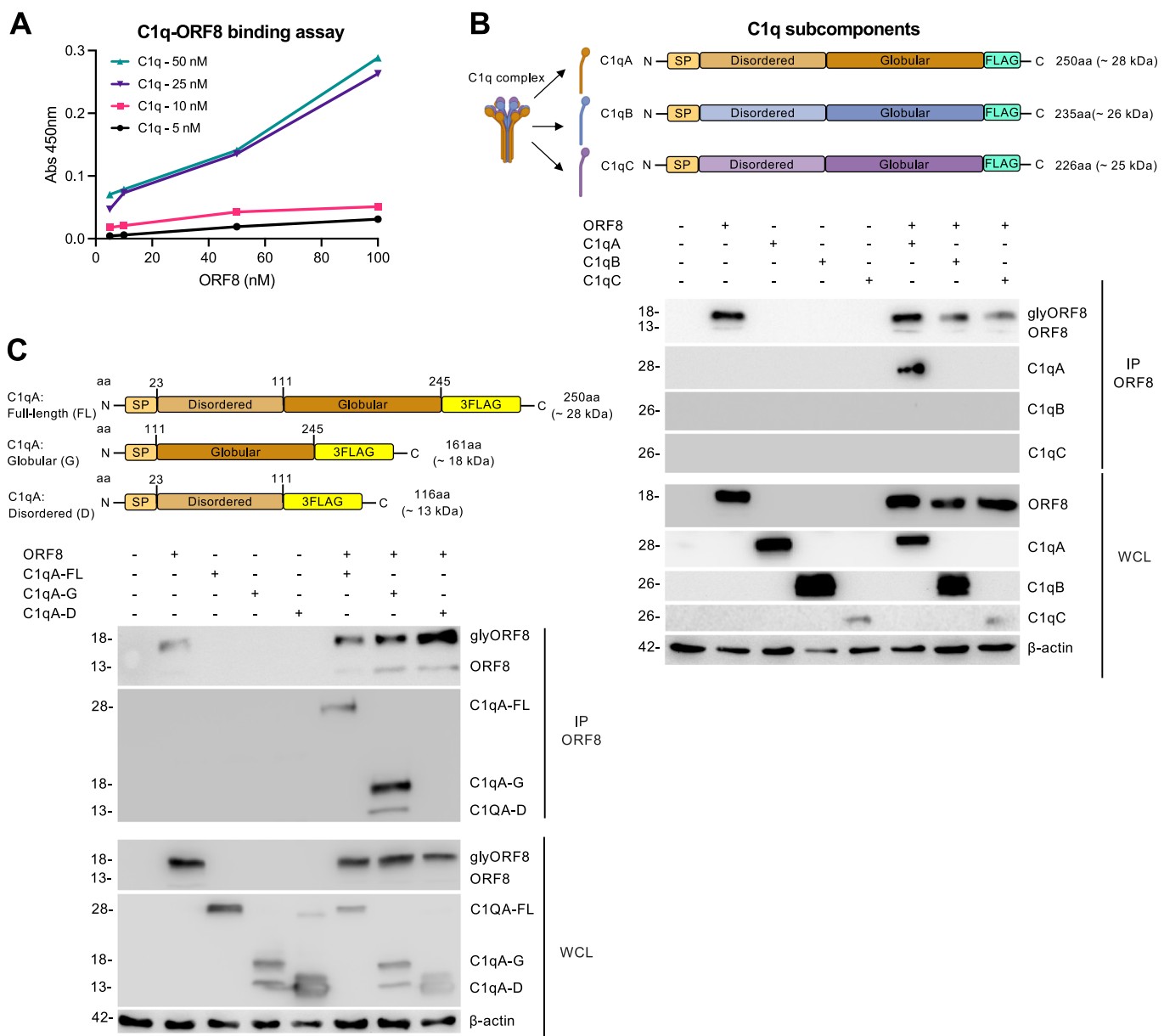

**Figure 7. SARS-CoV-2 ORF8 binds specifically to globular domain of complement C1q subcomponent subunit A in placental trophoblasts.**

(A) Line chart representing absorbance values obtained in the SARS-CoV-2 ORF8 - C1q in vitro binding assay using different molarities of C1q (5, 10, 20, and 50 nM) and SARS-CoV-2 ORF8 (0, 20, 40, 60, 80, and 100 nM). Absorbance calculated by discounting the blank (0 nM C1q). (B) Co-immunoprecipitation of SARS-CoV-2 ORF8 with C1qA, C1qB and C1qC. Graphical representation of the plasmid constructs containing the expression cassettes for the subcomponents C1qA, C1qB, and C1qC contained an N-terminal signal peptide (SP), and a C-terminal FLAG tag. Western blot images for co-immunoprecipitation using an anti-ORF8 antibody. (C) Co-immunoprecipitation of C1qA full-length (FL), globular (G), disordered domain (D) with SARS-CoV-2 ORF8. Graphical representation of the plasmid constructs containing expression cassettes for the C1qA-FL, C1qA-G, and C1qA-D, with N-terminal SP and a C-terminal 3FLAG tag, cloned in pIRES vectors. Western blot images for co-immunoprecipitation using an anti-ORF8 antibody. Co-immunoprecipitated product (IP). Whole-cell lysate (WCL). Source data are available online for this figure.

Despite relatively low rates of SARS-CoV-2 vertical transmission, immunopathological analysis of placentas from COVID-19-affected pregnancies demonstrated a high frequency of placental inflammation and structural damage upon delivery (Blasco Santana et al, 2021). Indeed, within our COVID-19 pregnancy cohort, we also observed several cases of placental inflammation, such as chorioamnionitis, chronic villitis, and deciduitis. Through transcriptomics and proteomics immune profiling of placental tissues obtained from COVID-19-affected pregnancies, we demonstrated robust inflammatory profiles in both maternal and fetal compartments. Specifically, we observed an overall upregulation of inflammatory responses, including NF-kB, IL-6, IL-1, and TNF in the maternal-skewed placental chorion tissues and fetal-skewed compartments (placental amnion tissue, amniotic fluid, umbilical cord, and cord blood). In addition, we observed unique overt complement activation (C1q, C3, C3b, C4, C5, and C5a) in the fetal

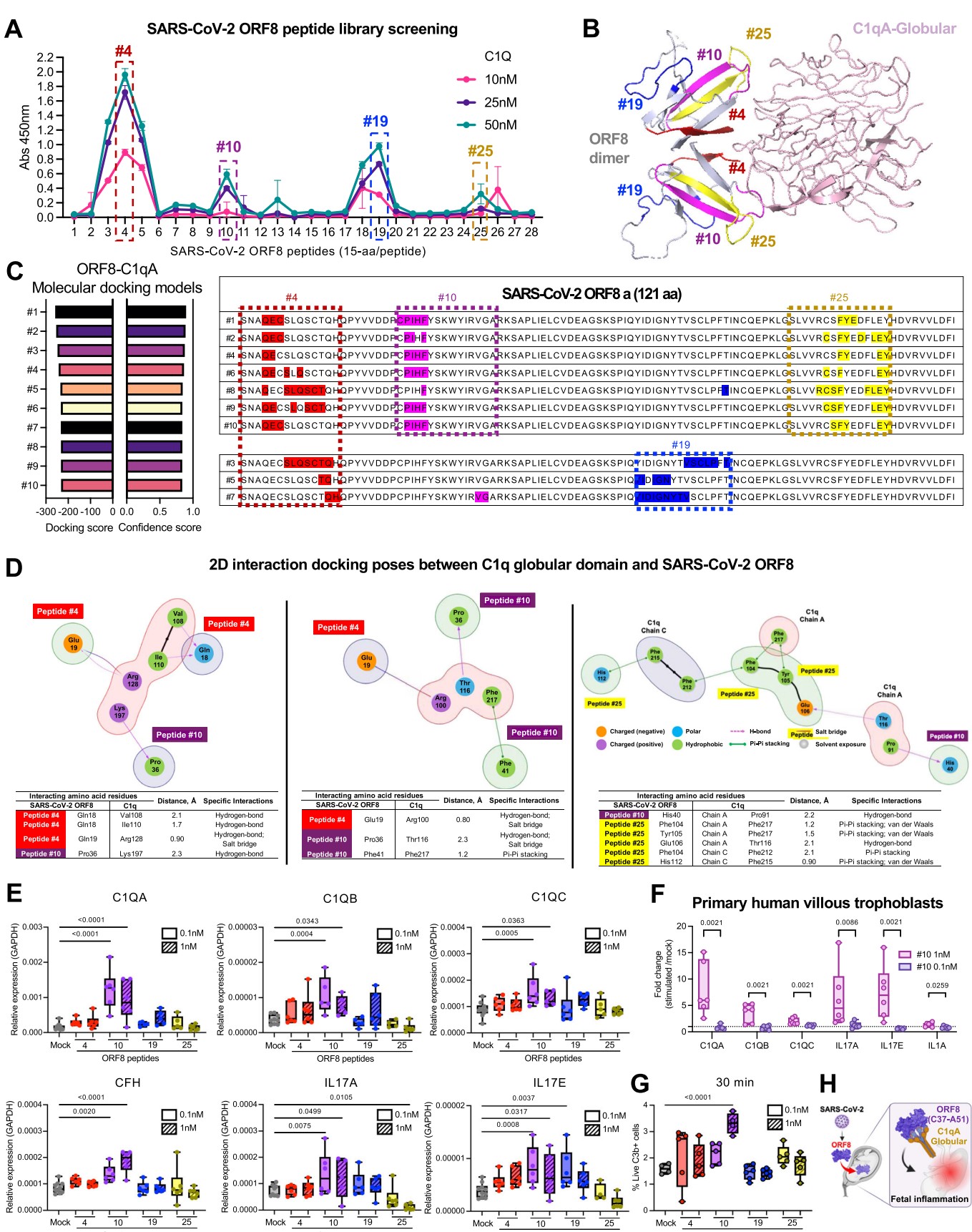

Figure 8.  Fifteen-aa residues of SARS-CoV-2 ORF8 are sufficient to bind C1q and elicit complement activation in placental trophoblasts.

(A) Line chart of absorbance values obtained in the SARS-CoV-2 ORF8 peptides - C1q binding assay using different molarities of C1q (10, 20, and 50 nM) and 1 mM of 28 different peptides spanning SARS-CoV-2 ORF8 region. Absorbance calculated by discounting the blank (0 nM C1q). (B) The ORF8 peptides presenting dose-dependent binding with C1q (#4 red, #10 purple, #19 blue, and #25 yellow) in molecular binding model between SARS-CoV-2 ORF8 dimers and C1qA-Globular (model 1). (C) Bar charts representing docking and confidence scores for in silico docking models SARS-CoV-2 ORF8 and C1qA-G. The ORF8 peptides #4, #10, #19, and #25 represented on amino acid (aa) sequence alignment of docking models. (D) Representative docking poses of the globular domain of C1q (PDB ID: 1PK6) with SARS-CoV-2 ORF8 protein (PDB ID:7JTL) in 2D interaction diagrams. The ORF8 and C1q residues with significant interactions from each protein are highlighted, labeled, and properly annotated. The hydrogen bonds, pi-pi, and salt bridges formed between ORF8 and C1q are also shown by colored dash or arrow lines. The polar (turquoise), hydrophobic (green), positively charged (purple), and negatively charged (orange) residues are represented in colored spheres. Shown below each of the interaction diagrams is a table of interacting or contact residues between the SARS-CoV-2 ORF8 protein and the globular C1q domain, including their aa positions, aa distance measured in Angstrom (Å), and the corresponding specific binding interactions. (E) Boxplots representing the relative expression of genes analyzed in HTR8/SVneo trophoblast treated with ORF8 peptides #4, #10, #19, and #25 using 0.1 nM and 1 nM ($n = 5$–6) or mock ($n = 7$–10). (F) Boxplots representing the relative expression of genes analyzed in primary human villous trophoblast (HVT) treated with 0.1 nM and 1 nM of ORF8 peptide #10 or Mock. Relative expression was calculated by DCt method using GAPDH as the normalizing gene. (G) Boxplots representing the percentage of live C3b+ cells in mock ($n = 4$) and ORF8 peptides #4, #10, #19, and #25 0.1 nM and 1 nM treated ($n = 4$–6) HTR8/SVneo trophoblasts. Data are presented in boxplot, middle line represents means, the bound of box represents interquartile range, and whiskers represent maximum and minimum values. (H) Graphical representation of main results. Comparisons using one-way ANOVA Kruskal–Wallis with Fisher test ($p < 0.05$). Source data are available online for this figure.

compartment, including umbilical cord, amniotic fluid, and cord blood plasma, but not on the maternal side. It is well documented that hyperactivation of the complement system leads to a spectrum of pregnancy complications, including preterm birth and pre-eclampsia (Amari Chinchilla et al, 2020). For instance, preeclampsia is characterized by high levels of C5a in maternal and umbilical cord plasma, which subsequently interacts with C5aR on trophoblasts, promoting the release of anti-angiogenic factors that impair normal placentation (Ma et al, 2018). C5a is also involved in fetal cortical brain damage, resulting in long-term cognitive, behavioral, attentional, or socialization deficits observed in children born preterm (Pedroni et al, 2014). Emerging clinical evidence has demonstrated a higher risk of abnormal neurodevelopmental outcomes in SARS-CoV-2-exposed (SE) infants (Edlow et al, 2022; Fajardo Martinez et al, 2023; Rasile et al, 2022). Our observations of increased complement activation and inflammation in the fetal compartment of the placenta may suggest its involvement in COVID-19-associated abnormal neurological consequences in some exposed infants. Paradoxically, the frequency of SARS-CoV-2 viral RNA detection in the fetal compartment did not correlate with the robust complement activation and inflammation observed in the majority of biospecimens derived from the fetal compartment (amnion, umbilical cord, amniotic fluid, and cord plasma). These controversial observations suggest an alternate viral mechanism that may trigger immune rewiring in the fetal compartment independently of SARS-CoV-2 vertical transmission.

The SARS-CoV-2 genome encodes for 29 viral proteins, which include four structural proteins, 16 non-structural proteins, and nine accessory proteins (Vinjamuri et al, 2022). Among all SARS-CoV-2 proteins, the 121-aa ORF8 is the only secreted viral protein with an immunoglobulin (Ig)-like structure (Flower et al, 2021). ORF8 has been identified as a multifaceted immunomodulatory viral cytokine, mimicking host IL-17 for the activation of IL-17 signaling, downregulation of MHC I, and antagonization of IFN signaling (Lin et al, 2021; Vinjamuri et al, 2022; Wu et al, 2022). Given the limited research on ORF8 in the context of SARS-CoV-2 infection during pregnancies, our study presents a unique cohort for investigating the potential vertical transmission of ORF8 and its complement activation ability from both maternal and fetal perspectives. ORF8 was notably present in 65.22% of maternal plasma samples and over 60% of fetal fluids at term delivery,

providing biological evidence of ORF8 translocating to the fetal compartment. The widespread presence of circulating ORF8 in maternal and fetal specimens up to 216 days (maternal) or 188 days (fetal) after SARS-CoV-2 infection contrasts with our cohort's 26% vertical transmission rate of SARS-CoV-2 viral RNA. Correlation analysis shown low positive correlation between days post infection and SARS-CoV-2 N1/N2 RNA in chorion, and amniotic fluid. No or negative correlation were observed in all other analyses including circulating ORF8 levels (Fig. EV7). These findings suggest the persistent presence of ORF8 at the maternal-fetal interface, particularly on the fetal side. Since ORF8 is the only secreted viral protein with an immunoglobulin (Ig)-like structure, these results led to us hypothesize that it mimics maternal IgG antibodies potentially aiding efficient placental translocation.

Maternal IgG antibodies are passively transferred across the placenta during pregnancy, contributing to early life immune development and conferring protection against pathogens during the first months of life (Langel et al, 2020). The ability of the human IgG to cross the histological barriers of the placenta (Ciobanu et al, 2020) and its role in driving cell-mediated effector functions by complement activation has been described extensively (Damelang et al, 2023). Thus, we further explored parallels between the SARS-CoV-2 ORF8 protein and the human IgG structures to determine whether similarities in their 3D conformation can shed light on the mechanism of how ORF8 can cross the fetal compartment. Our pairwise structural analysis showed that ORF8 has an Ig-like folding (Flower et al, 2021) and is aligned closely with the human IgG-Fc region, particularly the CH2 domain (Fig. EV8). The CH2 region interacts with the neonatal Fc receptor (FcRn), which is crucial for IgG placental transfer (Brambell et al, 1960; Firan et al, 2001). Furthermore, the ORF8-specific residues overlapping with the CH2 domain are proximal to the N297 glycan site, which likely enhances its placental barrier translocation efficiency. While our data supported ORF8's Ig-like placental translocation, further studies are warranted to conclude whether ORF8 follows the same mechanism as the human IgG. Understanding the exact mechanism of how ORF8 can efficiently cross the placental barrier is viral to understanding SARS-COV-2 pathogenesis and vertical transmission.

One repercussion of ORF8 persistence is distinct overt complement activation and inflammation identified in the fetal

compartment of COVID-19-affected placentas. Recent studies of ORF8 have suggested its immunoregulatory role in the complement cascade. Through host-protein interactome studies, ORF8 has been shown to interact with complement factors, including C5 (Stukalov et al, 2021). An In vitro study have demonstrated that ORF8 can bind to C3b, which leads to two outcomes: (1) ORF8 binding to C3b prevented Complement Factor I (CFI) from proteolytic cleavage of activated form of C3 – C3b, which can lead to accumulation of C3b, thus hyperactivation of complement cascade; and (2) binding of ORF8 to C3b impeded Complement Factor B (CFB), a component of alternative pathway, from binding to C3b, which can prevent the formation of alternative pathway C3 convertase (C3bBb) (Kumar et al, 2023). Putting these 2 observations together, ORF8 binding to C3b can prevent alternative pathway activation, but may drive hyperactivation of complement cascade through other complement pathways, such as classical pathway. Indeed, our study demonstrated the importance of ORF8-C1q-mediated classical pathway activation in contributing to fetal inflammation. Our findings of abundance co-localization of ORF8 and C3b staining in COVID-19 placental tissues, and increase downstream complement factor 5 (C5) detected in amniotic fluid and cord plasma of COVID-19 pregnancies, further confirmed the hyperactivation of complement cascade during COVID-19 pregnancies. Moreover, in silico analysis of 47 previously identified ORF8 interacting proteins (Gordon et al, 2020) predicted ORF8 involvement in the activation of the complement cascade, which may contribute to COVID-19 pathogenesis (Takatsuka et al, 2022).

Herein, we demonstrated that ORF8 led to the activation of the classical complement pathway. Fetal compartment tissues that tested positive for ORF8 exhibited heightened inflammation and complement activation compared to ORF8-negative COVID-19-affected pregnancies. Although hepatocytes mainly produce complement, many other cell types, such as mononuclear phagocytes, dendritic cells, endothelial cells, and placental trophoblasts, are also capable of producing complement (Bulla et al, 2009). Our findings were further validated with in vitro ORF8 treatment of placental trophoblasts, demonstrating elevated expressions of complement genes and C3b activation. We confirmed that peptides #4, #10, #19, and #25 sufficiently induced the expression of complement genes and C3b activation in placental trophoblasts, with peptides #4 and #10 driving significant C1q expression. Our findings highlight the role of ORF8 in overt complement activation at the maternal-fetal interface, which may impact the future immune development of children exposed to antenatal maternal COVID-19.

Finally, we demonstrated that SARS-CoV-2 ORF8 can directly bind to C1q, specifically with the globular domain subcomponent A (C1qA). The strong binding interactions between ORF8 and C1qA were verified through in silico molecular docking, where we identified residues distributed along C1qA subdomains extensively binding with several residues from two or more peptides. Interestingly, several residues from peptides #4, #10, and #25 were also engaged with other C1qA residues that further stabilize the complex. We propose that the C1qA-ORF8 complex follows a "ball-in-a-glove" binding model, with ORF8 peptides coordinating to stabilize its interaction with hotspot residues in the C1qA interface. The presence and multiple involvement of the His112 and

Val 114 residues proximal to peptide #25 may have also contributed to the enhanced stability of the complex. The critical amino acids in C1qA and ORF8 identified in this study are potential hotspot residues to guide future mutational experiments and provide promise for further structure-based studies and mechanistic evaluation of the C1qA-ORF8 complex.

Taken together, we identified an interaction between host complement C1q protein and viral ORF8 protein, which drives overt inflammation and potentially contributes to pregnancy complications in both mother and child during SARS-CoV-2 infection. One limitation of the present study is that an analysis of the association between ORF8 levels and the severity of COVID-19 was not possible, given that the clinical cohort comprised only those with mild cases of COVID-19, including both mothers and newborns. Nevertheless, the clinical association between ORF8 levels and complement and inflammation, and the mechanisms involved in this host-virus interaction, are clear. Further research efforts are necessary to follow up on SE children to determine the long-term immunological consequences of SARS-CoV-2 ORF8 exposure during pregnancy. These findings will provide new insight into the development of therapeutic interventions specifically targeted for pregnant patients infected with SARS-CoV-2, which may be beneficial for the improvement of pregnancy and fetal outcomes.

# Methods

**Reagents and tools table**

| Reagent/Resource | Reference or Source | Identifier or Catalog Number |
| --- | --- | --- |
| **Experimental Models** | | |
| HRT8/SVneo trophoblast human | ATCC | Ref.: CRL-3271 |
| Human villous trophoblast cells (HVT) | iXCells Biotechnologies | Ref.: 10HU-214-1M |
| HEK-293T | ATCC | Ref.: CRL-11268 |
| THP-1 | ATCC | REF.: TIB-202 |
| **Recombinant DNA** | | |
| pCDNA-C1qA-FL-Flag | GenScript | C1qA transcript variant 1 (NM_015991.4) |
| Recombinant SARS-CoV-2 ORF8 His-tag Protein | R&D Systems | Cat.: 10918-CV |
| C1q Human protein | Complement Tech | Cat.: A099 |
| pIRES-co-GFP | In house generated | |
| pIRES-3x Vector | In house generated | |
| pIRES-ORF8-3xFlag | In house generated | |
| pIRES-ORF8-HA | In house generated | |
| pIRES-C1qA-FL-3xFlag | In house generated | C1QA transcript variant 1 (NM_015991.4) |
| pIRES-C1qA-(SP) G-3xFlag | In house generated | |
| pIRES-C1qA-(SP) D -3xFlag | In house generated | |
| pcDNA-C1qB-FL-FLAG | GenScript | C1QB transcript variant 2 (NM_000491.5) |

| Reagent/Resource | Reference or Source | Identifier or Catalog Number |
|---|---|---|
| pcDNA-C1qC-FL-Flag | GenScript | C1QC transcript variant 2 (NM_172369.5) |
| pcDNA3.1+/C-(K)-DYK vector with a C-terminal FLAG tag | GenScript | |
| **Antibodies** | | |
| Rabbit polyclonal anti-SARS-CoV-2 ORF8 antibody | Genetex | Cat. No. GTX135591 Dilution WB: 1:5000 Dilution IHC/IF: 1:2000 |
| Monoclonal Rabbit anti-C3 antibody | Abcam | Clone: EPR19394 Cat.: ab200999 Dilution: 1:2000 |
| Monoclonal mouse SARS-CoV-2 ORF8 antibody | R&D Systems | Clone:#1041422 Cat.: MAB10820-100 Dilution: 1:500 |
| Zombie Violet™ Fixable viability Kit | Biolegend | Cat.: 423113 |
| FITC-conjugated mouse monoclonal IgG1 anti-human complement C3b/iC3b | Biolegend | Clone 3E7/C3b Cat.: 846108 |
| Monoclonal Rabbit anti-C1qA antibody | Abcam | Clone EPR14634 Cat.: ab189922 Dilution: 1:5000 |
| Monoclonal Rabbit anti-C1qB antibody | Abcam | Clone EPR2981 Cat.: ab92508 Dilution: 1:2500 |
| Monoclonal Rabbit anti-C1qC antibody | Abcam | Clone EPR2984Y Cat.: ab75756 Dilution: 1:5000 |
| Monoclonal Rabbit anti-b-actin antibody | Cell signaling | Clone: 13E5 Cat.: 49705 Dilution: 1:2500 |
| Monoclonal Mouse anti-human Cytokeratin 8/18 (KTR8/18) | Thermo fisher | Clone: 5D3 Cat.: MA5-14088 Dilution: 1:250 |
| Goat anti-mouse IgG (H + L) secondary antibody Alexa Fluor 488 | Invitrogen | Cat.: A-11001 Dilution: 1:1000 |
| Goat anti-rabbit IgG (H + L) secondary antibody Alexa Fluor 647 | Invitrogen | Cat.: A-21244 Dilution: 1:1000 |
| **Oligonucleotides and other sequence-based reagents** | | |
| SARS-CoV-2 droplet digital PCR (ddPCR) Kit | Bio-Rad | Cat.: 12013743 |
| PCR primers | This study | Table 2 |
| SARS-CoV-2 ORF8 peptides | GenScript | Cat.: GSCRPT-RP30017 |
| **Chemicals, Enzymes and other reagents** | | |
| TRIzol | Invitrogen | Cat. #15596026 |
| TMB Substrate Reagent Set (BD OptEIA™) | BD Biosciences | Cat.: 555214 |
| iScript cDNA synthesis kit | Bio-Rad | Cat.: 1708891 |
| SsoAdvanced Universal SYBR Green Supermix | Bio-Rad | Cat.: 1725270 |
| Antigen Unmasking Solution, Tris-Based | Vector Lab | Cat no. H-3301 |
| ReadyProbes™ Endogenous HRP and AP Blocking Solution (1X) | Invitrogen | Cat. no. R37629 |

| Reagent/Resource | Reference or Source | Identifier or Catalog Number |
|---|---|---|
| VECTASTAIN® ABC-HRP Kit (Anti-Rabbit) | Vector Lab | Cat. No. PK-4001 |
| DAB Substrate Kit, Peroxidase (HRP), with Nickel, (3,3'-diaminobenzidine) | Vector lab | Cat. No. SK-4100 |
| Hematoxylin QS Counterstain | Vector lab | Cat. No. H-3404-100 |
| VectaMount® AQ Aqueous Mounting Medium | Vector lab | Cat. No. H-5501-60 |
| Pierce™ Protein A/G Agarose beads | Thermo Fisher Scientific | Ref: 20421 |
| **Software** | | |
| Partek® Flow® software, version 10.0.23.0131 | Partek | |
| FlowJo software | BD | |
| Proteome Discoverer software | Thermo Fisher Scientific | |
| PyMOL Molecular Graphics System, Version 2.5.5 | PyMOL | |
| LigPrep Wizard version 3.3 | Schrödinger, LLC, New York | |
| Glide module of Schrödinger | Schrödinger, LLC, New York | |
| Schrödinger suite version 2015-1 | Schrödinger, LLC, New York | |
| GraphPad PRISM 9.0 software | GraphPad Software | |
| R, Rtsne package | R Software | |
| Ingenuity Pathway Analysis | QIAgen | |
| **Other** | | |
| RNeasy MinElute Cleanup Kit | QIAgen | REF.: 74204 |
| RiboPure blood Kit | Invitrogen | REF.: AM1928 |
| QIAmp viral RNA mini kit | QIAgen | REF.: 52904 |
| KAPA mRNA Hyperprep Kit | Roche | Kit Code: KK8581 |
| Human Cytokine/Chemokine 71-Plex Discovery Assay® Array | Eve Technologies | HD71 |
| Human MMP and TIMP Discovery Assay® Array for Cell Culture and non-blood samples | Eve Technologies | HMMP/TIMP-C, O |
| Human Complement 13-Plex Discovery Assay® Array | Eve Technologies | HDCMP13 |
| LEGEND MAX SARS-CoV-2 Spike S1 human IgM ELISA kit | Biolegend | Cat. #448207 |
| LEGEND MAX SARS-CoV-2 Nucleocapsid human IgG ELISA Kit | Biolegend | Cat. # 448107 |
| Normal Goat Serum | Cell signaling | Cat. #5425 |
| hcG ELISA kit | CALBIOTECH | Cat.: #HC251F |
| Trophoblast Growth Medium | iXCells Biotechnologies | Cat.: MD-0058 |
| Lipofectamine™ 3000 Transfection Reagent | Thermo Scientific | Cat.: L3000015 |
| 2x Laemmli Sample Buffer | Bio-Rad | Cat.: 1610737 |
| **Instruments** | | |
| BIO-RAD CFX Opus 384 Touch Real-Time PCR Detection System | Bio-Rad | |
| FACSCelesta | BD | |

| Reagent/Resource | Reference or Source | Identifier or Catalog Number |
|---|---|---|
| Reader Varioskan Lux, and SkanIt microplate reader software | Thermo Fisher Scientific | |
| NanoDrop 1000 spectrophotometer | Thermo Scientific | |
| Illumina SovaSeq 6000 | Illumina | |
| TapeStation 4200 | Agilent | Part number: G2991BA |
| TissuLyser II | QIAgen | |
| ChemiDoc Touch Imaging system | Bio-Rad | |
| *Fusion Lumos - Dionex Ultimate 3000 RSLCnano* | Thermo Scientific | |

## SARS-CoV-2-exposed maternal-fetal cohort and biospecimens collection

A retrospective clinical cohort of adult pregnant women (>18 years of age) with qRT-PCR confirmed SARS-CoV-2-positive nasopharyngeal swab test were enrolled in this COVID-19 pregnancy cohort study at the Department of Obstetrics and Gynecology & Women's Health Institute in the Cleveland Clinic. Clinical tests including SARS-CoV-2 qRT-PCR testing and complete blood count were performed at the Cleveland Clinic Pathology & Laboratory Medicine Institute (Robert J. Tomsich) between March 2020 and Feb 2021. Pregnant controls without COVID-19 or upper respiratory infection symptoms who tested negative for SARS-CoV-2 by qRT-PCR were concurrently recruited. The trimester of pregnancy in which SARS-CoV-2 detection occurred was defined as first trimester (0–13 weeks), second trimester (14–27 weeks) and third trimester (≥28 weeks). All cases were categorized as mild maternal COVID-19 disease. The maternal-fetal biospecimens were obtained on the day of admission for delivery. Biofluids collected included maternal plasma, umbilical cord plasma, newborn WB, and amniotic fluid which were stored at −80 °C for future analysis. Tissue specimens collected include placental membranes (amnion and chorion) and umbilical cords which were either (i) snap-frozen and stored at −80 °C or (ii) fixed in formalin and stored at 4 °C until use. The study cohort and sample collection were approved by the Cleveland Clinic Institutional Review Board (IRB) and Institutional Biosafety Committee (IBC) under IRB number 20-626. Informed consent for study participation was obtained for all participants before enrollment.

## Ultrasensitive droplet digital PCR (ddPCR) detection of SARS-CoV-2

The ultrasensitive detection of SARS-CoV-2 was conducted using SARS-CoV-2 droplet digital PCR (ddPCR) Kit (12013743, Bio-Rad) according to the manufacturer's instructions. The US FDA granted Emergency Use Authorization (EUA) for Bio-Rad's SARS-CoV-2 ddPCR Kit in the US on May 1st, 2020. Total RNA was extracted from amnion, chorion, and umbilical cord using TRIzol® (Invitrogen) according to the manufacturer's instructions. For the cell-free body fluids specimens, viral RNA was extracted from the amniotic fluid, maternal plasma and cord plasma using the QIAmp Viral RNA mini kit (QIAgen) according to the manufacturer's

instructions. Finally, newborn WB RNA was extracted using a RiboPure blood kit (Invitrogen) according to the manufacturer's instructions. For all ddPCR detection of SARS-CoV-2 nucleocapsid (N) 1, N2 genes and human RRP30 gene, 100 ng of extracted total RNA or viral RNA was used for the tissue and biofluid specimens, respectively. The cutoff of 2 counts for at least one viral target (N1 or N2), determined by the manufacturer, was used to classify samples as SARS-CoV-2 negative and positive.

## Transcriptomic profiling—bulk RNAseq

Snap frozen placental tissues including amnion, chorion and umbilical cord samples were subjected to tissue homogenization using Qiagen Tissuelyser II in TRIzol (Invitrogen), followed by total RNA extraction according to the manufacturer's instructions. Extracted total RNA were then subjected to RNA cleanup using RNeasy MinElute Cleanup Kit (Qiagen). Purified total RNA was processed into RNA sequencing libraries with the KAPA mRNA Hyperprep Kit according to the manufacturer's instructions (Roche). Quality control was tested before and after library preparation using an Agilent TapeStation 4200 and sequencing was performed by the Cleveland Clinic Genomics Core using the platform Illumina NovaSeq 6000 (Illumina). Demultiplexed fastq files were received from the Genomics Core and utilized for further analysis. Fastq files were analyzed with Partek® Flow® software, version 10.0.23.0131 (Partek). Alignment was performed using STAR 2.7.8a with hg38 and gene counts by Partek Expectation/ Maximization (E/M) algorithm. The transcript model was hg38-RefSeq Ensembl transcript release 108, and the normalization method used was Trimmed Mean of M-values (TMM). Significantly differentially expressed genes between the groups analyzed were identified in Partek Flow by fold-change ≥|2| and FDR-adjusted alue <0.05 using the Gene set Analysis (GSA) method (lognormal with shrinkage) for genes with an average coverage ≥1 read per million.

## Biofluids proteomics profiling

Biofluids proteomics profiling across 97 proteins was performed using three targeted bead-based multiplex immunoassays available from Eve Technologies. The three panels of proteins analyzed were: (i) inflammation—Human Cytokine/Chemokine 71-Plex Discovery Assay® Array (HD71); (ii) matrix metalloproteinases (MMPs) and tissue inhibitors of metalloproteinases (TIMPs)—Human MMP and TIMP Discovery Assay® Array for Cell Culture and non-blood samples (HMMP/TIMP-C, O); and (iii) complements—Human Complement 13-Plex Discovery Assay® Array (HDCMP13). Values in pg/mL or ng/mL were provided by Eve Technologies through interpolation with standard curves. The results are shown in the fold change. Fold change was calculated by the "Conc. of cytokine x in Sample/Average (Conc. of protein conc. of cytokine x in all Control samples)".

## SARS-CoV-2 ORF8 ELISA

Maternal plasma, umbilical cord plasma, and amniotic fluid samples were analyzed with the SARS-CoV-2 ORF8 ELISA, as previously published (Wu et al, 2022). In our study, we calculate the sensitivity and specificity of in-house ORF8 ELISA according to

the type of samples tested. Sensitivity and specificity describe the proportions of positive or negative results among individuals known to be exposed to the infection. Sensitivity is the probability of a positive outcome in patients (true positive) and is calculated as: Sensitivity = a (true positive)/a + c (true positive + false negative) × 100. Specificity is the probability of a negative result in non-patients (true negative) and is calculated as: Specificity = d (true negative)/b + d (true negative + false positive) × 100. The sensitivity and specificity observed for in-house ORF8 ELISA range from 60–65.2% and 100%, respectively.

## SARS-CoV-2 Spike S1 IgM and SARS-CoV-2 Nucleocapsid IgG ELISA

Enzyme immunoassays (ELISA) for quantification of anti-SARS-CoV-2 Spike S1 IgM (Cat. #448207, Biolegend) and anti-SARS-CoV-2 Nucleocapsid IgG (Cat. # 448107, Biolegend) were performed on maternal plasma, umbilical cord plasma, and newborn plasma according to the manufacturer's instructions.

## Immunohistochemistry

Placental amnion and chorion/choriodecidua tissues were fixed in formalin and paraffin-embedded prior to sectioning and immuno-histochemical staining. Ten-micron sections were dewaxed, rehydrated and stained with hematoxylin and eosin (H&E) or via immunohistochemical staining with rabbit polyclonal anti-SARS-CoV-2 ORF8 antibody (Cat. No. GTX135591, Genetex) and recombinant anti-C3 antibody (clone: EPR19394; Abcam) and counterstained with hematoxylin.

## iPSC-derived trophoblasts

Human iPSCs were cultured according to previously described methods to obtain terminally differentiated placental trophoblasts (Horii et al, 2016) with modifications. Briefly, human iPSCs (iPSC-9) were cultured in a 12-well StemPro (Thermo Fisher, MA)-based minimal medium [KnockOut DMEM/F12 containing 1% insulin-transferrin-selenium A, 1x nonessential amino acid, 2% (vol/vol) FBS, 2mM L-glutamine, 100 ng/ml heparin sulfate] designated as "EMIM" and rested for 2 days. After resting, the cells were considered in day 0 of differentiation. To achieve the intermediary step of differentiation, the cells were switched to EMIM medium supplemented with 10 ng/mL of human BMP4 (Peprotech, NJ) for up to 4 days, resulting in the formation of CTB stem-like colonies. To induce terminal differentiation, the cells were trypsinized and replated in FCM [KnockOut DMEM/F12 containing 20% (vol/vol) KnockOut serum replacement (Thermo fisher, MA), 1x GlutaMAX (Thermo Fisher, MA), 1x nonessential amino acid and 0.1 mM 2-mercaptoethanol]. The cells were supplemented with 10 ng/mL BMP4 for seven days (day 7). The culture supernatant underwent evaluation using the human chorionic gonadotropin (hCG) ELISA (Calbiotech) according to manufacturer's instructions. The transcripts associated with iPSC, CTB, STB, and EVT were quantified using qRT-PCR, as described below. Compared to day 0, the supernatant of iPSC-derived trophoblasts differentiated at day 7 exhibited high levels of the functional trophoblast marker, human chorionic gonadotropin (hCG) (Fig. EV5A). At the transcriptional level, day 7 iPSC-derived

trophoblasts showed a decrease in the expression of POU5F1, a transcript associated with iPSCs (Fig. EV5B). Furthermore, on day 7, there was a statistically significant increase in transcript levels correlated with trophoblast phenotypes of CTB (KRT7, CDX2, and TP63, Fig. EV5C), STB (CGA, CGB, and PSG4, Fig. EV5D), and EVT (HTRA4, Fig. EV5E). As a result, the in-house-developed iPSC trophoblasts demonstrated a blended phenotype of CTB/STB/EVT trophoblasts.

## In vitro SARS-CoV-2 ORF8 treatment in placental trophoblasts

Immortalized trophoblast cells HTR8/SVneo (CRL-3271, ATCC) were cultured in Roswell Park Memorial Institute (RPMI) 1640 Medium containing 10% Fetal Bovine Serum (FBS) and 1% Penicillin-Streptomycin (P/S) (all from Thermo Fisher Scientific) at 37 °C in 5% $CO_2$. Next, $2 \times 10^5$ HTR8/SVneo cells were plated in 24-well tissue culture plates (Genesee Scientific) and incubated overnight with RPMI supplemented with 10% FBS and 1% P/S. Cells were pre-incubated for 2 h with serum-free RPMI followed by incubation with 10, 20, or 100 ng/mL of HEK293T cells-derived recombinant SARS-CoV-2 ORF8 in RPMI supplemented with 10% FBS and 1% P/S for 8 or 16 h. Experimental conditions optimized with the HTR8/SVneo cells were applied to iPSC-derived trophoblasts described above and the primary human villous trophoblasts (HVT) (10HU-214-1M, iXCells) cultured in Trophoblast Growth Medium (MD-0058, iXCells) at 37 °C in 5% $CO_2$.

## Gene expression profiling by qRT-PCR

Total RNA was extracted using TRIzol® (Invitrogen) according to the manufacturer's instructions. RNA concentration was determined by NanoDrop 1000 spectrophotometer (Thermo Scientific). Extracted total RNA was reverse-transcribed using iScript cDNA synthesis kit (Bio-Rad) according to the manufacturer's instructions. Gene expression qRT-PCR profiling was performed with 10 ng of cDNA/well using SsoAdvanced Universal SYBR Green Supermix (Bio-Rad) and primer pairs listed in Table 2. All qRT-PCR reactions were performed using BIO-RAD CFX Opus 384 Touch Real-Time PCR Detection System on 384-well plates. Gene expression fold change was calculated with the ΔΔCt method using Microsoft Excel. Briefly, ΔΔCt = ΔCt(COVID-19 sample) − ΔCt((CDC)) with ΔCt = Ct(gene-of-interest)—Ct(housekeeping gene-GAPDH). The fold change for each gene is calculated as $2^{-\Delta\Delta Ct}$.

## Flow cytometry

Trophoblast cells were treated with TrypLE Express Enzyme (1X) no phenol red (Thermo Fisher Scientific) and washed with RPMI supplemented with 10% FBS and 1% P/S prior to flow cytometry staining. Live/dead staining was first performed using Zombie Violet™ Fixable viability Kit labeling (Biolegend) followed by surface staining with FITC-conjugated mouse monoclonal IgG1 anti-human complement C3b/iC3b (clone 3E7/C3b; Biolegend). Stained cells were fixed with 4% formaldehyde and flow cytometry analysis was conducted on BD FACSCelesta using BD FACSDiva acquisition software. All FACS data was analyzed using FlowJo software (BD Biosciences).

**Table 2. Primer pairs utilized in qPCR analysis.**

| Gene | Sequence forward | Sequence reverse |
|------|-----------------|------------------|
| GAPDH | ACCAGGTGGTCTCCTCTGA | TGTAGCCAAATTCGTTGTCATACC |
| C1QA | CAACAGGAGGCAGAGGCA | CTTTCTTCCCGTCTGGTGC |
| C1QB | GCTTCCCAGGAGGCGT | GAGCAGGAGCAACATCAGTA |
| C1QC | ATCTGAGGACATCTCTGTGC | ATCCCGGAGAAGGAACTGC |
| CFH | CTATTTGTGTAGCAGAAGATTGC | ATAGATAGCCTGGGTGCCTT |
| IL1A | ATCAGTACCTCACGGCTGCT | ATAGATAGCCTGGGTGCCTT |
| IL17A | AATCTCCACCGCAATGAGGA | ACGTTCCCATCAGCGTTGA |
| IL17E | CCCCTGGAGATATGAGTTGG | GTCTGGTTGTGGTAGAGCAG |
| POU5F1 | TGGGCTCGAGAAGGATGTG | GCATAGTCGCTGCTTGATCG |
| KRT7 | AGGATGTGGATGCTGCCTAC | CACCACAGATGTGTCGGAGA |
| CDX2 | TTCACTACAGTCGCTACATCACC | TTGATTTTCCTCTCCTTTGCTC |
| CGA | CAACCGCCCTGAACACATCC | CAGCAAGTGGACTCTGAGGTG |
| CGB | ACCCTGGCTGTGGAGAAGG | ATGGACTCGAAGCGCACA |
| PSG4 | CCAGGGTAAAGCGACCCATT | AGAATATTGTGCCCGTGGGT |
| HLAG | CAGATACCTGGAGAACGGGA | CAGTATGATCTCCGCAGGGT |
| HTRA4 | AAAGAACTGGGGATGAAGGATTC | TGACGCCAATCACATCACCAT |

## ELISA-based C1q-ORF8 binding assay

Microlon® high binding 96 wells polystyrene half area plates (Greiner Bio-one) were coated with 5 nM, 10 nM, 25 nM, or 100 nM of recombinant human C1q (Complement Technology) in phosphate buffered saline (PBS) pH 7.4 overnight at 2–8 °C. Plates were then washed two times with TBST (tris-buffered saline 0.05% tween-20, pH 8.0) and blocked with 100 mL/well of 3% BSA/TBST buffer for 1 h at room temperature. After washes, 50 μl of 5 nM, 10 nM, 50 nM, and 100 nM of SARS-CoV-2 ORF8 in PBS were applied in triplicates and incubated for 2 h at room temperature. Plates were washed 4 times and incubated for 1 h at room temperature with 1:500 of a polyclonal antibody anti-ORF8 (GTX135591, GeneTex) previously biotinylated. Revelation was conducted with TMB Substrate Reagent Set (BD OptEIA™, BD Biosciences) and 2N sulfuric acid, and the plate was read at 450 nm using the plate reader Varioskan Lux, and Skanlt microplate reader software (Thermo Fisher Scientific). The absorbance of each well was reduced from the blank wells.

## ELISA-based ORF8 peptides-C1q binding assay

28 SARS-CoV-2 ORF8 peptides of 15-aa in length spanning the full-length 365-aa ORF8 were synthesized by GenScript. Lyophilized ORF8 peptides were reconstituted based on the manufacturer's instructions. Microlon® high binding 96 wells polystyrene half area plates (Greiner Bio-one) were coated with 1 mM of each of the 28 different SARS-CoV-2 peptides (GenScript Biotech) in carbonate buffer 50 mM pH 9.6 overnight at 2–8 °C. After 1 h of incubation and washes, 5 nM, 10 nM, 25 nM, or 100 nM of recombinant human C1q (Complement Technology) in 3% BSA/TBST buffer was applied in triplicates and incubated for 2 h at room temperature. Plates were washed and incubated for 1 h at

room temperature with sheep polyclonal antibody anti-C1Q-HRP (Complement Technology) diluted 1:1000 in 3% BSA TBST buffer. Reaction revelation, read, and absorbance calculation were conducted as described above.

## Plasmid construction

Plasmid constructs containing human C1QA transcript variant 1 (NM_015991.4), C1QB transcript variant 2 (NM_000491.5), and C1QC transcript variant 2 (NM_172369.5) in a pcDNA3.1 + /C-(K)-DYK vector with a C-terminal FLAG tag were obtained commercially (GenScript). The SARS-CoV-2 ORF8 construct in pIRES vector with a C-terminal FLAG tag was constructed as previously described (Wu et al, 2022). All DNA cloning was performed using Gibson assembly in *Escherichia coli* Top10 competent cells, which include the construction of full-length and truncated C1qA plasmids in pIRES vector with a C-terminal 3xFLAG tag, and SARS-CoV-2 ORF8 construct in pIRES vector with a C-terminal HA tag. Briefly, PCR products were amplified from pcDNA3.1 p*C1QA* transcript variant 1 (NM_015991.4): (i) C1qA signal peptide + C1qA FL + 3xFLAG (785 bp), (ii) C1qA signal peptide + C1qA G + 3xFLAG (444 bp); and (iii) C1qA signal peptide + C1qA D + 3xFLAG (383 bp). PCR fragments were cloned into pIRES vector using NEBuilder® HiFi DNA Assembly Master Mix according to the manufacturer's instructions (New England BioLabs). A pIRES-SARS-CoV-2 ORF8-HA plasmid was obtained using the same approach.

## Cell transfection, co-immunoprecipitation, and immunoblotting

HEK-293T cells and immortalized trophoblast cells HTR8/SVneo (CRL-3271, ATCC) were transfected with empty vector pIRES, each

plasmid alone, or a pairwise combination of C1qA, B, or C with pIRES SARS-CoV-2 ORF8-FLAG using with standard polyethyleneimine (PEI) or Lipofectamine 3000 transfection procedures for 48 h. HEK-293T or HTR8/SVneo cells were transfected with (i) empty vector pIRES, each plasmid alone, or a pairwise combination of C1qA, B, or C with pIRES SARS-CoV-2 ORF8-FLAG or (ii) empty vector pIRES, each plasmid alone, or a pairwise combination of C1qA-FL, -G, or -D with pIRES-SARS-CoV-2 ORF8-HA using standard PEI (HEK-293T) or Lipofectamine 3000 (HTR8/SVneo) transfection procedures. At 48 h post-transfection, the cells were washed with PBS and lysed with NP40 lysis buffer (50 mM Tris-HCl (pH 7.5), 150 mM NaCl, 1% NP-40) supplemented with complete protease inhibitor EDTA-free cocktail (Roche) shaking gently for 45 min at 4 °C. Whole-cell extracts were centrifuged at maximum speed and filtered through a 0.45 μm filter (Thermo Fisher Scientific), followed by pre-clearing with Sepharose beads at 4 °C for 1 h under rotation. Pre-cleared whole-cell extracts were incubated overnight with either (i) mouse monoclonal antibody anti-SARS-CoV-2 (Mab10820, R&D Systems, Minneapolis, MN, USA) or (ii) mouse monoclonal anti-FLAG M2 (Sigma-Aldrich). The protein-antibody complex was incubated with Pierce™ Protein A/G Agarose beads (Thermo Fisher Scientific) at 4 °C for 2 h under rotation. Co-immunoprecipitants were eluted with 2x Laemmli Sample Buffer (Bio-Rad) at 60 °C for 10 min. Bead-bound immunocomplexes were suspended in 2xLaemmli Sample Buffer (Bio-Rad) and incubated at 60 °C for 10 min, followed by centrifugation to collect the unbound immunocomplexes released from the beads in the sample buffer. All protein samples were heated at 95 °C for 10 min prior to immunoblotting. Proteins were separated by SDS–polyacrylamide gel electrophoresis (SDS-PAGE) with Precision plus dual color protein standards (Bio-Rad) and examined by standard Western blotting procedures. Anti-b-actin was used as the internal loading control. Images were visualized using a ChemiDoc Touch Imaging system (Bio-Rad). The following antibodies were used for immunoblotting: rabbit monoclonal anti-b-actin (13E5, Cell Signaling), recombinant anti-C1QA antibody [EPR14634] (ab189922, Abcam), recombinant anti-C1QB antibody [EPR2981] (ab92508, Abcam), recombinant anti-C1QC antibody [EPR2984Y] (ab75756, Abcam), rabbit anti-FLAG antibody (F7425, Millipore), rabbit polyclonal SARS-CoV-2 ORF8 antibody (GTX135591, GeneTex).

### LC-MS method

For the protein digestion, samples were cut from the gel, and the bands were then washed/destained in 50% ethanol, 5% acetic acid and then dehydrated in acetonitrile. The samples were then reduced with DTT and alkylated with iodoacetamide prior to the in-gel digestion. All bands were digested in-gel using trypsin/chymotrypsin, by adding 10 μL of 5 ng/μL trypsin/chymotrypsin in 50 mM ammonium bicarbonate and incubating overnight at room temperature to achieve complete digestion. The peptides that were formed were extracted from the polyacrylamide in two aliquots of 30 μL of 50% acetonitrile with 5% formic acid. These extracts were combined and evaporated to <10 μL in Speedvac and then resuspended in 0.1% formic acid to make up a final volume of ~30 μL for LCMS analysis. The LC-MS system was a Finnigan LTQ-Obitrap Fusion Lumos hybrid mass spectrometer system. The HPLC column was a Dionex 15 cm × 75 μm id Acclaim Pepmap C18, 2 μm, 100 Å reversed-phase capillary chromatography column. Five μL volumes of the extract were injected and the peptides eluted from the column by an acetonitrile/0.1% formic acid gradient at

a flow rate of 0.25 μL/min were introduced into the source of the mass spectrometer on-line. The microelectrospray ion source was operated at 2.5 kV. The digest was analyzed using the data-dependent multitask capability of the instrument acquiring full scan mass spectra to determine peptide molecular weights and product ion spectra to determine amino acid sequence in successive instrument scans. Samples were analyzed in *Fusion Lumos - Dionex Ultimate 3000 RSLCnano* and the data were searched against the human SwissProtKB protein database with the program Proteome Discoverer.

### In silico SARS-Cov-2 ORF8 - C1q-G interaction

The crystal structures of SARS-CoV-2 ORF8 (PDB ID 7JTL) (Flower et al, 2021) and the Globular Head of C1q (PDB ID 1PK6) (Gaboriaud et al, 2003) were retrieved from The Protein Data Bank (PDB) (Berman et al, 2000). Protein–protein docking simulations were performed using the online platform HDOCK, where ten different models were generated and ranked according to the iterative scoring function ITScorePP or ITScorePR (docking score), and docking score-dependent confidence score. Models with a confidence score >0.7 and docking score < −200 are considered to have binding likeliness (Yan et al, 2020). Visualization of model structure was conducted using PyMOL Molecular Graphics System, Version 2.5.5.

To further investigate the residues, potential type, and strength of binding interactions involved between ORF8 peptides and C1qA, we performed a validation experiment using a parallel protein-ligand docking of the complex. Specifically, the crystal structures of C1qA and SARS-CoV-2 ORF8 were re-extracted and prepared using the LigPrep Wizard version 3.3 in the Schrödinger suite version 2015-1 (LigPrep, version 3.3; Schrodinger, LLC, New York (2015)). Molecular docking was performed in Glide module of Schrödinger (Schrödinger Release 2021-4: Glide, Schrödinger, LLC, New York, NY, 2021), with the docking grid size of $10 \times 10 \times 10$ Å set around the interaction site of ORF8 and C1qA complex. The grid points calculated within a region or an enclosing box defined with a set of predicted binding site residues, initially identified from peptides #4, #10, #19, and #25. The scoring function of Glide was used to predict the binding affinity scores (Schrödinger Release 2021-4: Glide (Schrödinger, LLC, 2021). The protein-ligand docking poses and interactions were analyzed in Maestro (Release, S. 1: Maestro (Schrödinger, LLC, 2017) - Schrödinger suite version 2015-1).

### Immunofluorescence staining and colocalization analysis

Placenta chorion/choriodecidua and amnion from control and COVID-19 pregnancy specimens were fixed in formalin and paraffin-embedded prior to sectioning and immunofluorescence staining. Ten-micron sections were dewaxed, rehydrated and stained with the following antibodies: rabbit anti-SARS-CoV-2 ORF8 (GTX135591; Genetex), mouse anti-Cytokeratin 8/18 (KTR8/18) (5D3) (MA5-14088; Thermo fisher), mouse anti-SARS-CoV-2 ORF8 (1041422) (MAB10820; R&D Systems) and rabbit anti-C3/C3b antibody (EPR19394) (ab200999; Abcam), followed by goat anti-mouse AF488 (Invitrogen) or goat anti-rabbit AF647 (Invitrogen) as secondary antibodies. Cells and placenta tissues were washed, mounted, and examined with an ECHO microscope. Images were collected and processed using

ImageJ software. Post-acquisition analysis of protein colocalization was performed using Fiji (ImageJ software), where circular regions of interest were defined and analyzed for intensity of ORF8 and C3b in Chorion and Amnion tissues using Coloc 2 analysis.

## In vitro SARS-CoV-2 ORF8 peptides treatment in placental trophoblasts

Immortalized trophoblast cells HTR8/SVneo (CRL-3271, ATCC) were cultured in Roswell Park Memorial Institute (RPMI) 1640 Medium containing 10% Fetal Bovine Serum (FBS) and 1% Penicillin-Streptomycin (P/S) (all from Thermo Fisher Scientific) at 37 °C in a 5% $CO_2$ atmosphere. For the standardization assay, HTR8/SVneo cells were starved for 2 h with serum-free RPMI 1640. After starvation, cells were incubated with 0.1 nM or 1 nM of SARS-CoV-2 ORF8 peptides 4, 10, 19, or 25 in RPMI 10% FBS 1% P/S during 8 or 16 h. Experimental conditions optimized with the HTR8/SVneo cells were applied to human villous trophoblasts (HVT) (10HU-214-1M, iXCells) cultured in Trophoblast Growth Medium (MD-0058, iXCells) at 37 °C in 5% $CO_2$.

### Data and statistical analyses

Data from global transcriptomics and proteomics were submitted to an exploratory analysis conducted in the R environment (R Core Team, 2020) or in GraphPad PRISM 9.0 software (GraphPad Software). tSNE plots were generated in R using the Rtsne package (Van der Maaten and Hinton, 2008). Enrichment analysis of the canonical pathways of the significantly differentially expressed genes, proteins, or metabolites were conducted using Ingenuity Pathway Analysis (QIAgen), Enrichr gene set enrichment analysis (Chen et al, 2013; Kuleshov et al, 2016; Xie et al, 2021) or STRING database. Further statistical tests were done either in R (R Core Team, 2020) or GraphPad PRISM 9.0 software (GraphPad Software, San Diego, California, USA). Comparisons between the two groups were performed using the Welch t-test and Mann–Whitney U test. For comparisons between more than two groups or time points, one-way ANOVA Kruskal–Wallis with Dunn's post-test was used. For correlation analysis simple linear regression and Spearman's rank correlation test was used.

## Data availability

The data have been deposited with links to BioProject accession number PRJNA1140360 in the NCBI BioProject database (https://www.ncbi.nlm.nih.gov/bioproject/) and all sequence data reported in this study will be publicly available on the NCBI GEO.

The source data of this paper are collected in the following database record: biostudies:S-SCDT-10_1038-S44318-024-00260-9.

## Peer review information

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

## Acknowledgements

This work was funded by Clinical and Translational Science Collaborative (CTSC) Pilot COVID Pilot fund #RES515531 (RMF) and US NIH grants: AI129534, AI298847 (KNS), R01AI140705, R01AI140718, R01AI116585 (JUJ), AI140718, AI172252 (JUJ and KNS), R00DE028573, R01DE033391 (WC). The Fusion Lumos instrument was purchased via an NIH shared instrument grant, 1S10OD023436-01. The clinical aspects of the study design were developed in conjunction with Dr. Rachel Pope from University Hospitals and Dr. Kelly Gibson from MetroHealth. The authors also acknowledge Mr. Muhammed Aadhil Muhammed Fazlul Haq (Biological and Agricultural Engineering, Texas A&M University) for the helpful comments on the protein-ligand docking verification in this study.

## Author contributions

**Tamiris Azamor**: Conceptualization; Data curation; Formal analysis; Investigation; Methodology; Writing—original draft. **Débora Familiar-Macedo**: Data curation; Formal analysis; Investigation; Methodology; Writing—review and editing. **Gielenny M Salem**: Formal analysis; Investigation; Methodology. **Chineme Onwubueke**: Investigation; Writing—original draft; Writing—review and editing. **Ivonne Melano**: Investigation; Writing—review and editing. **Lu Bian**: Investigation; Methodology. **Zilton Vasconcelos**: Investigation; Methodology. **Karin Nielsen-Saines**: Investigation. **Xianfang Wu**: Investigation; Methodology. **Jae U Jung**: Investigation; Methodology. **Feng Lin**: Investigation; Methodology. **Oluwatosin Goje**: Resources; Investigation; Methodology. **Edward Chien**: Resources; Investigation; Methodology. **Steve Gordon**: Resources; Investigation; Methodology. **Charles B Foster**: Resources; Investigation; Methodology. **Hany Aly**: Resources; Investigation; Methodology. **Ruth M Farrell**: Resources; Funding acquisition; Investigation; Methodology; Project administration. **Weiqiang Chen**: Conceptualization; Resources; Data curation; Formal analysis; Supervision; Funding acquisition; Validation; Investigation; Visualization; Methodology; Writing—original draft; Project administration; Writing—review and editing. **Suan-Sin Foo**: Conceptualization; Resources; Data curation; Formal analysis; Supervision; Funding acquisition; Validation; Investigation; Visualization; Methodology; Writing—original draft; Project administration; Writing—review and editing.

Source data underlying figure panels in this paper may have individual authorship assigned. Where available, figure panel/source data authorship is listed in the following database record: biostudies:S-SCDT-10_1038-S44318-024-00260-9.

## Disclosure and competing interests statement

The authors declare no competing interests.

# Expanded View Figures

**Figure EV1.　Complement-associated inflammatory responses in umbilical cord delivered from COVID-19-affected pregnancies.**　▶

(A) tSNE plot of global transcriptomics for chorion of controls (emerald) and COVID-19 (purple). Volcano plot representing genes upregulated (red dots) and downregulated (blue dots) in the umbilical cord of COVID-19 relative to controls. (B) Bar plot representing gene ontology biological pathways associated with upregulated DEGs in the umbilical cord of COVID-19 relative to controls. (C) Violin plots representing the expression count values for individual genes related to complement activation in control ($n = 5$–6) and umbilical cords from COVID-19-affected pregnancies ($n = 19$). (D) Network analysis of DEGs in umbilical cords from COVID-19-affected pregnancies relative to controls. (E) Violin plots representing the expression count values for individual genes related to complement-associated inflammation in control ($n = 6$) and umbilical cords from COVID-19-affected pregnancies ($n = 19$). (F) Comparative analysis of DEGs ($-2 < FC > 2$, $p < 0.05$) in COVID-19 umbilical cord and amnion transcriptomics, depicting genes related to complement and complement-associated inflammation. Upregulated genes in red, downregulated genes in blue. Data are presented as means ± SEMs, using Mann–Whitney U test ($p < 0.05$). Source data are available online for this figure.

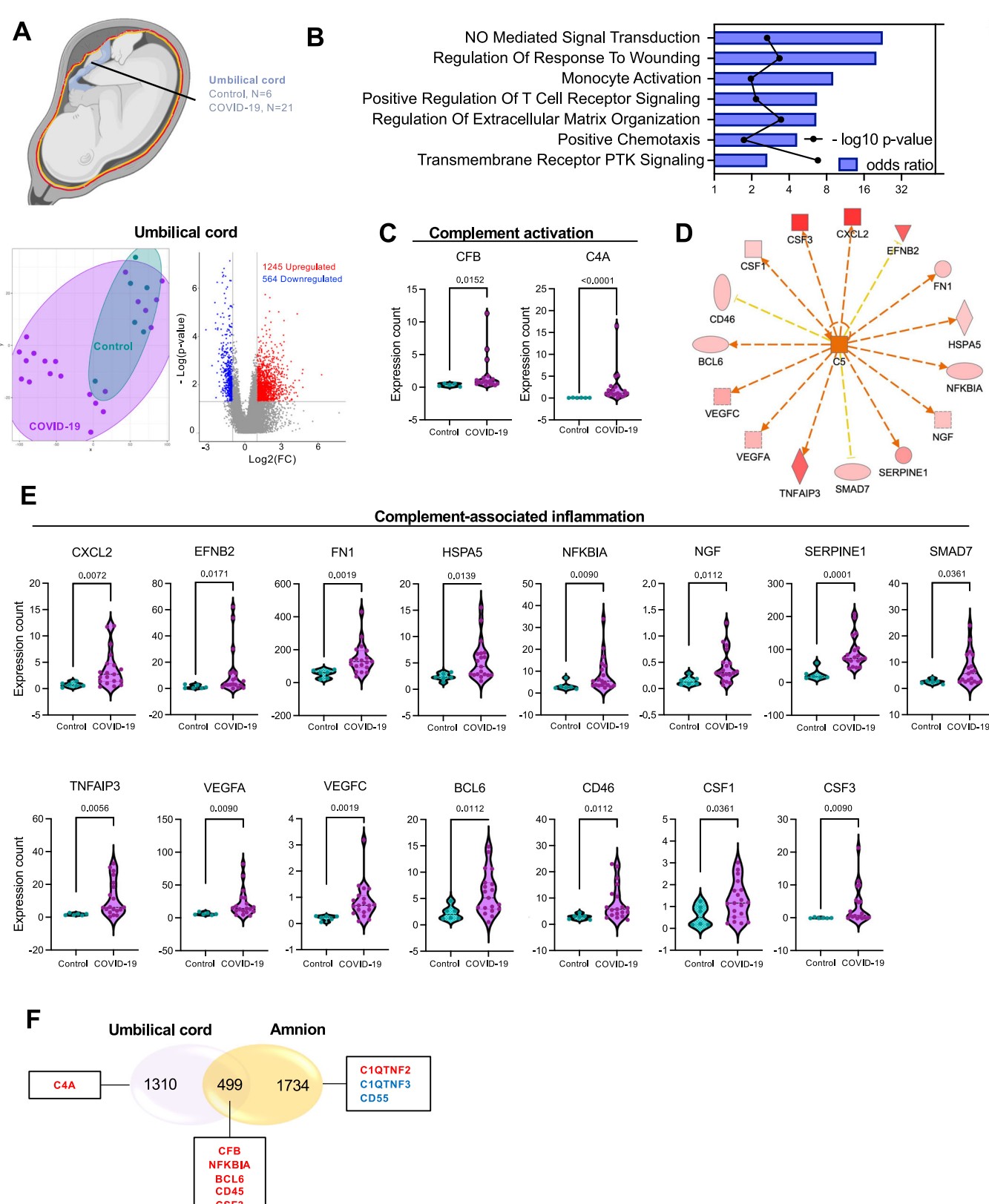

## A Differentially expressed genes/proteins: Upregulated

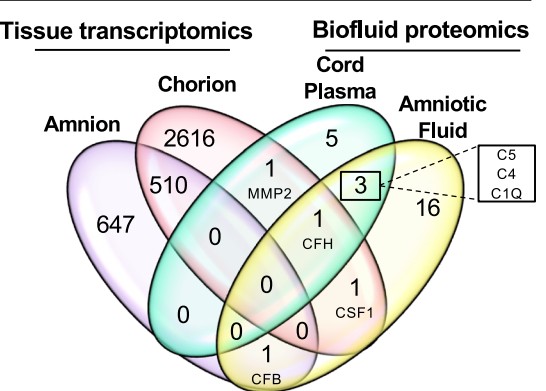

## B Differentially expressed genes/proteins: Downregulated

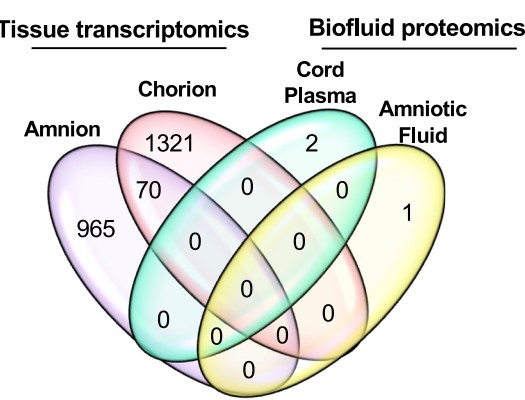

## C Upregulated pathways

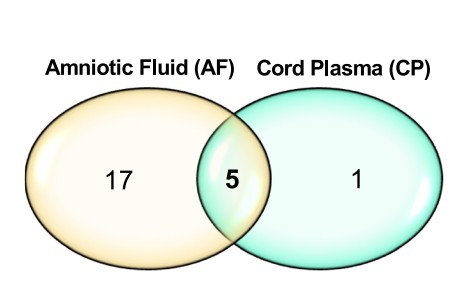

## D Commonly upregulated pathways in AF and CP

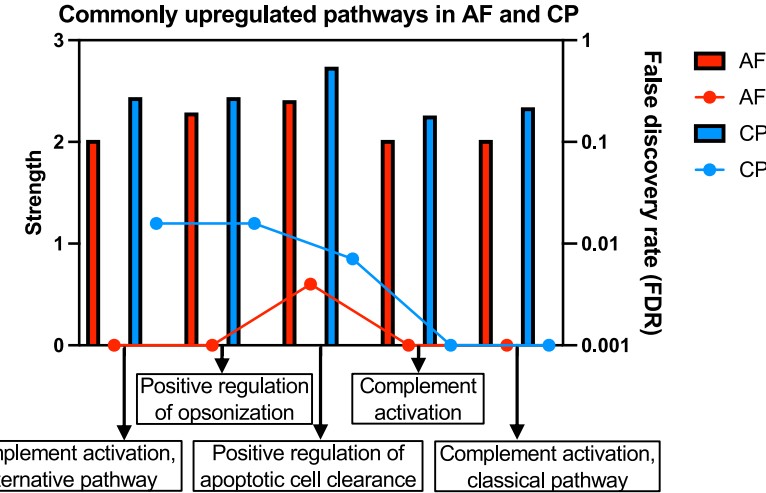

**E**

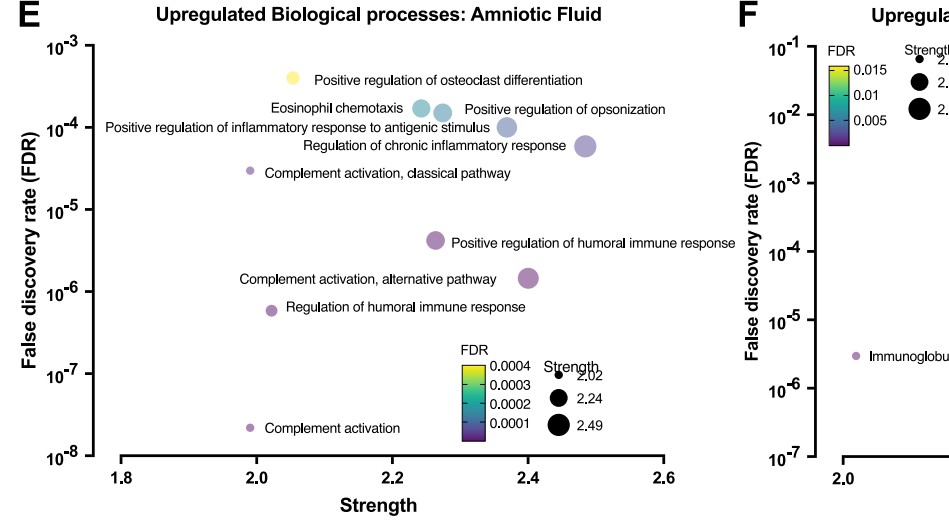

**F**

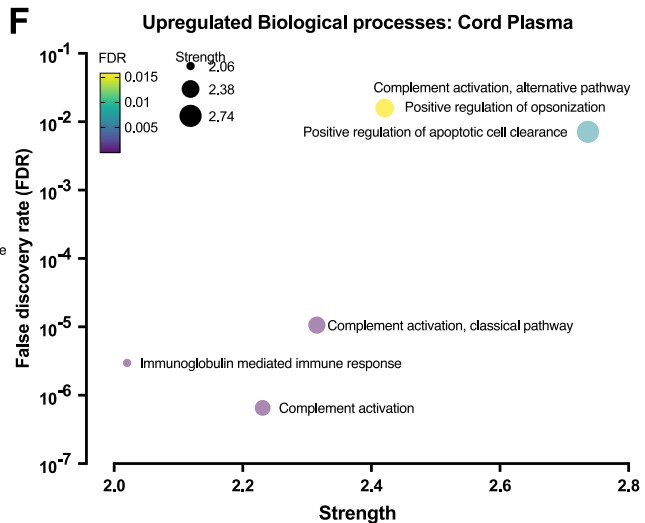

◀  **Figure EV2.  Comparison analysis of placental tissue transcriptomics and biofluids proteomics derived from COVID19-affected pregnancies at delivery.**

Graphical representation of (**A**) upregulated and (**B**) downregulated differentially expressed genes/proteins in tissue transcriptomics (Chorion, $n = 21$ and; Amnion, $n = 18$) and biofluid proteomics (Amniotic fluid, $n = 8$; and Cord plasma, $n = 20$) from COVID-19 affected pregnancies. (**C**) Comparative analysis of upregulated genes/proteins in amniotic fluid and cord plasma from COVID-19-affected pregnancies. (**D**) Commonly upregulated pathways in amniotic fluid and cord plasma from COVID-19-affected pregnancies. (**E, F**) Bubble plots representing upregulated biological process in (**E**) Amniotic fluid and (**F**) cord plasma. Strength and False discovery rate (FDR) were calculated using STRING database pathway analysis. Source data are available online for this figure.

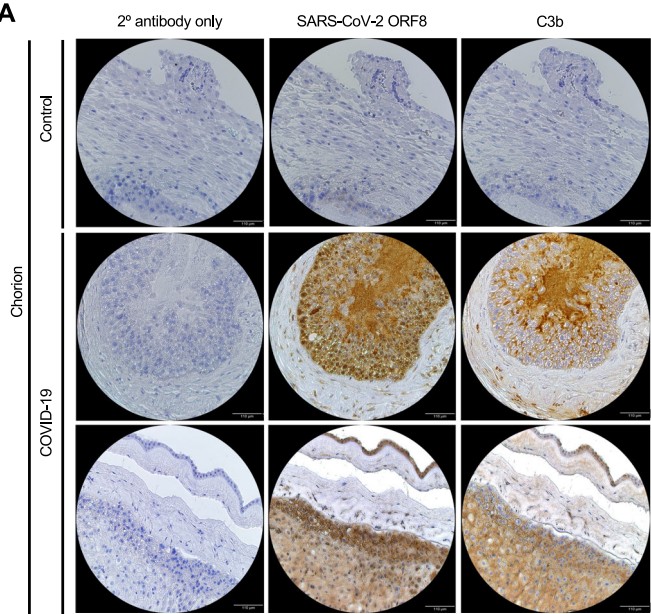

**Figure EV3. Chorion tissues from COVID-19-affected pregnancies exhibited augmented complement activation at ORF8-positive sites.**

(A) Chorion serial sections were analyzed by immunohistochemistry for detection of SARS-CoV-2 ORF8 and C3b. From left to right, tissues stained with (i) Hematoxylin (purple) and the secondary antibody anti-rabbit (Motulsky and Brown, 2006); (ii) Hematoxylin (purple) and anti-ORF8 produced (Motulsky and Brown, 2006); and (iii) Hematoxylin (purple) and anti-C3b (Motulsky and Brown, 2006). Images were taken at 20X magnification. Representative images from two control and two COVID-19 chorion specimens. Source data are available online for this figure.

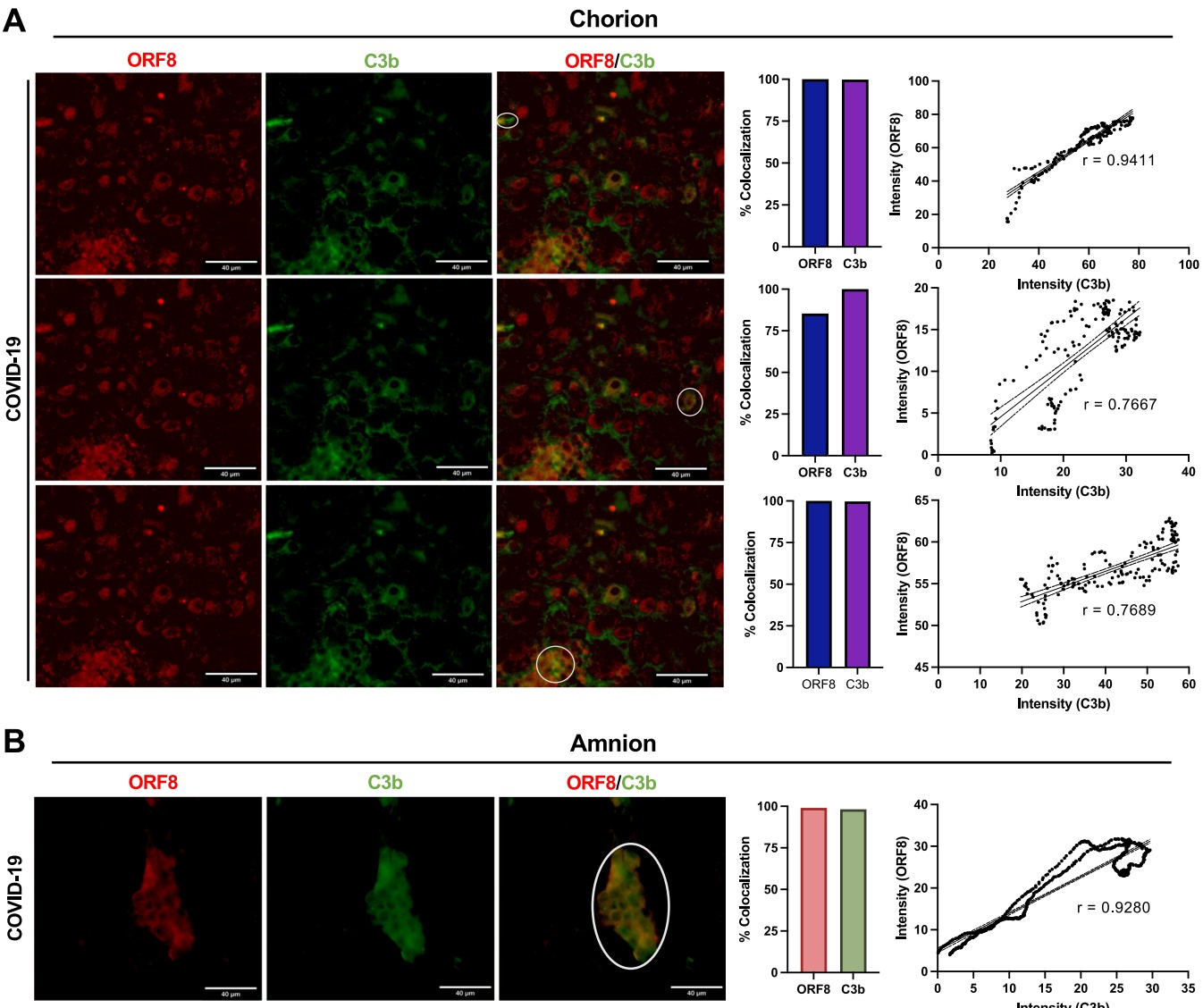

**Figure EV4. Confirmed colocalization of ORF8 and C3b in chorion and amnion tissues from COVID-19-affected pregnancies by fluorescence signals.**

(**A**) Chorion and (**B**) amnion serial sections were stained with SARS-CoV-2 ORF8 (red) and C3b (green). Mander's and Pearson's correlations coefficient between SARS-CoV-2 ORF8 and C3b. White circles represent the colocalization areas analyzed. Images were taken at 40x magnification. Images are representative of one COVID-19+ pregnancy. Scale bars: 40 μm. Source data are available online for this figure.

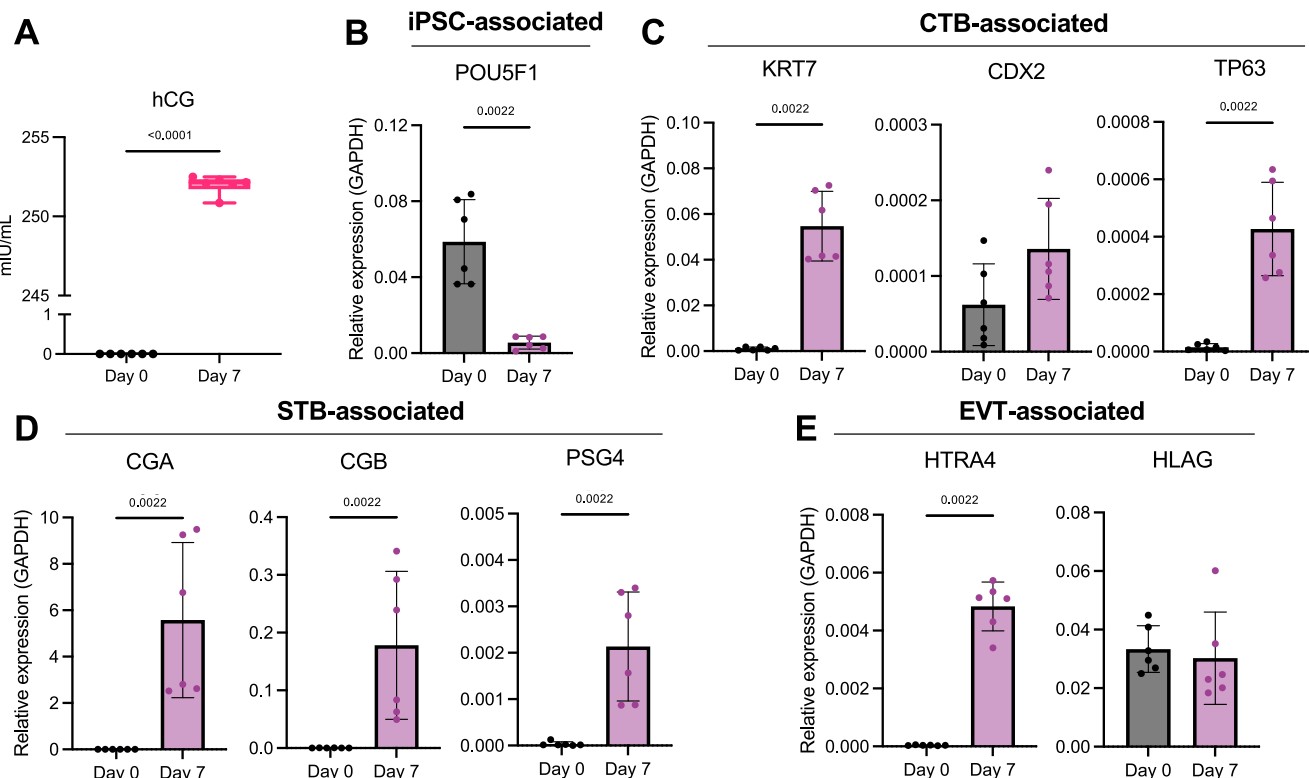

**Figure EV5. Phenotypic characterization of trophoblast delivered from iPSCs.**

(A) Boxplot showing the levels of human chorionic gonadotropin (hCG) in the supernatant of iPSC-derived trophoblasts cultures at day 0 and day 7 of differentiation. Data are presented in boxplot, middle line represents means, the bound of box represent interquartile range, and whiskers represent maximum and minimum values in mUI/mL from six independent experiments ($n = 6$). (B–E) Bar plots representing expression of genes associated as phenotypic markers in the iPSC-derived trophoblasts cultures ($n = 6$) at day 0 and day 7 of differentiation. (B) iPSC-associated transcripts—*POU5F1*, (C) CTB-associated transcripts—*KRT7*, *CDX2*, and *TP63*, (D) STB-associated transcripts—*CGA*, *CGB*, and *PSG4*, and (E) EVT-associated transcripts—*HTRA4*, and *HLAG*. Data are presented in scatter dot plot, line represent means, and whiskers standard derivation. Relative expression was calculated by DCt method using *GAPDH* as the normalizing gene. Data are presented as means ± SDs. Immortalized pluripotent stem cells (iPSC). Cytotrophoblast (CTB). Syncytiotrophoblasts (STB). Extravillous trophoblasts (EVT). POU Domain, Class 5, Transcription Factor 1 (*POU5F1*). Kerantin 7 (*KRT7*). Caudal Type Homeobox 2 (*CDX2*), Tumor Protein P63 (*TP63*). Glycoprotein Hormones, Alpha Polypeptide (*CGA*). Chorionic Gonadotropin Subunit Beta (*CGB*). Pregnancy Specific Beta-1-Glycoprotein 4 (*PSG4*). High-Temperature Requirement Factor A4 (*HTRA4*). Major Histocompatibility Complex, Class I, G (*HLAG*). Mann–Whitney U test was used for all analysis ($p < 0.05$). Source data are available online for this figure.

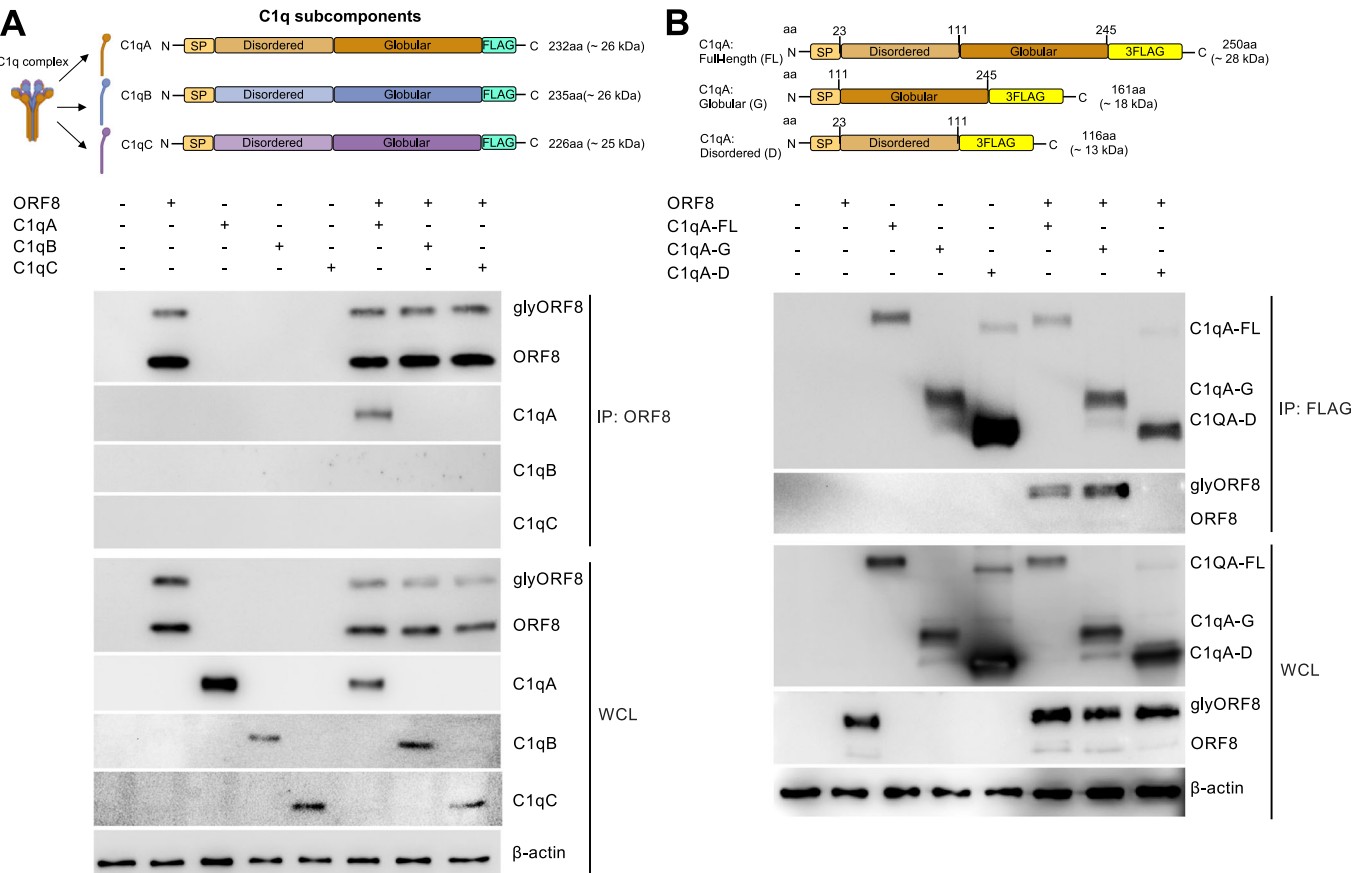

**Figure EV6.  SARS-CoV-2 ORF8 binds to globular domain of complement C1q subcomponent subunit A in HEK 293T cells.**

(A) Co-immunoprecipitation of SARS-CoV-2 ORF8 with C1qA, C1qB, and C1qC. Graphical representation of the plasmid constructs containing the expression cassettes for the subcomponents C1qA, C1qB, and C1qC contained an N-terminal signal peptide (SP), and a C-terminal FLAG tag. Western blot images for co-immunoprecipitation using an anti-ORF8 antibody. (B) Co-immunoprecipitation of C1qA full-length (FL), globular (G), disordered domain (D) with SARS-CoV-2 ORF8. Graphical representation of the plasmid constructs containing expression cassettes for the C1qA-FL, C1qA-G, and C1qA-D, with N-terminal SP and a C-terminal 3FLAG tag, cloned in pIRES vectors. Western blot images for co-immunoprecipitation using an anti-Flag antibody. Co-immunoprecipitated product (IP). Whole-cell lysate (WCL). Source data are available online for this figure.

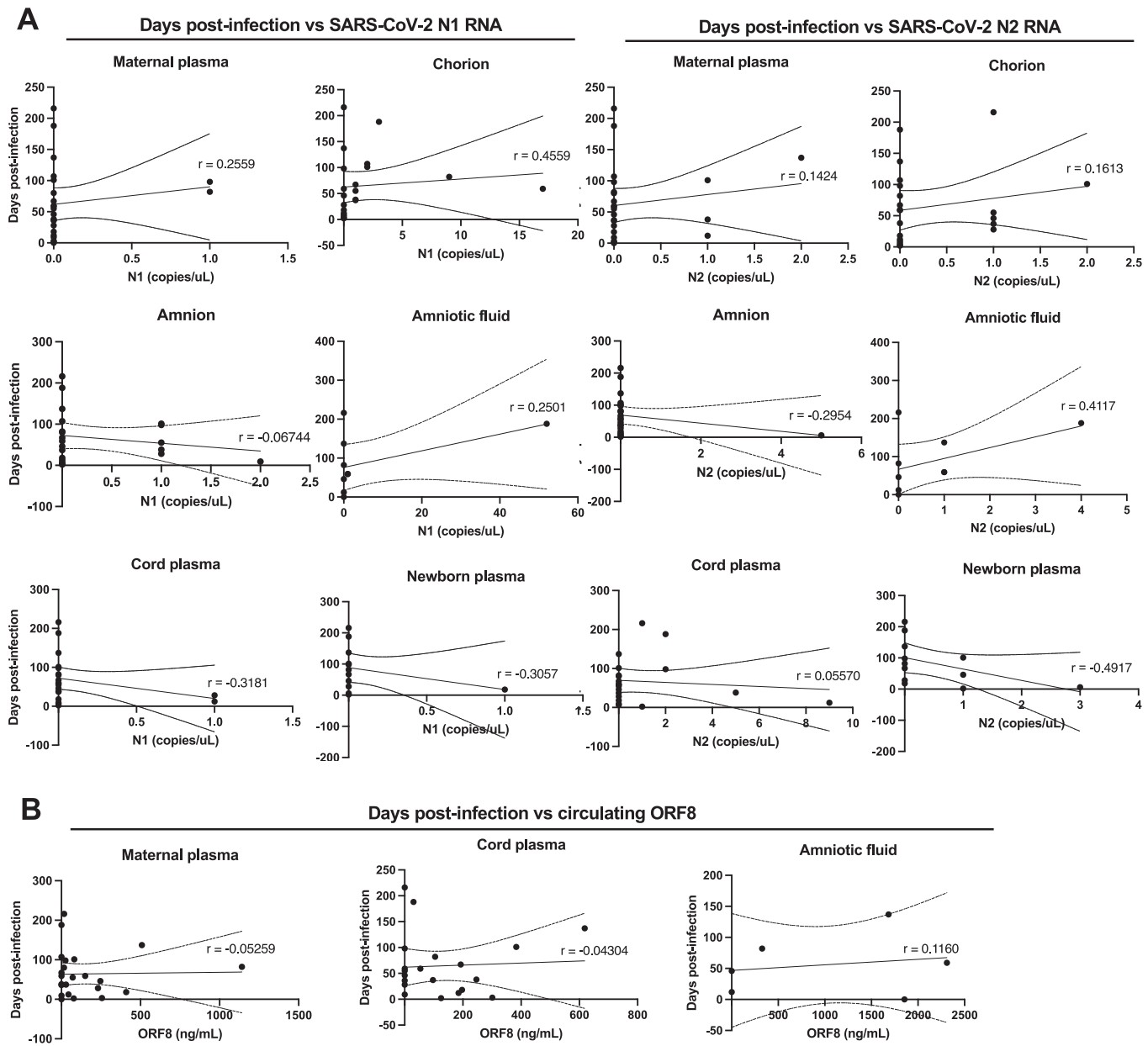

**Figure EV7.  Correlation analyses between the number of days post-infection to delivery and levels of SARS-CoV-2 N1/N2 RNA or circulating ORF8 in COVID-19 pregnancies.**

Correlation analyses between the number of days post SARS-CoV-2 infection to delivery and (**A**) SARS-CoV-2 N1 RNA copies, SARS-CoV-2 N2 RNA copies, and (**B**) circulating ORF8 levels in all biospecimens analyzed, including maternal plasma, chorion, amnion, cord plasma and newborn plasma. Simples linear regression and Spearman's rank correlation test was used for all the correlations analysis. Scatter plots with Pearson correlation coefficients (r) and dotted lines represent 95% confidence intervals. Source data are available online for this figure.

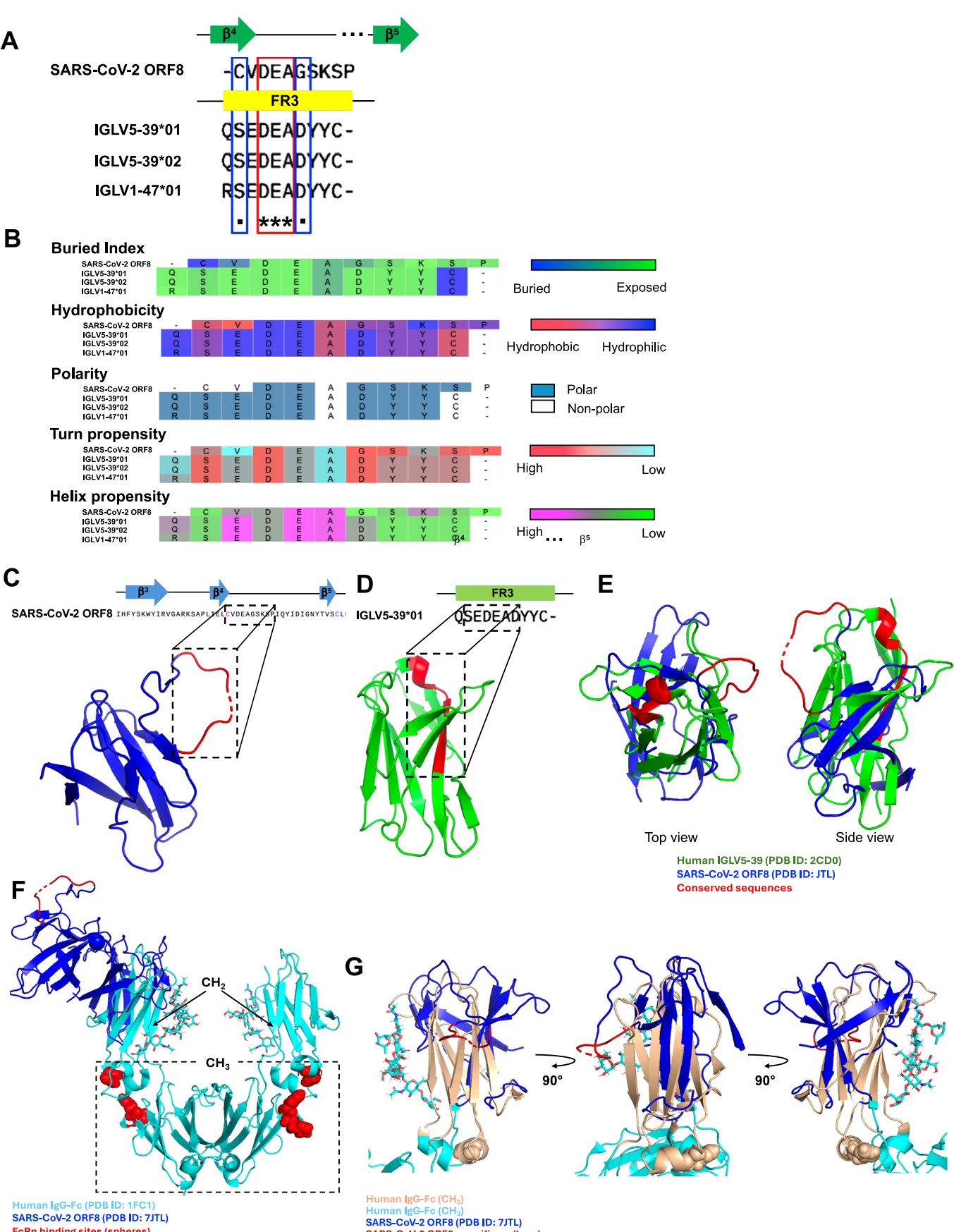

**Figure EV8.　Structural and sequence analysis of SARS-CoV-2 ORF8 and human immunoglobulin genes reveals conservation and functional insights.**

(A) Sequence alignment of SARS-CoV-2 ORF8 and human immunoglobulin variable lambda chain (IGLV) genes. Representative amino acid residues of SARS-CoV-2 ORF8-specific region located at the loop flanking $\beta^4$- $\beta^5$ strands were compared with a portion of the framework 3 (FR3) region of the variable light chain of human IGLV5-39*01, IGLV5-39*02, and IGLV1-47*01 germline gene sequences. Identical residues were highlighted in red box and marked by an asterisk (*), while semi-conserved positions due to minor structural differences are in blue boxes and are indicated by a period (.). The β strands are shown by green arrows above the ORF8 sequence while the spanning loops were represented by black lines (Flower et al, 2021). The FR3 region (shown in the yellow box) was defined according to the IMGT delimitations. (B) Comparison of the overlapping sequences of SARS-CoV-2 ORF8 and human IGLV genes highlighted in the dashed boxes using peptide properties: buried index, hydrophobicity, polarity, and turn and helix propensity. The legend for each property is shown as a gradient color scheme on the right. (C–E) Structures of ORF8 (C), IGLV5-39*01 (D), and superimposed images after pairwise structural alignment as viewed from the top (left) and side (right). PBD entries: 7JTL (SARS-CoV-2, blue) and 2CD0 (human IGLV5-39, green). Overlapping sequences between the two structures are highlighted in red. (F) Pairwise structural alignment of the human IgG-Fc region (cyan, PDB ID:1FC1) and ORF8 (in blue) showing close structural similarity at the CH2 domain (193 atoms aligned; RMSD: 5.9 Å). The CH2 and CH3 domains are highlighted, while residues in red sphere show the neonatal Fc receptor (FcRn) interaction site. (G) Superimposition of CH2 domain of the human IgG-Fc and ORF8 from different rotational perspectives. CH2 residues with the closest folding to ORF8 (blue) is shown in flesh while CH3 are in cyan. The glycosylation site at N297 at the CH2 interface is shown in cyan stick models.

