## [Peer Review File · The EMBO Journal]

Transplacental SARS-CoV-2 protein ORF8 binds to complement C1q to trigger fetal inflammation

Tamiris Azamor, Débora Familiar-Macedo, Gielenny Salem, Chineme Onwubueke, Ivonne Melano, Lu Bian, Zilton Vasconcelos, Karin Saines, Xianfang Wu, Jae Jung, Feng Lin, Oluwatosin Goje, Edward Chien, Steven Gordon, Charles Foster, Hany Aly, Ruth Farrell, Weiqiang Chen, and Suan-Sin Foo

Corresponding author(s): Suan-Sin Foo (foos@ccf.org) , Ruth Farrell (farrelr@ccf.org), Weiqiang Chen (chenw3@ccf.org)

Review Timeline:

Submission Date:	25th Jan 24
Editorial Decision:	13th Mar 24
Revision Received:	16th Jul 24
Editorial Decision:	25th Aug 24
Revision Received:	8th Sep 24
Accepted:	12th Sep 24

Editor: Kelly Anderson

Transaction Report:

Dear Dr. Foo,

Thank you for submitting your manuscript for consideration by the EMBO Journal. It has now been seen by three referees whose comments are shown below.

Given the referees' positive recommendations, I would like to invite you to submit a revised version of the manuscript, addressing the comments of all three reviewers. I should add that it is EMBO Journal policy to allow only a single round of revision, and acceptance of your manuscript will therefore depend on the completeness of your responses in this revised version. It would be good to discuss your plan to address referee concerns and I am available to do so via email or zoom in the coming weeks.

Thank you for the opportunity to consider your work for publication. I look forward to your revision.

Yours sincerely,

Kelly M Anderson, PhD
Editor, The EMBO Journal
k.anderson@embojournal.org

We realize that it is difficult to revise to a specific deadline. In the interest of protecting the conceptual advance provided by the work, we recommend a revision within 3 months (11th Jun 2024). Please discuss the revision progress ahead of this time with the editor if you require more time to complete the revisions. Use the link below to submit your revision:

Referee #1:

Transplacental SARS-CoV-2 ORF8 binds to complement C1q to trigger fetal inflammation

In this new work from Azamor et al. , the authors profiled patient samples from multiple tissues and fluids obtained from pregnancies complicated by SARS-CoV-2 infection for markers of inflammation, complement, and viral infection. Markers were investigated both transcriptionally and at the protein level and provided compelling data to indicate that infection activates complement signaling at the maternal-fetal interface. The authors also provided a plausible mechanism for activation by showing that the viral protein ORF8 from SARS-CoV-2 is capable of binding a functional domain of the complement protein C1q. Viral RNA and ORF8 protein were also detected in multiple tissues and fluids, on both the maternal and fetal side, at frequencies exceeding previously reported rates of vertical transmission of SARS-CoV-2. These results raise important questions regarding vertical transmission and persistence of viral RNA and proteins weeks after active infection has ceased. However, a number of important issues regarding experimental design need to be addressed to support the strong conclusions the authors draw from their results:

Major Comments

1. In Table 1, the authors present limited data regarding the demographics and medical backgrounds of the clinical cohorts. Key details such as racial/ethnic background, maternal age, mode of delivery and the presence of obstetrical and medical conditions including diabetes, obesity, asthma, hypothyroidism, and other concurrent infections are notably absent. Such detailed demographic and clinical information are crucial for identifying potential risk factors and for a more nuanced analysis of the virus impact on pregnancy and fetal health.
2. In table1, the authors should clarify how COVID-19 infection timing within specific trimesters correlates with the observed placental characteristics in the cohort, particularly in relation to the three women who were both COVID-19 positive and diagnosed with preeclampsia? Specifically, during which trimester were these women infected with COVID-19? Additionally, it is noted that placental characteristics such as villitis and chorioamnionitis were reported exclusively in three COVID-19 positive women; were these same women also the ones positive for preeclampsia?
3. In Figure 3, the study illustrates the levels of specific proteins associated with complement activation and proinflammatory cytokines in the amniotic fluid and cord plasma of both healthy individuals and COVID-19-affected groups, employing violin plots for this purpose. These plots represent the overall data distribution. Closer examination of the data specifically in panel D for C1q, panel E for CCL4, CCL13, TNF, IL-1alpha, and panel F for C2 and sCD40L, reveals significant outliers. These outliers appear as distinct and thinner extensions or as isolated data points diverging from the main data cluster, indicating potential anomalies within the datasets. Given the presence of three women with preeclampsia within the COVID-19 infection cohort, it's imperative to conduct a thorough outlier analysis. Such an analysis is crucial not only for understanding the impact of these outliers on the overall findings but also for ensuring the integrity and reliability of the conclusions drawn regarding the influence of SARS-CoV-2 on inflammatory responses in pregnancy.
4. In Figure 4, similar observations were made in panel E for C4a and C4b.
5. In Figure 4, the authors demonstrated the ability of the ORF8 protein to cross the placenta and reach the fetus by detecting ORF8 levels in various fluids and using immunohistochemistry (IHC) to confirm its presence on the fetal side of the placenta. However, this finding prompts several questions.
 - a. While the amniotic fluid exhibited the highest levels of ORF8, the study does not address whether ORF8 is also present in the villous tissue. Given that SARS CoV2 viral infection primarily targets the syncytiotrophoblasts at the maternal-fetal interface, understanding ORF8 distribution within trophoblasts could provide valuable insights into the mechanisms of fetal exposure to the virus.
 - b. Research indicates that SARS-CoV-2 expression in human term placentas is inconsistent or "patchy." Demonstrating the presence of ORF8 within placental villous trophoblasts could clarify whether the virus itself might not be present, yet ORF8 is still capable of reaching these cells.
 - c. Figure 4K illustrates ORF8 colocalizing with C3b via IHC. To enhance clarity and detail regarding this colocalization, performing an immunofluorescence study could offer a more focused view of the interaction between these proteins, providing a clearer picture of their spatial relationship and potential functional implications.
 - d. Related to Figure 4K, and in reference to another study cited by the authors (ref33), which suggests that ORF8 can bind to C3b and inhibit complement activation, thus evading the immune response, it raises the question of how this interaction between

ORF8 and C3b functions within fetal tissues.

e. Only amnion staining for ORF8 is shown-while the studies were performed in trophoblast cells of the placenta. Placental villous and decidual staining should be shown in addition to or instead of amnion staining.

6. The authors detected viral RNA and ORF8 protein in fetal samples at a much higher frequency than what has been previously reported for vertical transmission and claim that their results suggest that previous studies may have underestimated rates of vertical transmission. However, the sensitivity and specificity of the two techniques used to draw these conclusions (ddPCR and ELISA for ORF8) are not provided. Furthermore, neither technique was used on the COVID-negative controls to confirm specificity. To support these conclusions, the authors must:

a. Determine and provide the specificity and sensitivity of the in-house ORF8 ELISA

b. State the sensitivity and specificity of the ddPCR assay used and cited, and explain the selected cut-off of 2 counts/uL for a positive interpretation. Discussion of how this cutoff relates to qPCR Ct cutoffs typically used to determine positivity would also be valuable.

c. Conduct both the ORF8 ELISA and ddPCR assays in the COVID-negative controls to confirm the absence of false positives.

7. All qPCR done in placental tissues was normalized to a single housekeeping gene, GAPDH. Normalization should be conducted in relation to the geometric mean of at least two housekeeping genes, including those that exhibit higher stability in the placenta (i.e. YWHAZ).

8. Despite providing data on the trimester in which infection occurred, the authors did not conduct any comparisons between samples infected in different trimesters. It would be particularly interesting to know whether detection/levels of viral RNA and ORF8 protein in fetal samples is higher when infection occurred closer to delivery and sample collection.

9. The immunoprecipitation experiments showing that ORF8 directly interacts with complement proteins were conducted in HEK293T transfected with both ORF8 and C1q. The mechanistic claim that ORF8 directly interacts with C1q at the maternal-fetal interface would be strengthened if the authors can demonstrate that this interaction occurs in placental cell lines with endogenous expression of C1q.

10. The authors claim that ORF8 is "transplacental" and imply that it can cross the placenta (in the title of the paper). However, no data is provided to indicate that ORF8 is capable of being transported across the placenta. This conclusion is derived from detection of ORF8 protein within the fetal compartment; however, this is misleading as no evidence is provided to support that ORF8 crosses the placenta in the absence of vertical transmission of the virus itself.

11. The authors report a very high incidence of detection of ORF8 protein in maternal and fetal fluids (60-80%), despite lower rates of detection of viral RNA (26%) and sufficiently long periods between time of diagnosis and delivery. Why would ORF8 protein still be present at high levels in the absence of (and weeks after) active infection? What is the half-life for this protein in human serum?

Minor Comments

1. The control group should be referred to as "controls" or "SARS-CoV-2 negative/uninfected" as opposed to "healthy", as any other pregnancy complications were not provided as exclusion criteria. Furthermore, specific exclusion criteria for the human cohort should be provided.

2. Figure 3B: What are the z-scores provided in the heatmap relative to? This should be specified in the figure legend.

3. It should be specified in the text (results and methods) that the "proteomic" approach used to quantify cytokines and chemokines and presented in Figure 3 uses a targeted bead-based multiplexing technology, as "proteomics" is often used to refer to mass spectrometry-based proteomics, specifically.

4. In the results, the bottom of page 9 refers to cord blood plasma data in Figure 3E. This is referencing the incorrect figure; it should be Figure 3F.

5. A description of the location and function of the different tissues examined (i.e. chorion, amnion, etc) and their relevance to fetal defense should be provided in the text.

6. What explanation can the authors provide to explain the inverted dose-response curve in Figure 5 for gene expression after ORF8 treatment?

7. The protocol used for iPSC differentiation into the CTB/STB/EVT phenotype and Figure S2 should be referenced in the results.

Referee #2:

Azamor et al.'s research elucidated the mechanisms underlying the impact of SARS-CoV-2 infection on pregnancy and childbirth. Through transcriptomic and proteomic analyses of placental tissues, cord tissues/plasma, and amniotic fluid from 23 COVID-19 mother-infant pairs, they observed heightened inflammatory responses in both maternal and fetal compartments. Despite the low detection of the SARS-CoV-2 viral genome at the maternal-fetal interface of affected pregnancies, transplacental transfer of SARS-CoV-2 ORF8 proteins from maternal to fetal compartments was identified. Additionally, experimental evidence indicated that ORF8 treatment on placental trophoblasts triggered complement activation and inflammation. This study

addresses a critical medical concern for COVID-19 patients.

I have a few inquiries:

1. According to the two transcriptome analyses (Figures 2 and 3), up-regulation of numerous immune molecules is evident in both samples. Could the authors elucidate the extent of overlap in genes and pathways between these two sets of findings?
2. In recent years, numerous studies on Long-COVID have reported similar observations of upregulated immune systems in affected patients. Could the authors conduct a comparison to identify any shared sets of genes or common pathways?
3. Based on the transcriptomic evidence, one would seek to understand the factors driving the induced expression of immune molecules in patient organs. While the authors investigated the presence of ORF8 protein directly at the maternal-fetal interface, it is possible that other viral proteins may also play a significant role. Is there a systematic method to quantify the presence of other SARS-CoV-2 proteins in local cells?
4. Are there any overlaps or relationships between the differentially expressed genes (DEGs) identified in Figures 2 and 3, and those identified from transcriptomic analysis (Figure 4) of amnion and umbilical cord tissues between ORF8-positive (+) and ORF8-negative (-) COVID-19-affected pregnancies?
5. Similarly, is there a logical approach to screen potential targets of ORF8 in local cells, such as a pull-down experiment followed by mass spectrometry?

Referee #3:

In this Paper by Azamor et al., multi-omics analyses were done on biospecimens obtained from COVID-19 mother and infant pairs. Significant differences were observed in inflammatory responses, expressions of complement proteins and detection of SARS-CoV-2 ORF8 in both maternal and fetal tissues. The authors further suggested a role of ORF8 being responsible for the elevated inflammation and complement activation and uses a series of in vitro and ex vivo experimental systems to demonstrate the relationship. The group finally concluded by identifying the region within ORF8 that could be responsible for binding to C1q leading to eventually activation of the complement pathway. This manuscript is well written up and provides important immunological differences that were detected within the fetal compartment. The only limitation in this manuscript is the inability to perhaps draw links to the findings presented in this study to actual cases of severe pregnancy complications or fetal inflammation that leads to undesirable fetal outcomes. Otherwise, this manuscript present important findings that will bring value to the COVID-19 research community.

1. It would be good to include in Table 1 the information on how long did the infection in the pregnant women took to clear and whether if there is any correlation between the time taken to clear the infection versus the presence of SARS-CoV-2 N1 N1/2 detected in the various biospecimens collected, as well as if there is any correlation versus against the presence of ORF8 detected in the fetal biospecimens
2. It would be good to have in a separate table indicating the number of different samples collected from each maternal-fetal pair and highlight which samples were detected to be positive for M1 or N1+N2 proteins. This table would complement Fig 1B and makes it easier to visualize how the 26% of vertical transmission was obtained.
3. I am wondering how did the authors derived the numbers in Figure 2D? Is it just simple maths by taking the total numbers of DEGS (regardless of up or downregulated) from both Fig 2B and 2C and from there identify the common and exclusive DEGS? If this is how it is done, then the numbers do not add up.
4. In Figure 2H, why are these genes depicted? Are they depicted because these are the genes that are associated with the GO pathways mentioned in 2E, 2F and 2G? If so, this has to be indicated in the text or M&M or legend.
5. In Fig 7, authors identified 4 potential linear epitopes of ORF8 and treatment with the various peptides are able to induce the expression of C1 proteins. However, this is still showing that individually these peptides can induce C1 proteins expression. Structurally, can the authors comment of the proximity between peptides #4, 10 and 25 and discuss the possibility of the how the 3 peptides could potentially work synergistically to drive the expression of Complement proteins?

Minor

1. Consider reformatting the X and Y axes of Fig 1B to make it easier to read.
2. Check through the spelling of the words, there are some mistakes being spotted.
3. Bulk RNAseq was performed on 47 tissue specimens right (Fig 2A)? In the manuscript it was written 74 samples
4. It would be good to be consistent between Fig 2E, 2F and 2G to either indicate or remove the (GO numbers).
5. Please indicate exact P values as per EMBO requirements
6. In Fig 3A, it should be N=3 for healthy amniotic fluid, since in Fig 3B only 3 samples are shown in the heat map. However, in Fig 3E some of the cytokines were detected from n=7 samples? Why were these additional 4 samples not included in Fig 3B heatmap?

7. Please call out figure 3F. Instead of 3F, 3E was called out instead.
8. How many of the DEGS between Amnion and umbilical cord are common? I would assume it to be high, since they both show complement activation pathways?
9. In Fig 4B, for those amniotic fluid that were positive, were their corresponding maternal and cord plasma similarly positive for ORF8?
10. "we compared the transcriptomic profiles of amnion and umbilical cord tissues between the ORF8 positive (+) and ORF8 negative (-) COVID- 19-affected pregnancies. This analysis identified 400 DEGs exclusively affected in ORF8 (+) amnions (Fig. 4C)." This statement is incorrect, given that only the transcriptomics from Amnion was shown.
11. Does the 6/23 maternal-fetal pair that demonstrated vertical transmission typically exhibit higher levels of proinflammatory or complement pathways as compared those that did not exhibit vertical transmission? Also, are these samples all ORF8 positive as well?
12. Spelling error "suggestes" should be suggested
13. Is it possible to comment or discuss on the % similarity between human IgG and immunoglobulin-like domain of ORF8?
14. Although ORF8 is detected in the cord plasma, is it possible to detect for ORF8 from the blood plasma of the newborn?
15. Able to comment if in patients with serious complications during pregnancy leading to perhaps fetal abnormalities or death, the ORF8 levels are significantly elevated as well?
16. For the M&M, is there a study protocol number that should be listed down?

Referee #1

General comments:

In this new work from Azamor et al., the authors profiled patient samples from multiple tissues and fluids obtained from pregnancies complicated by SARS-CoV-2 infection for markers of inflammation, complement, and viral infection. Markers were investigated both transcriptionally and at the protein level and provided compelling data to indicate that infection activates complement signaling at the maternal-fetal interface. The authors also provided a plausible mechanism for activation by showing that the viral protein ORF8 from SARS-CoV-2 is capable of binding a functional domain of the complement protein C1q. Viral RNA and ORF8 protein were also detected in multiple tissues and fluids, on both the maternal and fetal side, at frequencies exceeding previously reported rates of vertical transmission of SARS-CoV-2. These results raise important questions regarding vertical transmission and persistence of viral RNA and proteins weeks after active infection has ceased. However, a number of important issues regarding experimental design need to be addressed to support the strong conclusions the authors draw from their results.

Response: We thank the reviewer for the positive and constructive feedback which has enabled us to improve our manuscript significantly through this revision. We have made a genuine attempt to thoroughly address all concerns that the reviewer raised, as described below.

Major Comments:

1. In Table 1, the authors present limited data regarding the demographics and medical backgrounds of the clinical cohorts. Key details such as racial/ethnic background, maternal age, mode of delivery and the presence of obstetrical and medical conditions-including diabetes, obesity, asthma, hypothyroidism, and other concurrent infections are notably absent. Such detailed demographic and clinical information are crucial for identifying potential risk factors and for a more nuanced analysis of the virus impact on pregnancy and fetal health.

Response: We thank the reviewer for the thoughtful comment. We have now included more demographics and medical backgrounds of the patient cohort in the **revised Table 1**. We also would like to emphasize that this study is focused on elucidating the molecular mechanism driving COVID-19 pregnancy-associated fetal inflammation. Hence, identifying potential risk factors associated with negative impact of COVID-19 pregnancy and fetal health is beyond the scope of the study.

2. In table1, the authors should clarify how COVID-19 infection timing within specific trimesters correlates with the observed placental characteristics in the cohort, particularly in relation to the three women who were both COVID-19 positive and diagnosed with preeclampsia? Specifically, during which trimester were these women infected with COVID-19? Additionally, it is noted that placental characteristics such as villitis and chorioamnionitis were reported exclusively in three COVID-19 positive women; were these same women also the ones positive for preeclampsia?

Response: We thank the reviewer for the thoughtful comments. Among the three women who developed COVID-19 and diagnosed with preeclampsia, one was positive in her 2nd trimester, and two in their 3rd trimesters. The women who were COVID-19 positive and diagnosed with placental pathologies such as villitis and chorioamnionitis were not the same women noted to have preeclampsia. All the additional information is now included in **revised Table 1**.

3. In Figure 3, the study illustrates the levels of specific proteins associated with complement activation and proinflammatory cytokines in the amniotic fluid and cord plasma of both healthy individuals and COVID-19-affected groups, employing violin plots for this purpose. These plots represent the overall data distribution. Closer examination of the data specifically in panel D for C1q, panel E for CCL4, CCL13, TNF, IL-1alpha, and panel F for C2 and sCD40L, reveals significant outliers. These outliers appear as distinct and thinner extensions or as isolated data points diverging from the main data cluster, indicating potential anomalies within the datasets. Given the presence of three women with preeclampsia within the COVID-19 infection cohort, it's imperative to conduct a thorough outlier analysis. Such an analysis is crucial not only for understanding the impact of these outliers on the overall findings but also for ensuring the integrity and reliability of the conclusions drawn regarding the influence of SARS-CoV-2 on inflammatory responses in pregnancy.

Response: We thank the reviewer for this thoughtful comment. As suggested by the reviewer, we have now conducted a thorough outlier analysis. Outliers were identified using the ROUT method [1] with a Q = 1% cutoff. Our results did not change significantly with or without the identified outliers in each figure. Due to the scarcity of these patients' specimens, we decided not to remove the outlier from the original analysis. For a better data presentation, we have now broken the Y-axis of the above-mentioned plots into 2 segments for a clearer view of the distribution of all samples. In addition, we have also included the adjusted plots without the outlier as shown alongside for reference.

4. In Figure 4, similar observations were made in panel E for C4a and C4b.

Response: We thank the reviewer for this thoughtful comment. As suggested by the reviewer, we have now conducted a thorough outlier analysis. Outliers were identified using the ROUT method [1] with a Q = 1% cutoff. Our results did not change significantly with or without the identified outliers in each figure. Due to the scarcity of these patients' specimens, we decided not to remove the outlier from the original analysis. For a better data presentation, we have now broken the Y-axis of the above-mentioned plots into 2 segments for a clearer view of the distribution of all samples in the **new Figure 4G**. In addition, we have also included the adjusted plots without the outlier as shown alongside for reference.

5. In Figure 4, the authors demonstrated the ability of the ORF8 protein to cross the placenta and reach the fetus by detecting ORF8 levels in various fluids and using immunohistochemistry (IHC) to confirm its presence on the fetal side of the placenta. However, this finding prompts several questions.

(a) While the amniotic fluid exhibited the highest levels of ORF8, the study does not address whether ORF8 is also present in the villous tissue. Given that SARS CoV-2 viral infection primarily targets the syncytiotrophoblasts at the maternal-fetal interface, understanding ORF8 distribution within trophoblasts could provide valuable insights into the mechanisms of fetal exposure to the virus.

(b) Research indicates that SARS-CoV-2 expression in human term placentas is inconsistent or "patchy." Demonstrating the presence of ORF8 within placental villous trophoblasts could clarify whether the virus itself might not be present, yet ORF8 is still capable of reaching these cells.

Response: We thank the reviewer for the critical comments. To demonstrate ORF8 distribution in the placental trophoblasts, we have performed immunofluorescence staining on the placental chorion and amnion tissues using antibodies against ORF8 and trophoblast marker – cytokeratin 8/18 (Krt8/18), which is found on syncytiotrophoblasts, villous trophoblasts in the fetus side (amnion) and extravillous trophoblasts in the maternal side (chorion). Specifically, our results demonstrated the presence of ORF8 within the extravillous trophoblasts in maternal chorion tissue. In the fetal amnion tissue, ORF8 is located

at close proximity to syncytiotrophoblasts and villous trophoblasts, but not within. We have now included this data as a **new Figure 5** of the revised manuscript.

(c) Figure 4K illustrates ORF8 colocalizing with C3b via IHC. To enhance clarity and detail regarding this colocalization, performing an immunofluorescence study could offer a more focused view of the interaction between these proteins, providing a clearer picture of their spatial relationship and potential functional implications.

Response: To demonstrate a clearer view of the spatial relationship between ORF8 and C3b in the placenta, we have performed immunofluorescence staining on the placental chorion and amnion tissues using antibodies against ORF8 and C3b. We were able to confirm the co-localization of ORF8 and C3b as observed in **Figure 4M**, using the immunofluorescence staining method. We have now included this data as a **new Figure 5** of the revised manuscript.

(d) Related to Figure 4K, and in reference to another study cited by the authors (ref33), which suggests that ORF8 can bind to C3b and inhibit complement activation, thus evading the immune response, it raises the question of how this interaction between ORF8 and C3b functions within fetal tissues.

Response: This is a really interesting point. Complement cascade is comprised of three activation pathways: (1) Classical pathway (by C1q), (2) Alternative pathway (by C3) and Lectin pathway. Indeed, Kumar *et al.* demonstrated that ORF8 can bind to C3b, which leads to two outcomes: (1) ORF8 binding to C3b prevented Complement Factor I (CFI) from proteolytic cleavage of activated form of C3 – C3b, which can lead to accumulation of C3b, thus hyperactivation of complement cascade; and (2) binding of ORF8 to C3b impeded Complement Factor B (CFB), a component of alternative pathway, from binding to C3b, which can prevent the formation of alternative pathway C3 convertase (C3bBb). Putting these 2 observations together, ORF8 binding to C3b can prevent alternative pathway activation, but may drive hyperactivation of complement cascade through other complement pathways, such as classical pathway. Indeed, our study demonstrated the importance of ORF8-C1q-mediated classical pathway activation in contributing to fetal inflammation. Our findings of (1) abundance co-localization of ORF8 and C3b staining in COVID-19 placental tissues (**new Figures 4M, 5, EV2**), and (2) increase downstream complement factor 5 (C5) detected in amniotic fluid and cord plasma of COVID-19 pregnancies (**Figures 3D and 3F**), further confirmed the hyperactivation of complement cascade during COVID-19 pregnancies. We have now included this in the **Discussion** section of the revised manuscript. On a side note, as requested by Reviewer 2, we performed mass spectrometry on ORF8-overexpressing trophoblasts for unbiased identification of other potential protein partners of ORF8. Indeed, we also identified C3 as one of the potential binding proteins to ORF8.

(e) Only amnion staining for ORF8 is shown-while the studies were performed in trophoblast cells of the placenta. Placental villous and decidual staining should be shown in addition to or instead of amnion staining.

Response: We have performed additional IHC and immunofluorescence staining of ORF8 and C3b in both maternal-skewed chorion/choriodecidua and fetal-skewed amnion tissues. These results are now presented as **new Figure EV2** and **new Figure 5** of the revised manuscript.

6. The authors detected viral RNA and ORF8 protein in fetal samples at a much higher frequency than what has been previously reported for vertical transmission and claim that their results suggest that previous studies may have underestimated rates of vertical transmission. However, the sensitivity and specificity of the two techniques used to draw these conclusions (ddPCR and ELISA for ORF8) are not provided. Furthermore, neither technique was used on the COVID-negative controls to confirm specificity. To support these conclusions, the authors must:

a. Determine and provide the specificity and sensitivity of the in-house ORF8 ELISA

Response: As suggested by the reviewer, we have calculated Specificity and Sensitivity based on the following method [2]:

	Disease present	Disease absent
Test Positive	a (TP)	b (FP)
Test Negative	c (FN)	d (TN)

TP: True positive
 FP: False positive
 FN: False negative
 FP: False positive

Sensitivity = $a / (a+c) * 100$

= a (true positive) / a+c (true positive + false negative)

Specificity = $d / (b+d) * 100$

= d (true negative) / b+d (true negative + false positive)

For the in-house ORF8 ELISA, the details are as follows:

- Maternal plasma**

Maternal plasma	Disease present	Disease absent
Test Positive	15	0
Test Negative	8	4

Control (N=4); COVID-19 (N=23)

Sensitivity = $15 / (15+8) * 100 = 65.2\%$

Specificity = $4 / (0+4) * 100 = 100\%$

- Amniotic fluid**

Amniotic fluid	Disease present	Disease absent
Test Positive	5	0
Test Negative	3	3

Control (N=3); COVID-19 (N=8)

Sensitivity = $5 / (5+3) * 100 = 65.2\%$

Specificity = $3 / (0+3) * 100 = 100\%$

- Cord plasma**

Cord plasma	Disease present	Disease absent
Test Positive	12	0
Test Negative	8	4

Control (N=4); COVID-19 (N=20)

Sensitivity = $12 / (12+8) * 100 = 60\%$

Specificity = $4 / (0+4) * 100 = 100\%$

Therefore, the observed sensitivity and specificity for our in-house ORF8 ELISA range from 60-65.2% and 100%, respectively. We have now included this information in the **Methods** section of "SARS-CoV-2 ORF8 ELISA".

b. State the sensitivity and specificity of the ddPCR assay used and cited, and explain the selected cut-off of 2 counts/uL for a positive interpretation. Discussion of how this cutoff relates to qPCR Ct cutoffs typically used to determine positivity would also be valuable.

Response: We thank the reviewer for the suggestion. The cut-off of 2 counts/ul was determined by the manufacturer of the commercial SARS-CoV-2 ddPCR kit (Bio-Rad, Cat. no. 12013743) used in our study. It is noteworthy that ddPCR offers an absolute quantitation of amplicons, while qPCR only offers relative quantitation. In addition, the mechanism of quantitation between qPCR and ddPCR are different which makes it impossible to relate qPCR Ct cutoffs to ddPCR absolute counts. In fact, Li *et al.* had conducted a correlation of output values in samples from hospitalized COVID-19 patients, demonstrating a strong positive correlation among them [3]. To clarify the sensitivity of ddPCR technique, we have included the following description in the **Methods** section: “For all ddPCR detection of SARS-CoV-2 nucleocapsid (N) 1, N2 genes and human RRP30 gene, 100 ng of extracted total RNA or viral RNA were used for the tissue and biofluid specimens, respectively. The cutoff of 2 counts for at least one viral target (N1 or N2), determined by the manufacturer, was used to classify samples as SARS-CoV-2 negative and positive.”

c. Conduct both the ORF8 ELISA and ddPCR assays in the COVID-negative controls to confirm the absence of false positives.

Response: We agree with the reviewer that COVID-negative controls should be included in both ORF8 ELISA and ddPCR assays to confirm the accuracy of these assays. We have included the COVID-19 negative controls in **revised Figures 1B and 4A**. In addition, we have also calculated the sensitivity and specificity of ORF8 ELISA as suggested in critique 6(a) and added to the **Methods** section of “SARS-CoV-2 ORF8 ELISA”.

7. All qPCR done in placental tissues was normalized to a single housekeeping gene, GAPDH. Normalization should be conducted in relation to the geometric mean of at least two housekeeping genes, including those that exhibit higher stability in the placenta (i.e. YWHAZ).

Response: In this study, we performed qPCR on placental trophoblast cells, using GAPDH as the housekeeping gene for qPCR analysis of trophoblast cells, which exhibited high stability in transcriptional expression. We have confirmed that across all qPCR performed, Ct values of the housekeeping gene were very consistent across all sample types. Specifically, the GAPDH Ct values for: HTR8 trophoblast cell line has an average of 17.33 (SD \pm 0.18), primary villous trophoblasts have an average of 17.36 (SD \pm 0.27), and iPSC-derived trophoblasts have an average of 18.2 (SD \pm 0.17).

8. Despite providing data on the trimester in which infection occurred, the authors did not conduct any comparisons between samples infected in different trimesters. It would be particularly interesting to know whether detection/levels of viral RNA and ORF8 protein in fetal samples is higher when infection occurred closer to delivery and sample collection.

Response: Due to the small sample number in first trimester (N=2), we were unable to perform statistical analysis among trimesters. However, we agree with the reviewer that it will be insightful to show the detection/levels of viral RNA and ORF8 protein in fetal compartment based on trimesters of infection. Therefore, as suggested by reviewer, we added **new Figures 1D and 4C** to demonstrate the trend of viral RNA and ORF8 levels, respectively, in different trimesters.

9. The immunoprecipitation experiments showing that ORF8 directly interacts with complement proteins were conducted in HEK293T transfected with both ORF8 and C1q. The mechanistic claim that ORF8 directly interacts with C1q at the maternal-fetal interface would be strengthened if the authors can demonstrate that this interaction occurs in placental cell lines with endogenous expression of C1q.

Response: Expression of complement factors are tightly regulated during pregnancy. While C1q has been shown to play an indispensable role at the maternal-fetal interface, it is typically expressed at low levels in the placenta. As shown in the Human Protein Atlas for C1QA, IHC staining of C1qA in the

placenta demonstrated low levels of C1qA expression in trophoblasts [4]. Indeed, we have attempted to perform pulldown of endogenous level of C1qA in HTR8 trophoblast cell line, but were unsuccessful, largely due to low endogenous level of C1qA. Nevertheless, to address reviewer's comment on demonstrating the interaction of ORF8 and C1qA in the placental cells, we performed co-immunoprecipitation (co-IP) of ORF8 and C1qA in HTR8 trophoblast cell line, now presented as **new Figure 7**. We were able to confirm the specific interaction between ORF8 and C1qA as previously shown in co-IP assay using HEK293T cells (now presented as **Figure EV4**).

10. The authors claim that ORF8 is "transplacental" and imply that it can cross the placenta (in the title of the paper). However, no data is provided to indicate that ORF8 is capable of being transported across the placenta. This conclusion is derived from detection of ORF8 protein within the fetal compartment; however, this is misleading as no evidence is provided to support that ORF8 crosses the placenta in the absence of vertical transmission of the virus itself.

Response: To confirm that the detection of ORF8 at fetal side is not due to vertical transmission of SARS-CoV-2, we have quantified the levels of anti-SARS-CoV-2 Nucleocapsid IgG and anti-SARS-CoV-2 Spike S1 IgM in maternal plasma and cord/newborn plasma specimens. During pregnancy, maternal IgG can be passively transferred across placenta but not IgM. Hence, a positive detection of SARS-CoV-2 IgM in the cord/newborn plasma would be indicative of an active SARS-CoV-2 infection. Indeed, our results show that all the newborns were negative for anti-SARS-CoV-2 S1 IgM, demonstrating the absence of vertical transmission of SARS-CoV-2 infection. These findings are now presented as **new Figure 4D** of the revised manuscript.

11. The authors report a very high incidence of detection of ORF8 protein in maternal and fetal fluids (60-80%), despite lower rates of detection of viral RNA (26%) and sufficiently long periods between time of diagnosis and delivery. Why would ORF8 protein still be present at high levels in the absence of (and weeks after) active infection? What is the half-life for this protein in human serum?

Response: We thank the reviewer for the insightful comment. To date, neither the half-life of ORF8 nor duration of circulating ORF8 after SARS-CoV-2 infection has been studied. In our study, we were able to detect circulating ORF8 in cord plasma/amniotic fluid up to 188 days or up to 216 days in maternal plasma after SARS-CoV-2 infection. Our results indicates that ORF8 can be stably circulated during pregnancy for several months post infection. This data is now presented as **new Table EV2** of the revised manuscript.

Minor Comments:

1. The control group should be referred to as "controls" or "SARS-CoV-2 negative/uninfected" as opposed to "healthy", as any other pregnancy complications were not provided as exclusion criteria. Furthermore, specific exclusion criteria for the human cohort should be provided.

Response: As requested by the reviewer, we have now edited the term "healthy" to "control" throughout the manuscript. As this study focused on the recruitment of pregnant women with or without COVID-19, no specific exclusion criteria was included.

2. Figure 3B: What are the z-scores provided in the heatmap relative to? This should be specified in the figure legend.

Response: We apologize for the mistake. The "z-scores" is in fact "Fold change" calculated by "Conc. of cytokine x in Sample / Average (Conc. of protein conc. of cytokine x in all Control samples)". We have

now corrected the label in **Figure 3B** and included the description of fold change calculation in the **Methods** section.

3. It should be specified in the text (results and methods) that the "proteomic" approach used to quantify cytokines and chemokines and presented in Figure 3 uses a targeted bead-based multiplexing technology, as "proteomics" is often used to refer to mass spectrometry-based proteomics, specifically.

Response: As suggested by the reviewer, we have modified the Results and Methods sections to reflect "targeted bead-based multiplexing technology".

4. In the results, the bottom of page 9 refers to cord blood plasma data in Figure 3E. This is referencing the incorrect figure; it should be Figure 3F.

Response: We apologize for the mistake and have corrected it in the revised manuscript.

5. A description of the location and function of the different tissues examined (i.e. chorion, amnion, etc) and their relevance to fetal defense should be provided in the text.

Response: We thank the reviewer for the thoughtful comment. We have now added a brief description of the location and immunological function of different placental tissues in the Introduction section of the revised manuscript.

6. What explanation can the authors provide to explain the inverted dose-response curve in Figure 5 for gene expression after ORF8 treatment?

Response: ORF8 is secreted from SARS-CoV-2 infected cells as homodimers, triggering host immune response. However, apart from being a homodimer, SARS-CoV-2 ORF8 is also able to form higher order functional multimers, such as tetramers and trimers [5]. We hypothesized that at low concentration of 10ng/ml, it may promote presence of ORF8 monomer, while high concentration of 100ng/ml may promote the formation of ORF8 oligomers, thereby impacting on the interaction with C1q. We have now included this in the **Results** section of the revised manuscript.

7. The protocol used for iPSC differentiation into the CTB/STB/EVT phenotype and Figure S2 should be referenced in the results.

Response: The new supplementary **Figure EV3** is now referenced in the **Results** section, and iPSC differentiation of CTB/STB/EVT protocol are referenced in the **Methods** section, "iPSC-derived trophoblasts" section.

Referee #2

General comments:

Azamor et al.'s research elucidated the mechanisms underlying the impact of SARS-CoV-2 infection on pregnancy and childbirth. Through transcriptomic and proteomic analyses of placental tissues, cord tissues/plasma, and amniotic fluid from 23 COVID-19 mother-infant pairs, they observed heightened inflammatory responses in both maternal and fetal compartments. Despite the low detection of the SARS-CoV-2 viral genome at the maternal-fetal interface of affected pregnancies, transplacental transfer of SARS-CoV-2 ORF8 proteins from maternal to fetal compartments was identified. Additionally,

experimental evidence indicated that ORF8 treatment on placental trophoblasts triggered complement activation and inflammation. This study addresses a critical medical concern for COVID-19 patients.

Response: We thank the reviewer for the positive and constructive feedback which has enabled us to improve our manuscript significantly through this revision. We have made a genuine attempt to thoroughly address all concerns that the reviewer raised, as described below.

Major comments:

1. According to the two transcriptome analyses (Figures 2 and 3), up-regulation of numerous immune molecules is evident in both samples. Could the authors elucidate the extent of overlap in genes and pathways between these two sets of findings?

Response: We thank the reviewer for the critical comment. We would like to clarify that the findings presented in **Figure 2** is RNAseq transcriptomics analysis of the chorion and amnion tissues, while **Figure 3** demonstrates the bead-based multiplexing proteomics analysis of amniotic fluid and cord plasma. Due to the vast difference in magnitude of genes (>22,000 genes) and proteins (97 cytokines) analyzed, it was difficult to compare the overlapping genes and pathways in these two sets of analyses.

2. In recent years, numerous studies on Long-COVID have reported similar observations of upregulated immune systems in affected patients. Could the authors conduct a comparison to identify any shared sets of genes or common pathways?

Response: Center for disease control and prevention [6] defined Long COVID as a chronic condition that occurs after SARS-CoV-2 infection and is present for at least 3 months. The patients can experience a wide range of ongoing symptoms and conditions that can last weeks, months, or even years after COVID-19 illness [6]. Cervia-Hasler *et. al* showed in a longitudinal analysis an increase of complement system activation driven by antigen–antibody complexes, involving autoantibodies and antibodies against herpesviruses [7]. Although Long COVID seems to present similar complement activation as we show in our manuscript, unfortunately, our results cannot be compared to long COVID studies. First, pregnancy is a unique state where the maternal immune system is characterized by a modulate network of recognition, trafficking, repair and protection even at the time of delivery. The pregnant immune system interacts with infectious agents differently than the non-pregnant immune system maintaining some sensitivity to pathogens [8]. The differential response to infection in pregnant patients can put them at increased risk for adverse effects from certain infections, with obvious risks to the fetus. Even if the mother is asymptomatic or mildly symptomatic during a viral infection, the fetus may still be negatively impacted. Taken together, it would be really complicated to compare results from pregnancy to data analyses in general population that includes men and women. Second, all the women and the babies at the time of collection didn't present any symptoms that could characterize them as Long COVID.

3. Based on the transcriptomic evidence, one would seek to understand the factors driving the induced expression of immune molecules in patient organs. While the authors investigated the presence of ORF8 protein directly at the maternal-fetal interface, it is possible that other viral proteins may also play a significant role. Is there a systematic method to quantify the presence of other SARS-CoV-2 proteins in local cells?

Response: We thank the reviewer for this insightful comment. Our study is indeed the first to demonstrate the presence of circulating ORF8 in COVID-19 pregnancies, and demonstrated the role of ORF8 in complement activation at the maternal-fetal interface. While we agree with the reviewer that it will be interesting to systematically evaluate the possible presence of other SARS-CoV-2 viral proteins, this is

beyond the scope of this study. We will certainly explore the possibility to evaluate other SARS-CoV-2 viral proteins in our pregnant cohort in future studies.

4. Are there any overlaps or relationships between the differentially expressed genes (DEGs) identified in Figures 2 and 3, and those identified from transcriptomic analysis (Figure 4) of amnion and umbilical cord tissues between ORF8-positive (+) and ORF8-negative [7] COVID-19-affected pregnancies?

Authors: We would like to clarify that the findings presented in **Figure 2** is RNAseq transcriptomics analysis of the chorion and amnion tissues, while **Figure 3** demonstrates the bead-based multiplexing proteomics analysis of amniotic fluid and cord plasma. Due to the vast difference in magnitude of genes (>22,000 genes) and proteins (97 cytokines) analyzed, it was difficult to compare the overlapping genes/cytokines between **Figures 2 and 3**. Also, since **Figure 4** is a ORF8-/+ subset reanalysis of COVID-19 pregnancies shown in **Figure 2**, in which we compared the transcriptomics profiles of amnion and umbilical cord tissues obtained from ORF8[7] and ORF8 (+) COVID-19 pregnancies, to identify the immunological impact ORF8 presence on the maternal-fetal interface. Therefore, it will not be feasible to perform overlap analysis across the findings from these three figures.

5. Similarly, is there a logical approach to screen potential targets of ORF8 in local cells, such as a pull-down experiment followed by mass spectrometry?

Response: We thank the reviewer for this suggestion. We agree with the reviewer that including a logical approach to unbiasedly screen for other potential targets of ORF8 in local cells will be insightful. Hence, to address the reviewer's concern, we have performed a pull-down of ORF8 in ORF8-overexpressing HTR8 placental trophoblast cell line. Comparing between the vector- and ORF8-expressing pull-down, we were able to identify two specific bands that were present in ORF8-expressing pull-down, but not in vector control. While mass spectrometry of the specific bands identified numerous potential protein targets, we found that another complement factor, Complement component 3 (C3) which plays a central role in all three complement pathways were found to be interacting with ORF8. We have included this in the **Results section** of the revised manuscript.

Referee #3

General comments:

In this Paper by Azamor et al., multi-omics analyses were done on biospecimens obtained from COVID-19 mother and infant pairs. Significant differences were observed in inflammatory responses, expressions of complement proteins and detection of SARS-CoV-2 ORF8 in both maternal and fetal tissues. The authors further suggested a role of ORF8 being responsible for the elevated inflammation and complement activation and uses a series of in vitro and ex vivo experimental systems to demonstrate the relationship. The group finally concluded by identifying the region within ORF8 that could be responsible for binding to C1q leading to eventually activation of the complement pathway. This manuscript is well written up and provides important immunological differences that were detected within the fetal compartment. The only limitation in this manuscript is the inability to perhaps draw links to the findings presented in this study to actual cases of severe pregnancy complications or fetal inflammation that leads to undesirable fetal outcomes. Otherwise, this manuscript present important findings that will bring value to the COVID-19 research community.

Response: We thank the reviewer for the positive and constructive feedback which has enabled us to improve our manuscript significantly through this revision. We agree with the reviewer that it will be insightful to draw links of our present study to actual cases of severe pregnancy complications and fetal inflammation. While we were unable to recruit pregnant COVID-19 patients with pregnancy complications, our study highlighted the long-term circulation of ORF8 in >60% COVID-19 pregnancies, triggering complement activation and fetal inflammation. Given the fact that majority of the COVID-19 pregnancies were not affected by severe pregnancy complications, we hope that our current cohort can represent the vast majority of COVID-19 pregnancies. Our findings highlighted the importance to study the maternal-fetal immune crosstalk even in seemingly healthy pregnancy outcomes in SARS-CoV-2-infected mothers with prenatal COVID-19. Nevertheless, we have made a genuine attempt to thoroughly address all concerns that the reviewer raised, as described below.

Major comments:

1. It would be good to include in Table 1 the information on how long did the infection in the pregnant women took to clear and whether if there is any correlation between the time taken to clear the infection versus the presence of SARS-CoV-2 N1 N1/2 detected in the various biospecimens collected, as well as if there is any correlation versus against the presence of ORF8 detected in the fetal biospecimens

Response: We thank the reviewer for the comment. To address the reviewer's concern, we have now performed correlation analyses between the number of days post SARS-CoV-2 infection to delivery and (i) SARS-CoV-2 N1 RNA copies, (ii) SARS-CoV-2 N2 RNA copies, and (iii) circulating ORF8 levels in all biospecimens analyzed, including maternal plasma, chorion, amnion, cord plasma and newborn plasma. Specifically, we only found low positive correlation between days post infection and SARS-CoV-2 N1/N2 RNA in chorion, and amniotic fluid. No or negative correlation were observed in all other analyses including circulating ORF8 levels. This finding is now presented as supplementary **Figure EV5** in the revised manuscript. In addition, as suggested by the reviewer, we also included a new supplementary **Table EV2** which outlines the average number of weeks post COVID-19 diagnosis till the time of delivery for positive detection of SARS-CoV-2 N1/N2 and circulating ORF8 in the maternal and fetal compartments.

2. It would be good to have in a separate table indicating the number of different samples collected from each maternal-fetal pair and highlight which samples were detected to be positive for M1 or N1+N2 proteins. This table would complement Fig 1B and makes it easier to visualize how the 26% of vertical transmission was obtained.

Response: We agree with the reviewer. We have included a new supplementary **Table EV1** in the revised manuscript which provides information on the detection of SARS-CoV-2 N1/N2 RNA and circulating ORF8 for each individual maternal-fetal pair of biospecimens evaluated.

3. I am wondering how did the authors derived the numbers in Figure 2D? Is it just simple maths by taking the total numbers of DEGS (regardless of up or downregulated) from both Fig 2B and 2C and from there identify the common and exclusive DEGS? If this is how it is done, then the numbers do not add up.

Response: We thank the reviewer for spotting this and we sincerely apologize for the mistake. We have corrected the numbers in **Figure 2D** and modified the results section to reflect the changes.

4. In Figure 2H, why are these genes depicted? Are they depicted because these are the genes that are associated with the GO pathways mentioned in 2E, 2F and 2G? If so, this has to be indicated in the text or M&M or legend.

Response: We thank the reviewer for the thoughtful comment. These genes are indeed associated with the pathways analyzed in **Figure 2E, 2F** and **2G**. For clarity, we have now modified the legend of Figure 2 to reflect this information.

5. In Fig 7, authors identified 4 potential linear epitopes of ORF8 and treatment with the various peptides are able to induce the expression of C1 proteins. However, this is still showing that individually these peptides can induce C1 proteins expression. Structurally, can the authors comment of the proximity between peptides #4, 10 and 25 and discuss the possibility of the how the 3 peptides could potentially work synergistically to drive the expression of Complement proteins?

Response: We thank and appreciate the reviewer for these questions and comments, which allowed us to further investigate the specific residues, potential type, and strength of binding interactions involved between ORF8 peptides and the globular C1qA domain. To address the reviewer's concern, we first validated the docking model (model 1) initially generated using HDOCK by performing a parallel protein-ligand docking of the complex in Schrodinger suite, using the crystal structures of SARS-CoV-2 ORF8 (PDB ID: 1PK6) and C1qA (PDB ID: 7JTL). We found extensive hydrogen-bonds, salt bridges, pi-pi stacking, and van der Waals interactions at the interface of C1qA and multiple residues in peptides #4, #10, #19, and #25. We used the 2D interaction diagrams to model these interactions, which are reflected in the revised **Figure 8D** panels. After identifying the amino acids, we focused on those R-groups (or side chains) whose protein-protein interactions were less than 4.0 Å, which signifies strong intermolecular binding interactions [9]. To address the proximity of the interacting residues, we also measured the amino acid distance [4] for each paired interaction and were tabulated below each panel. More importantly, we validated the results of the peptide library screening using a 3D model to elucidate the structural basis of potential synergism among the peptides #4, #10, and #25 stabilizing the complex. We found that the multiple electrostatic bonds such as hydrogen bonds and salt bridges were formed by specific C1qA that extensively bind at multiple sites located at two or more ORF8 peptides. Specifically, **Table EV3** showed eight C1qA residues distributed in both chains A and C that interact with several residues from peptides #4 and #25, peptides #4 and #10, or peptides #10 and #25. Several of these residues from peptides #4, #10, and #25 are also engaged with other C1qA residues in the globular domain that further stabilize the complex, indicating coordinate binding at multiple sites. These findings are now included in new **Figure 8D** and two additional supplementary **Tables EV3 and EV4** were appended in this manuscript.

Minor comments:

1. Consider reformatting the X and Y axes of Fig 1B to make it easier to read.

Response: We thank the reviewer for the suggestion and have made the changes as recommended in the revised manuscript.

2. Check through the spelling of the words, there are some mistakes being spotted.

Response: We apologize for the mistakes and have checked the manuscript thoroughly to correct any misspelled words in the revised manuscript.

3. Bulk RNAseq was performed on 47 tissue specimens right (Fig 2A)? In the manuscript it was written 74 samples

Response: In this study, we have indeed performed bulk RNAseq on a total of 74 samples which include chorion, amnion (**Figure 2**) and umbilical cord (**Figure EV1**). To avoid confusion, we have modified the **Results** section to clarify this information in the revised manuscript.

4. It would be good to be consistent between Fig 2E, 2F and 2G to either indicate or remove the (GO numbers).

Response: We thank the reviewer for the suggestion and have made the changes as recommended in the revised manuscript.

5. Please indicate exact P values as per EMBO requirements

Response: We thank the reviewer for the suggestion and have made the changes as recommended in the revised manuscript.

6. In Fig 3A, it should be N=3 for healthy amniotic fluid, since in Fig 3B only 3 samples are shown in the heat map. However, in Fig 3E some of the cytokines were detected from n=7 samples? Why were these additional 4 samples not included in Fig 3B heatmap?

Response: The proteomics analysis for amniotic fluid specimens were performed using three to seven controls. Due to the requirement of larger volume of specimens needed to run analyses for both complement factors and proinflammatory multiplex panels, only three specimens had sufficient volume to be included in both panels. Throughout the proteomics analysis, specimens with out-of-range protein detection was removed from the analysis, hence giving rise to slightly different number of samples as seen in **Figure 3D and 3E**. As a representative heatmap, we decided to show the three controls which we managed to run proteomics analyses for both complement and inflammatory panels.

7. Please call out figure 3F. Instead of 3F, 3E was called out instead.

Response: We thank the reviewer for the suggestion and have made the changes as recommended in the revised manuscript.

8. How many of the DEGS between Amnion and umbilical cord are common? I would assume it to be high, since they both show complement activation pathways?

Response: We thank you for this insightful suggestion. As recommended, we have conducted comparative analysis, and observed that the fetal-skewed umbilical cord and amnion are sharing 499 DEGs, including important genes related to complement (CFB) and complement-associated inflammation (NFKBIA, BCL6, CD46, and CSF3). These results are now reflected in new supplementary **Figure EV1F**

9. In Fig 4B, for those amniotic fluid that were positive, were their corresponding maternal and cord plasma similarly positive for ORF8?

Response: We have included a new supplementary **Table EV1** in the revised manuscript which provides information on the detection of SARS-CoV-2 N1/N2 RNA and circulating ORF8 for each individual maternal-fetal pair of biospecimens evaluated.

10. "we compared the transcriptomic profiles of amnion and umbilical cord tissues between the ORF8 positive (+) and ORF8 negative [7] COVID- 19-affected pregnancies. This analysis identified 400 DEGs exclusively affected in ORF8 (+) amnions (Fig. 4C)." This statement is incorrect, given that only the transcriptomics from Amnion was shown.

Response: We thank the reviewer for the suggestion and have corrected the statement as recommended in the revised manuscript.

11. Does the 6/23 maternal-fetal pair that demonstrated vertical transmission typically exhibit higher levels of proinflammatory or complement pathways as compared those that did not exhibit vertical transmission? Also, are these samples all ORF8 positive as well?

Authors: Interestingly, the frequency of SARS-CoV-2 viral RNA detection in the fetal compartment did not correlate with the robust complement activation and inflammation observed in the majority of biospecimens derived from the fetal compartment. This interesting question discussed by us in the **Discussion** section of the manuscript. Regarding the samples that shows vertical transmission by SARS-CoV-2 viral RNA detection, we observed that 50% (3/6) was ORF8 positive. We have included a new supplementary **Table EV1** in the revised manuscript which provides information on the detection of SARS-CoV-2 N1/N2 RNA and circulating ORF8 for each individual maternal-fetal pair of biospecimens evaluated.

12. Spelling error "suggestes" should be suggested

Response: We apologize for the mistake and have checked the manuscript thoroughly to correct any misspelled words in the revised manuscript.

13. Is it possible to comment or discuss on the % similarity between human IgG and immunoglobulin-like domain of ORF8?

Response: We thank the reviewers for this insightful question. We addressed the similarity of the human immunoglobulin G (IgG) and the Ig-like domain of SARS-CoV-2 ORF8 using three perspectives: (i) sequence similarity, (ii) protein folding, and (iii) mechanistic behavior and cellular tropism. Despite the remarkably divergent nature of SARS-CoV-2 ORF8 protein [10], we found that several amino acid (AA) residues within the ORF8-specific sequence showed 71–85% similarity with the human Ig variable lambda light (IGLV) chain genes. Previous work showed that SARS-CoV-2 ORF8 protein followed a predicted Ig-like folding. We used the overlapping sequences identified in Figure EVX-A to determine similarities in various protein properties, such as the buried index, hydrophobicity, polarity, turn, and helix propensity, where we found high congruence between these sequences. Most importantly, like the human IgG which can cross the histological layers of the placenta through the human IgG-Fc region [11], our study provided evidence on the transplacental transfer of ORF8 from the maternal to the fetal compartment when we detected ORF8 in the chorion and amnion of SARS-CoV-2 infected pregnant women (**Figure 4M, Figure 5A and B, Fig. EV2**). The mechanism how ORF8 crosses the placental barrier is largely unknown; however, these extensive similarities between the human IgG and SARS-COV-2 ORF8 proteins led us to hypothesize that ORF8 could access the fetal compartment by using the neonatal Fc receptor (FcRn). To gain further understanding on the 3D structural similarity and folding of the human IgG-Fc, we performed a similar pairwise structural alignment of the human IgG-Fc and ORF8 structures. We found that there exists a close structural similarity between ORF8 and the CH2 domain of the IgG-Fc, which is the IgG interface established to interact with the FcRn [12]. We have now included these findings as a new supplementary **Figure EV6** and in the **Discussion** section of the revised manuscript.

14. Although ORF8 is detected in the cord plasma, is it possible to detect for ORF8 from the blood plasma of the newborn?

Response: We agree with the reviewer that ORF8 detection in newborn plasma would be interesting. Unfortunately, due to the small volume of newborn blood collection from heel stick and regulation on the

amount of blood that can be collected from newborn, we were unable to evaluate the ORF8 levels in newborn blood which requires larger volume of specimens.

15. Able to comment if in patients with serious complications during pregnancy leading to perhaps fetal abnormalities or death, the ORF8 levels are significantly elevated as well?

Response: The authors recognize the necessity of establishing a direct correlation between ORF8 levels and the severity of COVID-19 in mothers and fetuses. However, the composition of our cohort, which consists solely of mild cases of COVID-19, restricts our ability to conduct such an analysis. To address this limitation, the following sentence was added to the last paragraph of the **Discussion**: “One limitation of the present study is that an analysis of the association between ORF8 levels and the severity of COVID-19 was not possible, given that the clinical cohort comprised only those with mild cases of COVID-19, including both mothers and newborns. Nevertheless, the clinical association between ORF8 levels, complement and inflammation, as well as the mechanisms involved in this host-virus interaction, are clear.”

16. For the M&M, is there a study protocol number that should be listed down?

Response: The study protocol number is now included in the **Methods** section of “SARS-CoV-2-exposed maternal-fetal cohort and biospecimens collection”.

References:

1. Motulsky, H.J. and R.E. Brown, *Detecting outliers when fitting data with nonlinear regression - a new method based on robust nonlinear regression and the false discovery rate*. BMC Bioinformatics, 2006. **7**: p. 123.
2. Parikh, R., et al., *Understanding and using sensitivity, specificity and predictive values*. Indian J Ophthalmol, 2008. **56**(1): p. 45-50.
3. Li, J., et al., *Comparison of reverse-transcription qPCR and droplet digital PCR for the detection of SARS-CoV-2 in clinical specimens of hospitalized patients*. Diagn Microbiol Infect Dis, 2022. **103**(2): p. 115677.
4. The human protein atlas. *Placenta - expression summary - C1qA*. [cited 2024 06/30]; Available from: <https://www.proteinatlas.org/ENSG00000173372-C1QA/tissue/placenta#img>.
5. Assadizadeh, M. and M. Azimzadeh Irani, *Oligomer formation of SARS-CoV-2 ORF8 through 73YIDI76 motifs regulates immune response and non-infusion antiviral interactions*. Front Mol Biosci, 2023. **10**: p. 1270511.
6. Center for Disease Control and Prevention (CDC). *Long COVID Basics*. 2024 [cited 2024 06/30]; Available from: <https://www.cdc.gov/coronavirus/2019-ncov/long-term-effects/index.html#:~:text=Long%20COVID%20is%20defined%20as,for%20at%20least%203%20months>.
7. Cervia-Hasler, C., et al., *Persistent complement dysregulation with signs of thromboinflammation in active Long Covid*. Science, 2024. **383**(6680): p. eadg7942.
8. Mor, G., et al., *Inflammation and pregnancy: the role of the immune system at the implantation site*. Ann N Y Acad Sci, 2011. **1221**(1): p. 80-7.
9. Tam, J.Z., et al., *Analysis of Protein-Protein Interactions for Intermolecular Bond Prediction*. Molecules, 2022. **27**(19).
10. Flower, T.G., et al., *Structure of SARS-CoV-2 ORF8, a rapidly evolving immune evasion protein*. Proc Natl Acad Sci U S A, 2021. **118**(2).
11. Ciobanu, A.M., et al., *Benefits and Risks of IgG Transplacental Transfer*. Diagnostics (Basel), 2020. **10**(8).

12. Brambell, F.W., et al., *The relative transmission of the fractions of papain hydrolyzed homologous gamma-globulin from the uterine cavity to the foetal circulation in the rabbit.* Proc R Soc Lond B Biol Sci, 1960. **151**: p. 478-82.

Dear Dr. Foo,

Congratulations on a great revision! Overall, the referees have been positive. However, referee 3 has a remaining concern and we ask that you please provide some insight into minimally the original point 1 by reviewer 2. When you submit your revised version, please also take care of the following editorial items and add this also to your point-by-point response:

1. Please add up to five keywords, which may or may not appear in the title, should be given in alphabetical order, below the abstract, each separated by a slash (/).
2. The clarify the following author name discrepancy: Oluwatosin Goje (manuscript) and Goje Oluwatosin (eJP).
3. Please ensure the ORCID ID for authors Chen and Farrell are uploaded.
4. Please move the Data Availability section to the end of the Methods section.
5. Please merge the funding section with the Acknowledgements section.
6. Please remove the Author Contribution section from the main manuscript.
7. Please rename the Conflict of Interest statement to Disclosure and Competing Interests Statement.
8. Please correct the reference format to the EMBO style: alphabetical order, 10 authors listed before et al.
9. Please upload the 4 EV tables as individual files.
10. We include a synopsis of the paper (see <http://emboj.embopress.org/>). Please provide me with a general summary statement and 3-5 bullet points that capture the key findings of the paper.
11. We also need a summary figure for the synopsis. The size should be 550 wide by 200-440 high (pixels). You can also use something from the figures if that is easier.
12. We require that all figures are referenced in the main manuscript. Please include a reference to Fig8H and Fig EV5.
13. Please move Table 2 to after the main figure legends.
14. Please change Supplementary Materials to Expanded View Figure Legend.
15. Please note that the exact p values are not provided in the legends of figures 2i-j; 3d-g; 4a, d, g-h, l; 6b-g; 8e-g; EV 1c, e; EV 3a-e.
16. Please indicate the statistical test used for data analysis in the legends of figures 2b-g.
17. Please note that in figures 4d; EV 3a-e; there is a mismatch between the annotated p values in the figure legend and the annotated p values in the figure file that should be corrected.
18. Please note that the box plots need to be defined in terms of minima, maxima, centre, bounds of box and whiskers, and percentile in the legends of figures 4d; 6d-f; 8e-g.
19. Please note that the box plots need to be defined in terms of bounds of box and whiskers, in the legends of figures 6b; EV 3a, c-e.
20. Please note that information related to n is missing in the legends of figures 1b, d; 2i-j; 3d-g; 4c-d, g-h, l; 6b-g; 8e-g; EV 1c, e.
21. Although 'n' is provided, please describe the nature of entity for 'n' in the legend of figure EV 3a.
22. Please note that the error bars are not defined in the legend of figure 1b.

Thank you for the opportunity to consider your work for publication, I look forward to your revision.

Warm wishes,
Kelly

Kelly M Anderson, PhD
Editor, The EMBO Journal
k.anderson@embojournal.org

Use the link below to submit your revision:

Referee #1:

The authors did a great job addressing my concerns and questions. One minor point to address: Figure 5: the authors suggest that ORF8 and C3b colocalize. However, review of the figure does not appear to support significant colocalization between these two proteins. Please provide additional quantitative data; for e.g., intensity plots for both channels would help in visualizing the degree of overlap and calculating and reporting Mander's coefficient and Pearson's correlation coefficient.

Referee #2:

The authors appear to have a misunderstanding of gene expression analysis. In their responses to my comments, they stated that they were unable to compare the different gene expression experiments because these were conducted on different tissues and/or under different experimental designs. However, despite these differences, it is still possible to identify common sets of genes or pathways shared across these assays. This is particularly relevant given the authors' claim that ORF8 is a major factor causing fetal inflammation. If this is the case, one would expect a significant overlap in the transcriptomic data between Figures 2 and 4. Similarly, although Figure 3 is based on proteomics, some level of overlap with the findings in Figure 2 should be anticipated, as both datasets reflect the effects of viral infection. Furthermore, the comparison between infected patients and those with Long COVID is crucial for understanding the broader implications of the study—a point the authors seem to have overlooked. Overall, the authors were unable to logically connect the experiments, resulting in a study that lacks coherence and a compelling narrative.

Referee #3:

Following the revision, the group has substantially improved on the quality of the submitted manuscript and has satisfactorily answered all my questions. To improve on the manuscript, the group took the effort to carry out additional experiments following the suggestions of all reviewers. These additional information has further strengthen the message that the authors are showing with this piece of work. In conclusion, the results published in this work would bring important findings to the COVID-19 research community.

EMBOJ-2024-116783R1

Editor's comments for authors:

General comments:

Congratulations on a great revision! Overall, the referees have been positive. However, referee 3 has a remaining concern and we ask that you please provide some insight into minimally the original point 1 by reviewer 2. When you submit your revised version, please also take care of the following editorial items and add this also to your point-by-point response:

Response: We sincerely thank the editor for opportunity to improve our manuscript and considering our manuscript for publication at EMBO Journal. All comments made by the reviewers and editor has enabled us to improve our manuscript significantly through the revisions. We have made a genuine attempt to thoroughly address all concerns that were raised by the reviewers and editor.

Specifically, as suggested, we have revisited Reviewer 2's concern from the first revision (comment 1). We have performed a comparison analysis of placental tissue transcriptomics (Fig. 2) and biofluids proteomics (Fig. 3) derived from COVID19-affected pregnancies at delivery. This is now described in the **Results** section and presented as new **Figure EV2** of the revised manuscript. In addition, as per Reviewer 1's suggestion, we have also added new colocalization analysis for Fig. 5, now presented as new **Figure EV4** of the revised manuscript.

Referee #1:

The authors did a great job addressing my concerns and questions. One minor point to address:

Figure 5: the authors suggest that ORF8 and C3b colocalize. However, review of the figure does not appear to support significant colocalization between these two proteins. Please provide additional quantitative data; for e.g., intensity plots for both channels would help in visualizing the degree of overlap and calculating and reporting Mander's coefficient and Pearson's correlation coefficient.

Response: We thank the reviewer for the positive and constructive feedback which has enabled us to improve our manuscript significantly through the revisions. As suggested by the reviewer, , to support our finding of the colocalization of ORF8 and C3b in placenta tissues, fluorescence signal was measured in the using the Fig. 5A and 5B, and analyzed by Mander's coefficient and Pearson's correlation coefficient. These results are now presented as **new Figure EV4** of the revised manuscript.

Referee #2:

The authors appear to have a misunderstanding of gene expression analysis. In their responses to my comments, they stated that they were unable to compare the different gene expression experiments because these were conducted on different tissues and/or under different experimental designs. However, despite these differences, it is still possible to identify common sets of genes or pathways shared across these assays. This is particularly relevant given the authors' claim that ORF8 is a major factor causing fetal inflammation. If this is the case, one

would expect a significant overlap in the transcriptomic data between Figures 2 and 4. Similarly, although Figure 3 is based on proteomics, some level of overlap with the findings in Figure 2 should be anticipated, as both datasets reflect the effects of viral infection. Furthermore, the comparison between infected patients and those with Long COVID is crucial for understanding the broader implications of the study—a point the authors seem to have overlooked. Overall, the authors were unable to logically connect the experiments, resulting in a study that lacks coherence and a compelling narrative.

Response: We thank the reviewer for the critical and constructive feedback which has enabled us to improve our manuscript significantly through the revisions. As suggested by the reviewer and the editor, we have revisited the reviewer's comment 1. We have now performed a comparison analysis of placental tissue transcriptomics (Fig. 2) and biofluids proteomics (Fig. 3) derived from COVID19-affected pregnancies at delivery. This is now described in the **Results** section and presented as new **Figure EV2** of the revised manuscript.

Referee #3:

Following the revision, the group has substantially improved on the quality of the submitted manuscript and has satisfactorily answered all my questions. To improve on the manuscript, the group took the effort to carry out additional experiments following the suggestions of all reviewers. This additional information has further strengthened the message that the authors are showing with this piece of work. In conclusion, the results published in this work would bring important findings to the COVID-19 research community.

Response: We thank the reviewer for the positive and constructive feedback which has enabled us to improve our manuscript significantly through the revisions.

Other editorial comments:

1. Please add up to five keywords, which may or may not appear in the title, should be given in alphabetical order, below the abstract, each separated by a slash (/).

Response: We have included keywords below the abstract, in the revised manuscript.

2. The clarify the following author name discrepancy: Oluwatosin Goje (manuscript) and Goje Oluwatosin (eJP).

Response: Our apologies for the mistake. We have corrected the author's name on eJP to reflect the correct name of Oluwatosin Goje (First name Last name) as presented in the manuscript.

3. Please ensure the ORCID ID for authors Chen and Farrell are uploaded.

Response: We thank you for the reminder, we have now checked and ensured that all three corresponding authors (Chen, Farrell and Foo) have uploaded their ORCID ID on eJP.

4. Please move the Data Availability section to the end of the Methods section.

Response: We have moved the **Data Availability** section to the end of the **Methods** section.

5. Please merge the funding section with the Acknowledgements section.

Response: As suggested, the two sections have been merged.

6. Please remove the Author Contribution section from the main manuscript.

Response: The **Author Contribution** section has been removed from the main manuscript.

7. Please rename the Conflict-of-Interest statement to Disclosure and Competing Interests Statement.

Response: The section name has been renamed as requested.

8. Please correct the reference format to the EMBO style: alphabetical order, 10 authors listed before et al.

Response: We have corrected the reference format to EMBO style.

9. Please upload the 4 EV tables as individual files.

Response: We have now uploaded the 4 EV tables as individual files on eJP.

10. We include a synopsis of the paper (see <http://emboj.embopress.org/>). Please provide me with a general summary statement and 3-5 bullet points that capture the key findings of the paper.

Response: As suggested, we have now included a synopsis of the manuscript.

11. We also need a summary figure for the synopsis. The size should be 550 wide by 200-440 high (pixels). You can also use something from the figures if that is easier.

Response: As suggested, we have now included a summary figure for the synopsis.

12. We require that all figures are referenced in the main manuscript. Please include a reference to Fig8H and Fig EV5.

Response: We apologize for the missing reference in the text. We have now referenced **Fig. EV5** in the **Discussion** section and **Fig. 8H** in **Results** section.

13. Please move Table 2 to after the main figure legends.

Response: We have moved **Table 2** to after main figure legends.

14. Please change Supplementary Materials to Expanded View Figure Legend.

Response: We have re-named **Supplementary Materials** to “**Expanded View Figure Legend**”

15. Please note that the exact p values are not provided in the legends of figures 2i-j; 3d-g; 4a, d, g-h, l; 6b-g; 8e-g; EV 1c, e; EV 3a-e.

Response: As suggested, all figures were updated with the exact p values where applicable in the revised manuscript.

16. Please indicate the statistical test used for data analysis in the legends of figures 2b-g.

Response: We have updated the figure legend to reflect the statistical test used for analysis.

17. Please note that in figures 4d; EV 3a-e; there is a mismatch between the annotated p values in the figure legend and the annotated p values in the figure file that should be corrected.

Response: As suggested, all figure legends were updated.

18. Please note that the box plots need to be defined in terms of minima, maxima, centre, bounds of box and whiskers, and percentile in the legends of figures 4d; 6d-f; 8e-g.

Response: As suggested, all figure legends were updated.

19. Please note that the box plots need to be defined in terms of bounds of box and whiskers, in the legends of figures 6b; EV 3a, c-e.

Response: As suggested, all figure legends were updated.

20. Please note that information related to n is missing in the legends of figures 1b, d; 2i-j; 3d-g; 4c-d, g-h, l; 6b-g; 8e-g; EV 1c, e.

Response: As suggested, all figure legends were updated.

21. Although 'n' is provided, please describe the nature of entity for 'n' in the legend of figure EV 3a.

Response: As suggested, the legend for Fig. EV3a was updated.

22. Please note that the error bars are not defined in the legend of figure 1b.

Response: As suggested, the legend for Fig.1b was updated.

Dear Dr. Foo,

Congratulations on an excellent manuscript, I am pleased to inform you that your manuscript has been accepted for publication in the EMBO Journal. Thank you for your comprehensive response to the referee concerns and for providing detailed source data. It has been a pleasure to work with you to get this to the acceptance stage.

I will begin the final checks on your manuscript before submitting to the publisher next week. Once at the publisher, it will take about 3 weeks for your manuscript to be published online. As a reminder, the entire review process including referee concerns, and your point-by-point response will be available to readers.

I will be in touch throughout the final editorial process until publication. In the meantime, I hope you find time to celebrate!

Yours sincerely,
Kelly

Kelly M Anderson, PhD
Editor, The EMBO Journal
k.anderson@embojournal.org
